# Dolphin: A multimodal large language model for Ultrasound Understanding

## Abstract

Ultrasound is one of the most widely used imaging modalities in clinical practice. Unlike CT and MRI, ultrasound imaging is highly operator dependent, with significant variations across different anatomical regions. Therefore, the ultrasound domain has a significant need for a comprehensive understanding of general ultrasound imaging. To address this, we introduce Dolphin, the first multimodal large language model for ultrasound understanding, including chat and reasoning version. We curate a training dataset with over 2,000,000 instruction–response pairs, integrating domain-specific knowledge in the field of ultrasound, including textbooks, specific clinical guidelines, and public ultrasound datasets, along with synthetic samples and general-domain corpora. Meanwhile, we established the Dolphin Ultrasound Data Protocol to standardize various types of ultrasound data, ensuring consistency, interoperability, and quality across the dataset. While most multimodal medical models do not emphasize medical reasoning capabilities, ultrasound understanding demands strong comprehension and reasoning abilities due to the inherent complexity and variability of ultrasound data. However, real-world ultrasound reasoning data is scarce and difficult to collect, which limits the development of models with advanced understanding. To address this, We propose a three-stage training strategy using easily accessible ultrasound question-answering data and synthetic deep-reasoning general-domain data, combining post-training, instruction tuning, and Ultrasound Answer Reward Preference Optimization based reinforcement learning to progressively improve reasoning. On the U2-Bench benchmark across eight clinical ultrasound tasks, Dolphin establishes a new state-of-the-art with a U2-score of 0.5835. Our experiments show that, with reasoning data from other domains, Dolphin exhibits robust reasoning capabilities on complex ultrasound tasks. Moreover, Dolphin achieves higher diagnostic accuracy in deep reasoning mode than in standard mode, indicating that generalized reasoning skills can be effectively transferred to specialized medical domains.[1]

## 1 Introduction

Ultrasound is widely used in clinical practice due to its real-time capability, portability, and cost-effectiveness(Hewson &Bedforth, 2023). However, ultrasound interpretation remains challenging compared to CT, MRI, and PET which provide standardized views and consistent quality. Ultrasound is operator-dependent, susceptible to artifacts, and exhibits significant variability(Sharma et al. (2021)). Diagnostic accuracy relies heavily on individual expertise and training.

Currently, multimodal large language models (MLLMs) have demonstrated significant potential in the medical field, offering advanced capabilities for integrating and analyzing diverse data types, including medical images (MRI,CT,PET; Lee et al. (2024); Volkov et al. (2025); Shui et al. (2025)), electronic health records(AlSaad et al. (2024)), genomic data(Fallahpour et al. (2025)), and clinical notes(Han et al. (2024)). While these models have achieved remarkable success in processing static, high-resolution medical images and structured clinical data, their application to ultrasound imaging presents unique challenges that remain largely unaddressed.

The translation of multimodal approaches to ultrasound is hindered by several critical limitations. First, unlike the standardized acquisition protocols for CT or MRI, ultrasound examinations are

---

[1]https://anonymous.4open.science/r/Dolphin-Ultrasound-DC44/

highly operator-dependent, resulting in significant variability in image quality and anatomical views (Stasi &Ruoti (2015)). Second, the dynamic nature of ultrasound video sequences - often containing crucial temporal information about tissue motion and blood flow - is fundamentally different from the static images that current MLLMs typically process (Szabo (2013)). Third, the characteristic speckle noise and lower signal-to-noise ratio of ultrasound images compared to other modalities create additional challenges for feature extraction and interpretation (Loizou &Pattichis (2022)). Furthermore, existing MLLMs struggle with the sparse and often subjective nature of ultrasound reports, which typically lack the detailed anatomical descriptions found in radiology reports for other imaging modalities(Wang et al. (2024)). Given the aforementioned challenges, developing MLLMs specifically tailored for ultrasound imaging has become an urgent necessity.

Medical ultrasound understanding requires precise perception of visual features, along with sophisticated reasoning to interpret dynamic, operator-dependent imaging and derive clinically meaningful conclusions. However, most existing multimodal medical models treat ultrasound tasks in isolation and focus primarily on perceptual accuracy, neglecting the development of systematic reasoning capabilities. Moreover, the scarcity of real-world diagnostic reasoning data in clinical settings poses a significant barrier to training models with robust medical reasoning skills. To address these challenges, we introduce Dolphin, a general-purpose multimodal ultrasound understanding model capable of jointly interpreting ultrasound images, videos, dynamic scanning sequences, and associated clinical text. By leveraging a three-stage training framework with hybrid reasoning, Dolphin achieves deep cognitive capabilities in ultrasound analysis.

To support model development, we curate a large-scale multimodal ultrasound dataset containing over 2,000,000 high-quality samples sourced from medical textbooks, public repositories, clinical guidelines, and synthetic data. Additionally, we propose the Dolphin Ultrasound Data Protocol (DUDP), a standardized data interface that unifies and formalizes a wide range of ultrasound tasks, unifying ultrasound tasks into four categories: classification, detection, regression, and generation.

To effectively train a medical ultrasound language model with strong generalization and diagnostic reasoning capabilities, we adopt a progressive three-stage training strategy. The first stage leverages post-training on mixture of ultrasound-specific and general-domain data to build both medical foundation and broad language understanding. The second stage focuses on instruction tuning with multi-turn dialogues and deep-reasoning tasks, enhancing the model's ability to engage in complex clinical discussions and perform step-by-step analysis. The final stage introduces UARPO, a reinforcement learning method guided by Ultrasound Answer Reward (UAR), which combines assessment of formatted answers and alignment with true diagnostic outcomes.

Dolphin achieves state-of-the-art performance with a U2-Bench (Le et al., 2025b) score of 0.5835, outperforming all compared methods across eight ultrasound tasks. During the first stage of post-training, we train the model using easily accessible ultrasound QA data along with pure-text deep reasoning datasets from other domains. We observe that the model spontaneously developed a deep reasoning capability in ultrasound vision. We unify vanilla and deep reasoning within a Bayesian framework by modeling the latent chain-of-thought variable $c$. By introducing mode-dependent priors $P_\theta(c \mid x, m)$, we theoretically characterize cross-domain reasoning transfer: although deep reasoning data lacks target-domain samples, structured reasoning patterns generalize via shared parameters and prior bias. Our results highlight a critical threshold in reasoning data volume: effective reasoning emerges only when training exceeds this point. Meanwhile, experimental results show that deep reasoning mode improves diagnostic accuracy by 2.4%, reduces measurement RMSE by 10.6%, and increases detection accuracy by 16%. We find that base model performance determines the UARPO phase ceiling, with stronger foundation models achieving higher reinforcement learning gains.

Our key contributions can be summarized as follows:

- We introduce Dolphin, the first multimodal large language model tailored for ultrasound reasoning, with chat and reasoning versions. We also build a large-scale ultrasound dataset of over 2 million samples covering 16 body regions to enhance the model's foundational capabilities.

- We adopt a three-stage training strategy with mixed reasoning and UARPO-based reinforcement learning, using deep-reasoning data to improve the model's visual reasoning for ultrasound. Ultimately, Dolphin successfully develops an emergent capacity for deep reasoning about ultrasound.

- Dolphin achieves state-of-the-art performance on U2-Bench, surpassing all existing models across all six tasks with a remarkable U2-Score of 0.58, while also demonstrating strong generalization capabilities on mainstream benchmark datasets. Notably, Dolphin has been successfully deployed on real-world ultrasound devices, marking a significant step toward practical clinical integration by bringing AI directly into physicians' daily workflows.

# 2 DATA CURATION

## 2.1 POST-TRAINING

To develop expert-level ultrasound understanding capabilities, we construct a large-scale, high-quality multimodal dataset from four primary sources: ultrasound teaching materials, public ultrasound datasets, general medical data, and distilled knowledge data. We augment this collection with synthetic data generated through structured question templates and knowledge distillation techniques. Our data generation pipeline incorporates medical expert validation at multiple stages to ensure clinical accuracy and minimize hallucination risks.

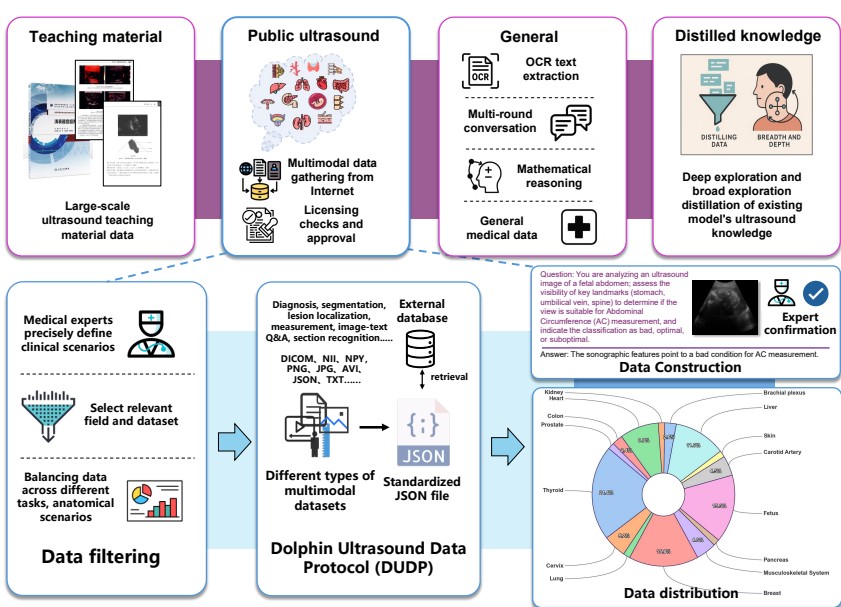

Figure 1: Above: Distribution of the large-scale dolphin ultrasound dataset. Below: Public data collection process, including data cleaning, standardization via the Dolphin Ultrasound Data Protocol (DUDP), data construction, and expert validation. The bottom-right corner shows the data distribution across different anatomical regions.

### 2.1.1 ULTRASOUND TEACHING MATERIAL

We curate classic ultrasound textbooks and industry guidelines, converting them into standardized format after quality control preprocessing. From these materials, we construct three specialized datasets as follows: (1) **Ultrasound Question-Answering Dataset**, where textbook content was segmented and processed with GPT to generate contextual questions, with self-distillation ensuring question-answer consistency; (2) **Ultrasound Caption Dataset**, where systematic image-caption pairs were structured through automated matching to ensure precise correspondence, including multi-image cases for enhanced comprehension capabilities; (3) **Ultrasound Image-Text Explanation Dataset**, where medical image descriptions were processed through GPT self-distillation to generate question-answer pairs, ensuring accuracy while enhancing response diversity and instruction-following capabilities. Throughout this process, we leverage LLM for question-answer pair generation and caption diversity enhancement, detailed prompt templates provided in Appendix K.1.

### 2.1.2 PUBLIC ULTRASOUND DATASETS

To enhance the model's diagnostic capabilities across various tasks, we collect a substantial number of publicly available ultrasound datasets and establish a unified data processing pipeline. We develop a unified file standard called **Dolphin Ultrasound Data Protocol (DUDP)** to harmonize diverse ultrasound tasks by converting them into a standardized JSON format (detailed in Appendix C). The protocol comprehensively handles diverse ultrasound tasks, supporting multimodal data types including ultrasound images, video sequences, segmentation masks, and quantitative measurements.

Specifically, we categorize ultrasound tasks into the following standardized classes: (1) **Anatomical Recognition** (2) **Standard View Identification** (3) **Diagnostic Classification** (4) **Tissue Segmentation** (5) **Biometric Measurement**. Corresponding ultrasound question-answering datasets are constructed for each task category. To mitigate hallucination risks, we develop over a hundred fixed question templates for each task category. For each sample, we randomly select a question template and populate it with JSON-formatted content, thereby constructing a high-quality ultrasound question-answering dataset. Appendix G shows some of the constructed QA pairs.

Furthermore, to ensure data diversity, we employ GPT for partial data augmentation and utilize another LLM to evaluate response quality, guaranteeing that the JSON-formatted content remains embedded within the generated responses. Appendix K.2 presents the prompt templates we adopted, including formatted QA templates, question enhancement templates, and various prompts used in general data processing.

### 2.1.3 GENERAL DATA

To enhance the model's general capabilities while incorporating ultrasound-specific knowledge, we select corresponding public datasets based on different competency requirements. The selected public datasets are primarily utilized to enhance the following model capabilities: (1) **Comprehension**; (2) **Conversational**; (3) **Deep reasoning**; (4) **Medical**; (5) **Image-text dialogue**; The primary public datasets adopted in our study are as follows: LLaVA-One-Vision. (Li et al., 2024), AutoIF. (Dong et al., 2024), Alpaca-GPT4 (Liu et al., 2023), Medical-O1-Reasoning (Chen et al., 2024). These datasets are meticulously curated to comprehensively enhance the model's performance across multiple dimensions.

### 2.1.4 DISTILLED KNOWLEDGE DATA

Following the methodology of (Xu et al., 2023a), we first construct a set of ultrasound data question templates. The difficulty level of questions is enhanced through depth-wise augmentation, while their diversity is improved via breadth-wise expansion. All question categories are strictly standardized within the ultrasound medical domain, and we distill ultrasound-specific data from both GPT and DeepSeek. Additionally, we design a data screening pipeline that employs large language models to assess the logical consistency of outputs, ultimately yielding a high-quality distilled dataset.

## 2.2 DATA FILTERING

To ensure the quality and reliability, we implement a rigorous data filtering pipeline consisting of multiple stages. We employ LLM-based semantic filtering with GPT-4 to assess relevance, accuracy, and logical consistency of responses. Rule-based filtering removes incorrectly generated samples through character matching in JSON files and textbook captions, while screening for proper format by filtering out samples with overly short responses or excessive images. Expert validation further assesses the rationality of dataset construction through medical expert evaluation. Specifically, our medical experts consist of three ultrasound physicians, each with at least five years of clinical experience. These experts are responsible for validating the quality of the constructed question-answer (QA) pairs, focusing on linguistic style, overall coherence, and whether the diagnostic information is adequately reflected. To ensure high standards, we require full agreement among all three physicians for a QA pair to be accepted; in cases of disagreement, we iteratively refine the data construction process accordingly. The expert review primarily focuses on QA pairs derived from textbooks and publicly available datasets. For public datasets, we exclusively select those accompanied by ground-truth annotations to ensure accuracy. For textbook-derived data, the physicians verify that

each QA pair is logically sound and that both questions and answers are faithfully extracted from the source material. The consistency verification templates are further detailed in Appendix K.2.

# 3 MODEL TRAINING

Dolphin is built upon the Qwen2.5-VL architecture (Bai et al., 2025), utilizing a redesigned Vision Transformer for enhanced visual processing. We train two model variants with 7B and 72B parameters through a three-stage progressive framework: (1) **Post-training**, (2) **Instruction Tuning**, and (3) **UARPO-based Reinforcement Learning**. The first two stages produce the **Dolphin Chat** model, while the third stage yields the **Dolphin Reasoning** model with enhanced reasoning capabilities.

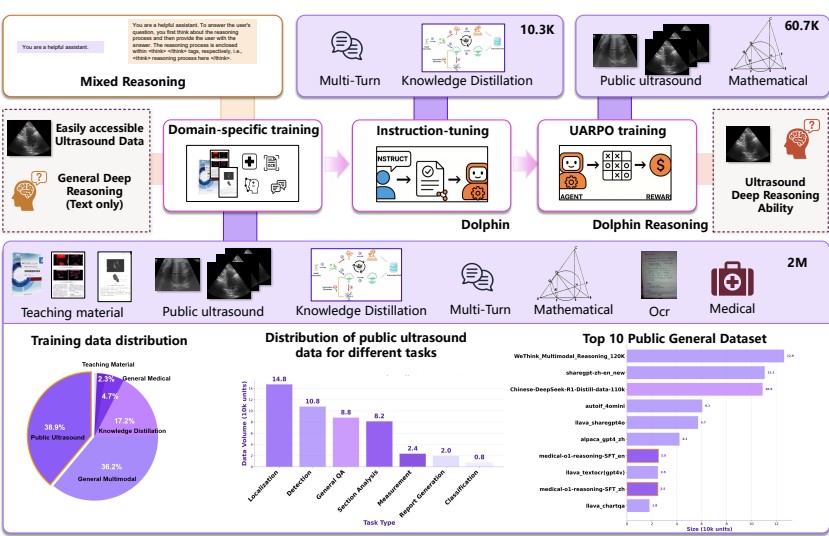

Figure 2: The training pipeline of the Dolphin series models consists of three key stages: domain-specific pre-training on over 2,000,000 multimodal data samples, instruction fine-tuning using 10,300 carefully selected samples to enhance the model's multi-turn dialogue and reasoning capabilities, and UARPO reinforcement learning based on 60,700 interactions to further improve reasoning and decision-making abilities.

## 3.1 POST-TRAINING

In the post-training stage, we inject ultrasound-specific knowledge while preserving general capabilities. The model is trained on a total of 2.03 million paired samples, comprising ultrasound textbooks, public datasets, and diverse general-domain data to develop domain understanding and maintain core competencies including **reasoning**, **multi-turn dialogue**, **instruction-following**, **OCR detection**, and **mathematical reasoning**. Specifically, the composition of the training data includes 38.9% from public datasets, 36.2% general-domain data, 17.2% from knowledge distillation, 4.7% general medical data, 2.3% textbook-derived data, and 20% dedicated reasoning data (see Figure 2 for detailed distribution). This balanced approach prevents catastrophic forgetting while enabling effective domain specialization. We observe that proportional increases in both ultrasound-related and general data lead to synergistic improvements across all capabilities, indicating a positive interaction between domain-focused learning and broad cognitive skill retention.

## 3.2 INSTRUCTION TUNING

The instruction tuning stage enhances the model's deep reasoning and conversational abilities through curated instructional data. We construct a small-scale instruction dataset consisting of 10.3k samples, including 1.4k multi-turn dialogue instances and 8.9k distilled instruction-following examples. This dataset leverages distilled knowledge to teach structured response patterns and improve dialogue

coherence. Fine-tuning employs a small learning rate and a single epoch to preserve existing capabilities while enhancing dialogue quality and reasoning depth.

### 3.2.1 UARPO-BASED REINFORCEMENT LEARNING

The final stage employs Group Relative Policy Optimization (GRPO) (Shao et al., 2024) with Ultrasound Answer Reward (UAR) to enhance reasoning. Instead of supervised fine-tuning, GRPO enables the model to autonomously explore reasoning paths using a 60.7k dataset: 58.6k samples from our curated public data (1k per task) and 2.1k from Geometry3K for geometric reasoning. UAR provides reward signals that guide preference learning, improving robustness in medical diagnosis.

**GRPO Algorithm.** Given a policy $\pi_\theta$ and a group of $K$ responses $\{y_1, y_2, ..., y_K\}$ for prompt $x$, GRPO optimizes the objective:

$$\mathcal{L}_{\text{GRPO}} = \mathbb{E}_{x, y_i \sim \pi_\theta(\cdot|x)} \left[ \frac{\exp(r(x, y_i))}{\sum_{j=1}^{K} \exp(r(x, y_j))} \log \pi_\theta(y_i|x) \right] \tag{1}$$

where $r(x, y)$ represents the reward function.

**UAR Design.** Our Ultrasound Answer Reward consists of two components with equal weights:

$$\text{UAR}(x, y) = 0.5 \times R_{\text{format}}(y) + 0.5 \times R_{\text{outcome}}(x, y) \tag{2}$$

where $R_{\text{format}}(y) = 1$ if the response follows the required format, and $R_{\text{outcome}}(x, y) = 1$ if the answer matches the ground truth. Ultrasound imaging exhibits unique complexity and versatility in the medical imaging domain, differing significantly from other modalities (e.g., MRI, CT). Its imaging scope covers multiple anatomical regions across the body. Based on the clinical workflow characteristics of ultrasound examinations, we systematically abstract all tasks into four core categories: View Identification, Disease Diagnosis, Lesion Localization, and Biometric Measurement. We convert these four types of tasks into two verifiable reward outcomes: (1) definitive multiple-choice answers (A/B/C/D) for classification tasks, and (2) accurate diagnostic conclusions for open-ended medical assessments including disease identification, anatomical localization, and quantitative measurements. This binary reward system ensures both structural compliance and clinical accuracy.

## 4 MIXED REASONING STIMULATES MEDICAL THINKING

### 4.1 MIXED INFERENCE STIMULATES THE MODEL'S ULTRASONIC VISUAL REASONING ABILITY

We adopt a single model trained under two system prompts, thereby inducing a *hybrid reasoning* capability. The training employs a sample-level mixture about of 0.8:0.2 between vanilla and deep-reasoning data. For details on the system prompts, refer to Appendix A.3.

Although the deep-reasoning subset contains no ultrasound-related samples, after joint training the model displays improved reasoning on ultrasound diagnosis. This indicates that conditioning on a reasoning-oriented prompt induces a structured prior over latent chains of thought, which can transfer across domains even without in-domain supervision.

### 4.2 BAYESIAN INTERPRETATION: PROMPT-CONDITIONED TRANSFER

Let the input be $x$, label be $y$, and the prompt-conditioned mode be $m \in \{q, r\}$ (vanilla vs. deep-reasoning). We introduce a latent variable $c$ representing a chain of thought. The predictive distribution can be expressed as

$$P_\theta(y \mid x, m) = \sum_c P_\theta(y \mid x, c) P_\theta(c \mid x, m). \tag{3}$$

Training uses a mixed distribution over modes $\mathcal{D}_{\text{mix}} = (1 - \gamma) \mathcal{D}_q + \gamma \mathcal{D}_r$ and maximizes the expected log-likelihood. The deep-reasoning prompt modifies the prior over chains $P_\theta(c \mid x, m = r)$, biasing toward structured and reliable reasoning. Although the deep-reasoning subset lacks ultrasound samples, parameters are shared across modes; thus the likelihood term $P_\theta(y \mid x, c)$ learned from vanilla data transfers to the deep-reasoning mode.

The posterior over chains given correctness satisfies $P_\theta(c \mid x, y, m) \propto P_\theta(y \mid x, c) P_\theta(c \mid x, m)$, so when $m = r$ concentrates prior mass on generally useful chains, the posterior selects such chains even in the ultrasound domain, explaining the observed gains.

### 4.3 FROM INTERFERENCE TO TRANSFER: CROSS-DOMAIN REASONING DYNAMICS

Our experimental findings reveal a striking non-monotonic relationship between reasoning data proportion $\gamma$ and diagnostic accuracy: initial increases in reasoning data cause performance degradation, followed by recovery and significant improvement beyond baseline. This counterintuitive pattern unveils fundamental insights into cross-domain reasoning transfer mechanisms. Appendix A.5 presents the model performance scores under different mixing ratios.

#### 4.3.1 THE INTERFERENCE-TO-TRANSFER TRANSITION

As reasoning data grows, the model shows two shifting performance trends:

**Initial Interference Phase ($\gamma$ small):** When reasoning data is insufficient, the model fails to learn reliable reasoning priors $P_\theta(c \mid x, m = r)$. This leads to *negative interference*, where the reasoning mode selects inappropriate reasoning patterns despite having access to correct medical knowledge from vanilla training. The model essentially "overthinks" with poor reasoning strategies, degrading performance below vanilla baseline.

**Positive Transfer Phase ($\gamma$ sufficient):** Beyond a critical threshold, sufficient cross-domain reasoning data enables the model to learn discriminative priors that concentrate on effective reasoning patterns. These domain-general reasoning skills exhibit remarkable transferability to medical diagnosis, ultimately achieving superior performance through enhanced analytical capabilities.

#### 4.3.2 THEORETICAL FOUNDATION: POSTERIOR OPTIMIZATION DYNAMICS

The transition mechanism centers on optimizing posterior probabilities of beneficial reasoning chains:

$$\text{Reasoning mode:} \quad \max_\theta P_\theta(c_{\text{med}} \mid x, y, m = r) \tag{4}$$

$$\text{Vanilla mode:} \quad \max_\theta P_\theta(c_{\text{gt}} \mid x, y, m = q) \tag{5}$$

where $c_{\text{med}}$ represents medically appropriate reasoning chains and $c_{\text{gt}}$ represents vanilla mode reasoning patterns.

The performance transition occurs when reasoning data sufficiency enables: $P_\theta(c_{\text{med}} \mid x, y, m = r) > P_\theta(c_{\text{gt}} \mid x, y, m = q)$, signifying that reasoning mode successfully selects superior analytical pathways compared to vanilla responses.

#### 4.3.3 CROSS-DOMAIN REASONING UNIVERSALITY

The key theoretical insight underlying successful transfer is that *effective reasoning exhibits domain-invariant characteristics*. Analytical patterns that consistently produce correct outcomes across mathematics, science, and logical reasoning domains also enhance medical diagnostic capabilities. This universality explains why non-medical reasoning training can ultimately improve medical performance once sufficient data enables proper prior learning.

This finding has profound implications for medical AI development: rather than requiring domain-specific reasoning data, models can leverage diverse reasoning experiences to develop transferable analytical capabilities that enhance medical understanding.

The detailed mathematical analysis, including maximum likelihood dynamics and statistical learning theory foundations, is provided in Appendix A.

## 5 EXPERIMENT

We evaluate Dolphin on comprehensive benchmarks spanning ultrasound-specific tasks, general vision understanding, and cross-modal medical imaging. For ultrasound evaluation, we use U2-Bench Le et al. (2025a), a specialized benchmark covering eight clinical tasks. General vision

capabilities are assessed using standard benchmarks including MMBench Xu et al. (2023b) for multimodal reasoning, MME Zhang et al. (2021) for comprehensive evaluation, SEED-Bench Li et al. (2023a) for generative comprehension, and TextVQA Singh et al. (2019) for text reading abilities. Medical imaging performance across modalities is evaluated using MedFrameQA Yu et al. (2025) for clinical reasoning and radiology VQA datasets Lau et al. (2018) for cross-modal understanding. Unless otherwise specified, we use a temperature of 0, top-p of 0.001, a maximum generation length of 2048 tokens, and keep the system prompt consistent across all evaluations. The training setup for Dolphin is detailed in Appendix E. The results comparing our model with GPT are presented in Appendix I.3. Unless otherwise specified, we use a default temperature of 0, top-p sampling set to None, a maximum generation length of 2048 tokens, and keep the system prompt consistent across all evaluations to ensure fair and reproducible comparisons.

Table 1: Results of different models on the U2-Bench.

| Models | DD | | VRA | | LL | OD | KD | CVE | | | RG | | | CG | | | U2-Score |
|---|---|---|---|---|---|---|---|---|---|---|---|---|---|---|---|---|---|
| | Acc. | F1 | Acc. | F1 | Acc. | Acc. | Acc. | RMSE | MAE | %tol | BLEU% | Rouge% | BERT% | BLEU% | Rouge% | BERT% | |
| Random Guessing | 0.4143 | 0.4135 | 0.3195 | 0.3184 | 0.1118 | 0.0680 | 0.1120 | 0.5472 | 0.4352 | 18.776 | - | - | - | - | - | - | 0.2125 |
| *Medical-Specific Models* | | | | | | | | | | | | | | | | | |
| MiniGPT-Med | 0.3468 | 0.2828 | 0.1800 | 0.1048 | 0.1728 | 0.1789 | 0.0840 | 0.3056 | 0.2600 | 33.2259 | 6.4700 | 20.1300 | 74.6900 | 30.2000 | 47.7500 | 80.5000 | 0.2375 |
| MedDr | 0.4508 | 0.3118 | 0.2071 | 0.1214 | 0.0720 | 0.0881 | 0.0900 | 0.2144 | 0.1786 | 38.2642 | 2.7998 | 13.5060 | 72.2050 | 33.4939 | 49.6236 | 81.2078 | 0.2373 |
| Lingshu-7B | 0.4589 | 0.2755 | 0.2625 | 0.1490 | 0.1265 | 0.2005 | 0.1140 | 0.2581 | 0.1908 | 27.8302 | 1.9974 | 15.7764 | 67.8138 | 4.0058 | 12.3106 | 62.0800 | 0.2704 |
| MedGemma-4B-it | 0.5005 | 0.4336 | 0.3071 | 0.1520 | 0.2750 | 0.0858 | 0.0200 | 0.1667 | 0.1316 | 55.0962 | 1.5360 | 15.0348 | 74.0205 | 4.8777 | 35.9803 | 76.7859 | 0.2668 |
| llava-med-v1.5-mistral-7b30 | 0.3036 | 0.2249 | 0.2078 | 0.1701 | 0.4710 | 0.4002 | 0.4740 | 0.3656 | 0.3444 | 22.3764 | 5.8686 | 17.8618 | 71.9013 | 2.9986 | 17.0438 | 70.9995 | 0.3268 |
| *Open-Source Multimodal Models* | | | | | | | | | | | | | | | | | |
| Qwen-2.5-VL-3B-Instruct | 0.4503 | 0.3591 | 0.2097 | 0.1492 | 0.0696 | 0.0649 | 0.0894 | 0.5008 | 0.4519 | 18.9055 | 3.5018 | 15.0327 | 72.8419 | 27.6748 | 44.7618 | 79.8849 | 0.2095 |
| Qwen-2.5-VL-7B-Instruct | 0.4821 | 0.3860 | 0.2181 | 0.1665 | 0.0750 | 0.0704 | 0.1000 | 0.4646 | 0.4337 | 19.7115 | 3.7100 | 15.5600 | 73.1500 | 29.4400 | 47.0000 | 81.1500 | 0.2235 |
| Qwen-2.5-VL-32B-Instruct | 0.4812 | 0.3860 | 0.2864 | 0.2071 | 0.1700 | 0.0755 | 0.0880 | 0.3414 | 0.3015 | 27.4038 | 1.1900 | 13.0100 | 68.1400 | 14.7700 | 38.6800 | 77.3900 | 0.2449 |
| Qwen-2.5-VL-72B-Instruct | 0.4895 | 0.4556 | 0.2559 | 0.1789 | 0.1150 | 0.0660 | 0.0860 | 0.3224 | 0.2733 | 37.9370 | 3.0900 | 15.0600 | 72.6600 | 28.1600 | 44.2800 | 80.9100 | 0.2421 |
| DeepSeek-VL2 | 0.4126 | 0.3190 | 0.2268 | 0.1111 | 0.2950 | 0.1682 | 0.1320 | 0.2956 | 0.2505 | 12.3355 | 7.4700 | 20.5400 | 75.3800 | 11.4200 | 34.8500 | 77.2400 | 0.2630 |
| InternVL3-9B-Instruct | 0.4447 | 0.3716 | 0.1926 | 0.1083 | 0.3000 | 0.1416 | 0.0940 | 0.2429 | 0.1733 | 50.8738 | 2.1600 | 14.7000 | 72.2100 | 21.5900 | 43.1300 | 80.9800 | 0.2566 |
| LLaVA-1.5-13B | 0.4321 | 0.3055 | 0.1731 | 0.0755 | 0.1700 | 0.1259 | 0.1100 | 0.2307 | 0.1976 | 24.7964 | 6.2400 | 18.5800 | 73.7900 | 10.8300 | 29.4000 | 75.5000 | 0.2378 |
| Phi-4-Multimodal-Instruct | 0.3686 | 0.1148 | 0.2452 | 0.0537 | 0.0350 | 0.0815 | 0.1060 | 0.2249 | 0.2006 | 16.1972 | 3.2700 | 16.5800 | 73.2700 | 3.8700 | 22.9800 | 73.0800 | 0.2168 |
| Mistral-Small-3.1-24B-Inst. | 0.4359 | 0.0936 | 0.1964 | 0.0664 | 0.1300 | 0.0910 | 0.1060 | 0.1675 | 0.1331 | 45.9459 | 1.8000 | 14.9000 | 71.7200 | 20.7700 | 42.1200 | 80.7400 | 0.2356 |
| *Closed-Source Multimodal Models* | | | | | | | | | | | | | | | | | |
| Doubao-1.5-Vision-Pro-32k | 0.5580 | 0.2597 | 0.2922 | 0.2147 | 0.1700 | 0.0729 | 0.1240 | 0.3664 | 0.3377 | 33.1731 | 0.7100 | 6.6450 | 72.4000 | 8.6400 | 33.3000 | 78.4200 | 0.2587 |
| GPT-4o-Mini | 0.4924 | 0.3784 | 0.1922 | 0.1272 | 0.1357 | 0.0846 | 0.0960 | 0.2267 | 0.1976 | 19.2308 | 4.9400 | 17.5200 | 74.1300 | 11.7300 | 36.2900 | 77.5300 | 0.2388 |
| GPT-4o | 0.4928 | 0.4132 | 0.1504 | 0.0974 | 0.1161 | 0.0850 | 0.0840 | 0.3712 | 0.3527 | 15.7895 | 2.6800 | 14.7700 | 73.3500 | 33.7700 | 49.9600 | 81.5800 | 0.2253 |
| GPT-5 | 0.5366 | 0.4590 | 0.4573 | 0.3550 | 0.2662 | 0.1767 | 0.1080 | 0.3097 | 0.1878 | 36.1867 | 1.0641 | 8.7440 | 66.8302 | 7.9669 | 23.3116 | 72.2203 | 0.3250 |
| Gemini-1.5-Pro | 0.3781 | 0.2247 | 0.0909 | 0.0476 | 0.2700 | 0.0661 | 0.0980 | 0.2772 | 0.2205 | 40.7051 | 0.5800 | 9.9400 | 70.5500 | 28.5800 | 45.9200 | 80.0200 | 0.1999 |
| Gemini-2.0-Pro-Exp | 0.4925 | 0.4194 | 0.1648 | 0.1323 | 0.1714 | 0.0945 | 0.0820 | 0.1945 | 0.1498 | 53.3333 | 0.2600 | 6.9200 | 40.2400 | 31.1800 | 48.6000 | 81.6000 | 0.2438 |
| Gemini-2.5-Pro-Preview | 0.4256 | 0.3112 | 0.2098 | 0.1493 | 0.2709 | 0.2714 | 0.2518 | 0.2937 | 0.2672 | 34.4970 | 5.5030 | 18.0180 | 74.4930 | 15.0110 | 38.0070 | 75.9890 | 0.2968 |
| Claude-3.7-Sonnet | 0.2121 | 0.0449 | 0.1453 | 0.0479 | 0.1356 | 0.0540 | 0.0760 | 0.1764 | 0.1500 | 36.0215 | 0.6900 | 12.2300 | 68.7400 | 1.2900 | 16.6600 | 71.6600 | 0.1596 |
| Qwen-Max | 0.4566 | 0.2676 | 0.1925 | 0.0871 | 0.1606 | 0.0761 | 0.0940 | 0.1248 | 0.0843 | 69.2308 | 3.5000 | 17.0200 | 73.9600 | 30.6700 | 49.0000 | 82.5500 | 0.2445 |
| Dolphin-7B | 0.6704 | 0.5743 | 0.6681 | 0.5329 | 0.3185 | 0.4380 | 0.2200 | 0.3131 | 0.2304 | 30.4534 | 2.7357 | 14.8181 | 73.0423 | 41.4477 | 58.5160 | 89.3257 | 0.4905 |
| Dolphin | 0.6530 | 0.5667 | 0.6900 | 0.5791 | 0.4028 | 0.5302 | 0.3620 | 0.2717 | 0.2454 | 39.0725 | 3.8512 | 16.4303 | 74.1854 | 49.0289 | 64.2466 | 91.1290 | 0.5390 |
| Dolphin Reasoning | 0.6819 | 0.5155 | 0.6943 | 0.5821 | 0.4775 | 0.6003 | 0.5080 | 0.2430 | 0.2273 | 38.6458 | 3.2193 | 15.1170 | 72.7287 | 54.0634 | 76.0111 | 92.9601 | 0.5835 |

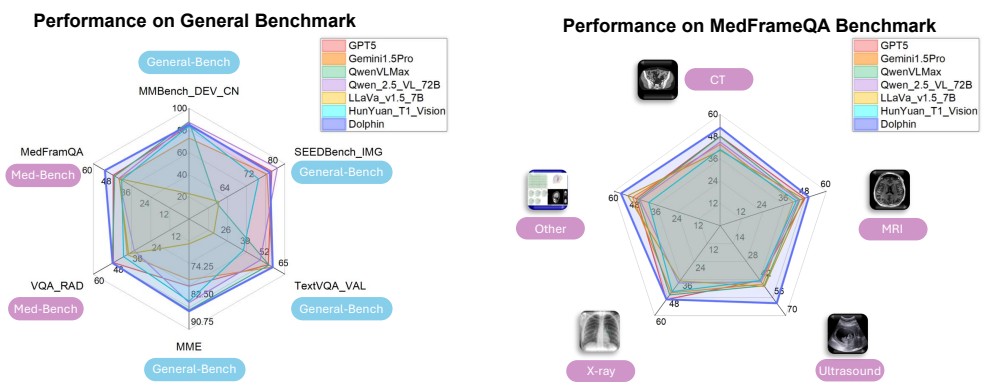

Figure 3: Model performance on General & Medical Bench

## 5.1 PERFORMANCE OVERVIEW

**U2-Bench Analysis.** Dolphin Reasoning establishes new state-of-the-art performance with a U2-Score of 0.5835, significantly outperforming existing models. The model excels in visual understanding tasks, achieving top performance in View Recognition and Assessment, Lesion Localization,

Organ Detection, and Keypoint Detection. Performance limitations appear in Clinical Value Estimation and Report Generation, indicating areas for future improvement.

**General Capability Analysis.** On MedFrameQA medical imaging benchmarks, Dolphin achieves superior ultrasound performance (55.76 vs. 47.29 next best) while maintaining strong CT (51.61) and MRI (51.95) capabilities. For general vision, Dolphin surpasses Gemini-1.5-Pro and Qwen2.5-VL-72B on MMBench (85.31), TextVQA (61.73), and MME (87.54), demonstrating that domain-specific training enhances rather than compromises general visual understanding. Comprehensive benchmark results are detailed in Appendix D.

## 5.2 EFFECT OF DEEP REASONING MODE

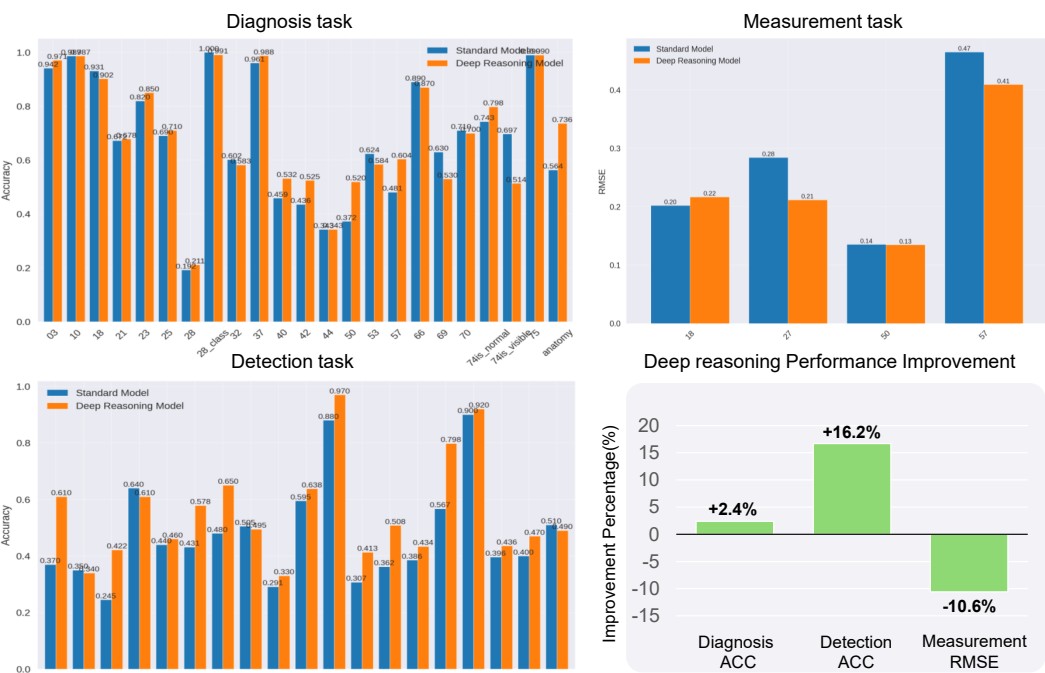

Figure 4: Comparison of Standard and Deep Reasoning mode.

We compare the model's outputs in deep thinking mode, indicated by parsing </think>, with those in standard mode to evaluate its performance on diagnostic, detection, and measurement tasks using the U2-score. Figure 4 presents the test results. The results show that, in deep thinking mode, the model achieves a 2.4% improvement in accuracy (ACC) on diagnostic tasks, a 16.2% increase in ACC on detection tasks, and a 10.6% reduction in mean squared error (MSE) on measurement tasks. This indicates that the model operating in deep thinking mode possesses enhanced capability for understanding ultrasound images and associated text. The supplementary material in I.1 demonstrates the model's capability for deep reasoning regarding ultrasound images.

## 5.3 SFT-RL PERFORMANCE BOUNDS

We find that the capability of the initial model constrains the upper bound of performance achievable by reinforcement learning. Our experiments reveal a critical limitation: UARPO cannot overcome the performance boundaries established during the SFT phase. Using two initial models with different U2-Scores (0.54 and 0.46), we observe that reinforcement learning improvements are strictly bounded by the foundation model's capability. The higher-performing initial model achieves a maximum accuracy of 0.81 after UARPO, while the lower-performing model is locked at a ceiling of 0.70. This fundamental constraint suggests that the SFT phase establishes an inescapable upper bound for subsequent reinforcement learning optimization. Rather than creating new capabilities, UARPO

merely refines and activates existing knowledge within the predetermined limits of the foundation model. The comparative table is presented in Appendix A.4.

# 6 RELATED WORK

Recent vision-language models have been adapted to medicine. MiniGPT-Med (Wu et al., 2023) provides a general interface for radiology diagnosis across X-ray, CT, and MRI, showing notable gains in medical report generation. RadFM (Chao et al., 2024) supports both 2D and 3D imaging, trained on 13M images and 615K scans, and outperforms prior multimodal models on radiology tasks. MedDr (Zhang et al., 2024) broadens coverage to radiology, pathology, dermatology, and more, using diagnosis-guided bootstrapping to achieve strong results. Med-Gemini (Saab et al., 2024) spans multiple modalities including ultrasound, but its capability remains limited to caption generation. Several models target ultrasound understanding. EchoCLIP (Christensen et al., 2024) aligns cardiac ultrasound videos with expert reports, achieving strong performance on ejection fraction prediction and device identification. LLaVA-Ultra (Guo et al., 2024) adapts general MLLMs to ultrasound via domain-specific fine-tuning. EchoPrime (Vukadinovic et al., 2024) advances multi-view cardiac analysis, training on over 12M video–report pairs. However, these models remain limited to specific anatomies or modalities and lack the reasoning capabilities needed for comprehensive ultrasound understanding. Our work addresses these gaps by developing a general-purpose foundation model with enhanced reasoning.

# 7 CONCLUSION

We introduce Dolphin, a multimodal large language model designed for comprehensive ultrasound understanding. Key contributions include: (1) a large-scale multimodal dataset standardized via the Ultrasound Data Protocol, (2) a three-stage progressive training framework balancing domain specialization and generalization, and (3) a mixed reasoning mechanism enabling both efficient and deep clinical reasoning. Dolphin achieves best performance on U2-Bench. Additionally, we highlight that the initial model quality sets a performance ceiling for subsequent reinforcement learning, underscoring the critical role of foundation model quality in medical AI.

## REPRODUCIBILITY STATEMENT

We affirm that all results presented in this work are reproducible. The code, experimental protocols necessary to reproduce the findings are either included in the supplementary materials or publicly available through the following repositories: Dolphin Ultrasound. All experiments were conducted using well-documented procedures, and detailed hyperparameters, random seeds, and preprocessing steps are provided in the Methods section and/or accompanying code documentation. We have taken care to ensure transparency and replicability so that other researchers can validate and build upon our work.

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

# A MATHEMATICAL ANALYSIS OF MIXED REASONING DYNAMICS

This appendix provides detailed mathematical derivations and theoretical analysis supporting the main findings on mixed reasoning dynamics presented in Section 4.

## A.1 MAXIMUM LIKELIHOOD ANALYSIS OF REASONING DATA EFFECTS

Our experimental observations reveal a non-monotonic relationship between reasoning data proportion $\gamma$ and diagnostic accuracy. We provide a rigorous analysis based on maximum likelihood training dynamics and data sufficiency conditions.

### A.1.1 TRAINING OBJECTIVES AND GROUND TRUTH ALIGNMENT

The fundamental goal is to maximize the posterior probability of correct reasoning chains for each mode:

$$\text{Reasoning mode:} \quad \max_\theta \ P_\theta(c_{\text{med}} \mid x, y, m = r) \tag{6}$$

$$\text{Vanilla mode:} \quad \max_\theta \ P_\theta(c_{\text{gt}} \mid x, y, m = q) \tag{7}$$

where $c_{\text{med}}$ represents reasoning chains that lead to correct medical diagnosis, and $c_{\text{gt}}$ represents the ground truth reasoning implicit in the vanilla training data.

Under maximum likelihood training, the model optimizes:

$$\mathcal{L}(\theta) = \mathbb{E}_{(x,y,m)\sim\mathcal{D}_{\text{mix}}}\big[\log P_\theta(y \mid x, m)\big], \tag{8}$$

where $P_\theta(y \mid x, m) = \sum_c P_\theta(y \mid x, c) P_\theta(c \mid x, m)$.

### A.1.2 DATA SUFFICIENCY AND PRIOR ESTIMATION QUALITY

The critical insight is that $P_\theta(c \mid x, m)$ estimation quality depends on the amount of mode-specific training data. Under mixture training $\mathcal{D}_{\text{mix}} = (1 - \gamma)\mathcal{D}_q + \gamma\mathcal{D}_r$, the effective sample sizes are:

$$N_q = (1 - \gamma) \cdot |\mathcal{D}_{\text{mix}}| \quad \text{(vanilla data)} \tag{9}$$

$$N_r = \gamma \cdot |\mathcal{D}_{\text{mix}}| \quad \text{(reasoning data)} \tag{10}$$

From statistical learning theory, the estimation error of mode-specific priors scales as:

$$\text{Estimation Error} \propto \frac{1}{\sqrt{N_{\text{mode}}}}. \tag{11}$$

**Small $\gamma$ regime ($N_r$ insufficient):** When reasoning data is scarce, the maximum likelihood estimator for $P_\theta(c \mid x, m = r)$ has high variance and poor discrimination. The learned prior approaches:

$$\hat{P}_\theta(c \mid x, m = r) \to \text{Uniform}(\mathcal{C}), \tag{12}$$

due to insufficient data to distinguish between different reasoning strategies.

This leads to *random reasoning selection*: the model samples reasoning chains without preference, including those $c_{\text{wrong}}$ that, despite being trained on medical likelihoods $P_\theta(y \mid x, c)$, are misaligned with medical reasoning requirements. The result is:

$$P_\theta(c_{\text{wrong}} \mid x, y, m = r) \approx P_\theta(c_{\text{med}} \mid x, y, m = r), \tag{13}$$

causing performance degradation relative to the well-trained vanilla mode.

**Large $\gamma$ regime ($N_r$ sufficient):** When reasoning data is abundant, the maximum likelihood estimator converges to the true underlying distribution. Crucially, cross-domain reasoning data contains examples where reasoning chains $c_{\text{good}}$ consistently lead to correct outcomes across diverse domains.

Through maximum likelihood training, the model learns to concentrate prior mass on reasoning patterns that demonstrate cross-domain effectiveness:

$$\hat{P}_\theta(c_{\text{good}} \mid x, m = r) \gg \hat{P}_\theta(c_{\text{wrong}} \mid x, m = r), \tag{14}$$

where $c_{\text{good}} \subset c_{\text{med}}$ represents reasoning chains that generalize well across domains.

### A.1.3 CROSS-DOMAIN REASONING TRANSFER MECHANISM

The key theoretical insight is that effective reasoning exhibits domain-invariant characteristics. Reasoning chains that consistently produce correct outcomes in mathematics, science, and logic also tend to be beneficial for medical diagnosis, despite domain differences.

During training on reasoning data $\mathcal{D}_r$, the model observes pairs $(x_r, y_r, m = r)$ where certain reasoning patterns $c_{\text{effective}}$ repeatedly correlate with correct outcomes. The maximum likelihood principle concentrates $P_\theta(c \mid x, m = r)$ on these patterns:

$$P_\theta(c_{\text{effective}} \mid x, m = r) \propto \exp\left(\frac{1}{N_r} \sum_{i=1}^{N_r} \log P_\theta(y_r^{(i)} \mid x_r^{(i)}, c_{\text{effective}})\right). \tag{15}$$

When applied to medical inputs, these domain-general reasoning patterns $c_{\text{effective}}$ align with medically appropriate reasoning $c_{\text{med}}$, enabling:

$$P_\theta(c_{\text{med}} \mid x, y, m = r) > P_\theta(c_{\text{gt}} \mid x, y, m = q), \tag{16}$$

explaining the superior performance of reasoning mode once sufficient training data is available.

### A.1.4 THEORETICAL PREDICTION OF OPTIMAL $\gamma$

The optimal reasoning data proportion $\gamma^*$ satisfies the data sufficiency condition:

$$\gamma^* \cdot |\mathcal{D}_{\text{mix}}| \geq \text{Critical Sample Size for Prior Discrimination}. \tag{17}$$

Below this threshold, poor prior estimation causes negative interference. Above this threshold, effective cross-domain reasoning patterns are successfully learned and transferred to medical diagnosis, resulting in performance gains that justify the observed U-shaped accuracy curve.

## A.2 STATISTICAL LEARNING THEORY FOUNDATIONS

This section provides additional theoretical foundations from statistical learning theory that support our analysis.

### A.2.1 BIAS-VARIANCE DECOMPOSITION

The estimation quality of mode-specific priors can be analyzed through bias-variance decomposition. For the reasoning mode prior $P_\theta(c \mid x, m = r)$, the expected squared error decomposes as:

$$\mathbb{E}\left[\|\hat{P}_\theta(c \mid x, m = r) - P^*(c \mid x, m = r)\|^2\right] = \text{Bias}^2 + \text{Variance} + \text{Noise}, \tag{18}$$

where $P^*(c \mid x, m = r)$ is the true underlying distribution.

When $\gamma$ is small, the variance term dominates due to insufficient training samples, leading to poor estimation quality. As $\gamma$ increases, both bias and variance decrease, enabling better prior estimation and improved reasoning performance.

### A.2.2 SAMPLE COMPLEXITY BOUNDS

Using VC theory and Rademacher complexity, we can derive sample complexity bounds for learning effective reasoning priors. For a hypothesis class $\mathcal{H}$ of reasoning chain distributions, the sample complexity for achieving $\epsilon$-accurate estimation with probability $1 - \delta$ is:

$$N_r \geq O\left(\frac{d \log(1/\epsilon) + \log(1/\delta)}{\epsilon^2}\right), \tag{19}$$

where $d$ is the VC dimension of $\mathcal{H}$.

This bound explains why a critical sample size $N_r^*$ exists below which reasoning performance degrades, and above which positive transfer emerges.

## A.3 MIXED INFERENCE SYSTEM PROMPT WORDS

- **Vanilla QA**: `You are a helpful assistant.`
- **Deep-Reasoning**: `You are a helpful assistant. To answer the user's question, you first think about the reasoning process and then provide the user with the answer. The reasoning process is enclosed within <think> </think> tags, respectively, i.e., <think> reasoning process here </think>.`

## A.4 INITIAL U2-SCORE DETERMINES THE ACHIEVABLE CEILING OF UARPO PERFORMANCE.

| Init U2-Score | Best Acc after UARPO | U2-Score after UARPO |
|---|---|---|
| 0.53 | 0.81 | 0.58 |
| 0.46 | 0.70 | 0.47 |

## A.5 DD ACCURACY COMPARISON ACROSS DIFFERENT DATA MIXING RATIOS

| Version | Reasoning Data (%) | DD Accuracy | Improvement (%) |
|---|---|---|---|
| Dataset 3 | 7.88 | 0.6408 | -4.3% |
| Dataset 2 | 11.72 | 0.6352 | -5.2% |
| Dataset 1 | 14.32 | 0.5652 | -15.6% |
| Dataset 5 | 19.56 | 0.6530 | -2.5% |
| Dataset 4 | 21.49 | **0.6698** | - |

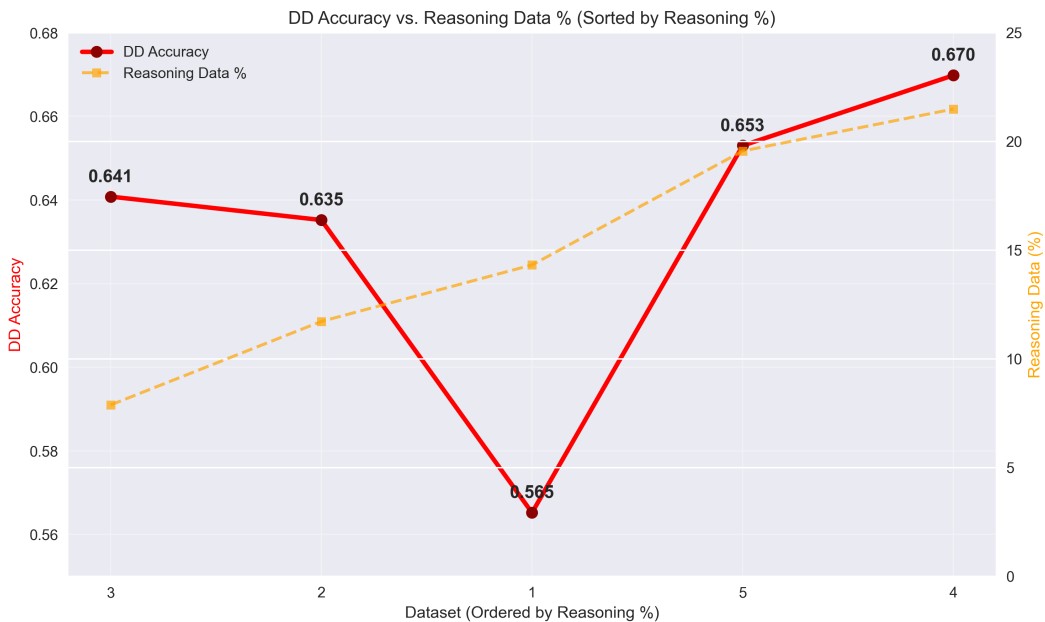

## B  CLINICAL EXPERT EVALUATION STUDY

### B.1  CLINICAL RATIONALITY ASSESSMENT

We randomly selected 30 ultrasound images and had Dolphin and GPT5 output diagnostic results respectively. Two ultrasound physicians with over five years of experience then conducted a clinical rationality assessment, with scores ranging from 1 to 5 points (5 being the highest). The following results were obtained.

Table 2: Clinical Relevance Assessment Scores

| Model | Mean Score | Variance | Minimum Score |
|---|---|---|---|
| Dolphin | 4.00 | 0.276 | 3 |
| GPT5 | 3.73 | 0.890 | 1 |

It can be observed that Dolphin significantly outperforms GPT-5 in both overall clinical relevance and variance. Additionally, Dolphin's minimum score is 3, while GPT-5's minimum score is 1, demonstrating that Dolphin's diagnostic performance on ultrasound images is more reliable.

### B.2  DOLPHIN VS HUMAN EXPERTS

We extracted diagnostic tasks from u2-bench to construct a test subset and developed a web-based application to evaluate both human performance and Dolphin's capabilities. Eight experienced ultrasound physicians, each with over five years of clinical practice, participated in the evaluation to establish a human baseline. The results are summarized in Table 3. Dolphin achieved a DD accuracy of 60.2% and an F1 score of 46.3%, outperforming the human baseline in DD accuracy (52.08%) and significantly surpassing it in DD F1 (29.95%). In the VRA task, Dolphin achieved an accuracy of 73.5% and an F1 score of 77.7%, compared to the human baseline of 60.63% accuracy and 49.91% F1. These results demonstrate that Dolphin surpasses experienced human physicians in interpreting ultrasound images on this benchmark, highlighting its strong diagnostic potential.

Table 3: Performance comparison between Dolphin and human experts on the u2-bench diagnostic tasks.

| Task | Model | Accuracy (%) | F1 Score (%) |
|---|---|---|---|
| VRA | Dolphin | 73.5 | 77.7 |
|  | Human ($n = 8$) | 60.63 | 49.91 |
| DD | Dolphin | 60.2 | 46.3 |
|  | Human ($n = 8$) | 52.08 | 29.95 |

### B.3  ROBUSTNESS ANALYSIS ACROSS RANDOM SEEDS

To verify the robustness and generalizability of our proposed method, we conduct comprehensive experiments across four different random seeds. Table 4 presents the mean and standard deviation of performance metrics across these random initializations.

Table 4: Performance comparison across four random seeds. Results are reported as mean and standard deviation (in separate rows).

| Model | DD | | VRA | | LL | OD | KD | CVE | | | RG | | | CG | | | U2-Score |
|---|---|---|---|---|---|---|---|---|---|---|---|---|---|---|---|---|---|
| | Acc. | F1 | Acc. | F1 | Acc. | Acc. | Acc. | RMSE | MAE | %tol | BLEU% | Rouge% | BERT% | BLEU% | Rouge% | BERT% | |
| Dolphin(Mean) | 0.65 | 0.56 | 0.68 | 0.56 | 0.44 | 0.55 | 0.41 | 0.24 | 0.22 | 41.86 | 2.01 | 13.30 | 72.34 | 42.51 | 60.23 | 90.05 | 0.54 |
| Dolphin(Std) | 0.00 | 0.02 | 0.01 | 0.01 | 0.03 | 0.01 | 0.03 | 0.02 | 0.02 | 3.84 | 1.23 | 2.08 | 1.23 | 5.03 | 3.29 | 0.91 | 0.01 |
| Dolphin Reasoning(Mean) | 0.67 | 0.57 | 0.70 | 0.60 | 0.47 | 0.59 | 0.52 | 0.24 | 0.23 | 44.68 | 2.38 | 14.32 | 72.91 | 46.68 | 64.29 | 90.89 | 0.58 |
| Dolphin Reasoning(Std) | 0.01 | 0.03 | 0.01 | 0.01 | 0.01 | 0.01 | 0.01 | 0.01 | 0.00 | 5.01 | 0.58 | 0.54 | 0.15 | 5.41 | 7.99 | 1.43 | 0.01 |

The results demonstrate that our method achieves consistent improvements across different random seeds, indicating the robustness and generalizability of the proposed approach. Specifically,

DolphonUltrasound72BV1.6-R1 consistently outperforms the baseline DolphinUltrasoundV16 across the majority of evaluation metrics, with relatively small standard deviations. This suggests that the observed performance gains are not artifacts of a particular random initialization, but rather reflect genuine improvements introduced by our method.

The low variance in most metrics (particularly in DD, VRA, LL, OD, KD, and U2-Score) further confirms the stability of our approach. Even for metrics with slightly higher variance (such as CVE and CG tasks), our method maintains superior average performance compared to the baseline, achieving an average U2-Score improvement from 0.55 to 0.58. This consistency across multiple random seeds provides strong evidence for the practical applicability and reliability of our proposed method in real-world scenarios.

## C  DOLPHIN ULTRASOUND DATA PROCESSING PROTOCOL (DUDP)

To handle various complex types of ultrasound data, we have specified a unified ultrasound data processing protocol DUDP. The protocol generates structured JSON metadata files that provide comprehensive information about each dataset and its constituent data samples.

### C.1  JSON SCHEMA STRUCTURE

The UDP protocol generates JSON files with the following hierarchical structure:

#### C.1.1  DATASET-LEVEL METADATA

The root level contains dataset-wide information:

- `DatasetName`: Dataset identifier
- `CreateTime`: Dataset creation timestamp
- `DataType`: Data format (`img`, `video`, or `mixture`)
- `DataNum`: Total number of samples
- `AnatomyLocation`: List of anatomical regions covered
- `ClassesDict`: Disease classification categories
- Boolean flags indicating data availability (`IncludeSeg`, `IncludeClasses`, etc.)

#### C.1.2  SAMPLE-LEVEL DATA STRUCTURE

Each data sample is stored under `DataInfo` with the following fields:

- `data_path`: Relative path to image/video file
- `seg_path`: Path to segmentation mask (if available)
- `classes`: Disease classification labels
- `anatomy_location`: Anatomical region identifier
- `box`: Bounding boxes in normalized coordinates $[x_{center}, y_{center}, width, height]$
- `keypoints`: Anatomical landmarks in relative coordinates $[x, y] \in [0, 1]$
- `measurement`: Ultrasound-specific measurements (e.g., ejection fraction)
- `demographic`: Patient demographics (age, gender, BMI)
- `biochemical`: Clinical laboratory results
- `patient_id`: Unique patient identifier
- `split`: Data partition assignment (`train`/`val`/`test`)

## C.2 COMPLETE JSON EXAMPLE

A complete UDP-generated JSON structure with all possible fields:

```
{
    "DatasetName": "ultrasound-dataset",
    "DatasetDescription": "Multi-organ ultrasound dataset",
    "CreateTime": "2025-04-15 10:30:00",
    "CreateUser": "researcher",
    "DataNum": 1000,
    "DataType": "img",
    "IncludeSeg": true,
    "IncludeClasses": true,
    "IncludeCaption": true,
    "IncludeReport": true,
    "IncludeDemographic": true,
    "IncludeBiochemical": true,
    "IncludeMeasurement": true,
    "IncludeKeypoints": true,
    "IncludeBox": true,
    "IncludeSplit": true,
    "IncludeNotes": true,
    "IncludePatientID": true,
    "SegChannel": 2,
    "AnatomyLocation": ["breast", "liver", "kidney"],
    "ClassesDict": {
        "malignancy": ["benign", "malignant"],
        "organ_condition": ["normal", "abnormal"]
    },
    "MeasuresList": ["EF", "ESV", "EDV", "diameter"],
    "KeypointsList": ["landmark1", "landmark2", "center"],
    "BoxList": ["tumor", "cyst", "lesion"],
    "Notes": "Research dataset for AI development",
    "PatientIDFormat": {
        "description": "Numeric ID",
        "min_value": 1,
        "max_value": 1000,
        "pattern": "Integer",
        "example": "123"
    },
    "DataInfo": {
        "case000001": {
            "anatomy_location": "breast",
            "split": "train",
            "data_path": "dataset/img/case000001.png",
            "original_path": null,
            "seg_path": "dataset/seg/case000001.npy",
            "seg_channel_name": ["tumor", "background"],
            "classes": {"malignancy": "malignant", "organ_condition": "abnormal"},
            "caption": "Breast ultrasound showing suspicious mass",
            "report": "Hypoechoic mass measuring 2.1cm with irregular borders",
            "box": [{"tumor": [0.45, 0.32, 0.12, 0.08]}],
            "measurement": {"diameter": 2.1, "depth": 1.5},
            "keypoints": {"center": [0.45, 0.32], "landmark1": [0.40, 0.28]},
            "demographic": {"Gender": "female", "Age": 45, "BMI": 24.5},
            "biochemical": {"CA15-3": 25.2, "CEA": 3.1},
            "patient_id": "123",
            "notes": "Follow-up recommended"
        },
        "case000002": {
            "anatomy_location": "liver",
            "split": "val",
            "data_path": "dataset/img/case000002.png",
            "original_path": "/original/data/source.png",
            "seg_path": null,
```

```
            "seg_channel_name": null,
            "classes": {"organ_condition": "normal"},
            "caption": null,
            "report": null,
            "box": null,
            "measurement": null,
            "keypoints": null,
            "demographic": {"Gender": "male", "Age": 35},
            "biochemical": null,
            "patient_id": "124",
            "notes": null
        }
    }
}
```

## D  GENERAL & MEDICAL BENCHMARK

Table 5: General Benchmark Performance

| Models | MMBench | CCBench | SEEDBench | TextVQA | MME | VQA | MedFramQA |
|---|---|---|---|---|---|---|---|
| GPT5 | 84.36 | 75.49 | 77.63 | 58.74 | 80.08 | 44.12 | 47.42 |
| Gemini1.5Pro | 73.20 | 28.40 | 76.00 | 61.60 | 78.12 | 38.73 | 43.60 |
| QwenVLMax | 88.06 | 73.33 | 59.21 | 59.24 | 87.19 | 47.98 | 46.58 |
| Qwen2.5-VL-72B | 87.40 | 73.90 | 79.50 | 53.30 | 84.97 | 34.38 | 43.99 |
| LLaVa-V1.5-7B | 22.94 | 8.63 | 60.01 | 21.73 | 67.41 | 37.92 | 43.77 |
| HunYuan-T1-Vision | 84.02 | 70.00 | 73.20 | 41.85 | 84.54 | 41.02 | 42.24 |
| llava-med-v1.5-mistral-7b | 35.07 | 29.85 | 32.57 | 42.4 | 50.97 | 30.16 | 30.13 |
| Dolphin | 85.31 | 67.45 | 77.00 | 61.73 | 87.54 | 47.89 | 52.73 |

Table 6: Detail performance on MedFramQA

| Models | CT | MRI | Ultrasound | X-ray | Other |
|---|---|---|---|---|---|
| GPT5 | 48.21 | 47.93 | 42.60 | 49.30 | 47.37 |
| Gemini1.5Pro | 43.87 | 41.26 | 45.51 | 44.81 | 49.54 |
| QwenVLMax | 48.04 | 45.05 | 46.77 | 46.18 | 46.21 |
| Qwen2.5-VL-72B | 45.15 | 44.42 | 43.67 | 38.21 | 45.45 |
| LLaVa-V1.5-7B | 40.49 | 47.03 | 47.29 | 36.88 | 52.27 |
| HunYuan-T1-Vision | 41.07 | 43.10 | 42.63 | 43.98 | 40.37 |
| llava-med-v1.5-mistral-7b | 30.26 | 29.31 | 30.90 | 35.11 | 29.81 |
| Dolphin | 52.77 | 50.06 | 60.31 | 49.17 | 56.15 |

Table 7: Model Performance on MedQA Benchmarks

| Model | MedQA_USMLE | MedQA_MCMLE |
|---|---|---|
| QwenVLMax | 74.86 | 85.44 |
| GPT5 | 76.67 | 83.07 |
| LLaVa_v1.5_7B | 43.91 | 29.98 |
| Qwen-2.5-VL-72B | 84.21 | 63.71 |
| llava-med-v1.5-mistral-7b | 41.01 | 25.66 |
| Dolphin | 72.51 | 73.61 |

## E  EXPERIMENTAL SETUP

All experiments were conducted on NVIDIA H100 clusters. For the 72B model, both supervised fine-tuning (SFT) and UARPO reinforcement learning (RL) were performed on 96 H100 GPUs; for the 7B model, both SFT and UARPO RL were performed on 32 H100 GPUs. We adopted DeepSpeed ZeRO-3 for memory-efficient distributed training, FP16 mixed-precision arithmetic for improved throughput and reduced memory footprint, and AdamW as the optimizer. The per-GPU batch size was fixed at 1 in all runs, yielding an effective global batch size equal to the number of GPUs (i.e., 96 for the 72B setting and 32 for the 7B setting), unless otherwise specified. The learning rate was set to $2 \times 10^{-6}$ for both SFT and GRPO RL. During multimodal training, each sample could include up to seven images at $512 \times 512$ resolution.

## F  THE USE OF LARGE LANGUAGE MODELS (LLMs)

In the preparation of this work, we utilized artificial intelligence (AI) tools to assist with data organization, code formatting, and textual refinement. Specifically, AI was employed to help clean and structure datasets, improve code readability, and provide language suggestions for enhancing clarity and coherence in the manuscript. All technical content, conceptual development, and final decisions regarding methodology and presentation were made by the authors. The use of AI was strictly limited to auxiliary tasks, and the intellectual contributions remain entirely those of the human authors.

## G  EXAMPLES OF CONSTRUCTED ULTRASOUND QUESTION–ANSWER PAIRS

Examples of constructed ultrasound question–answer pairs. The dataset integrates diverse clinical tasks, including fetal abdominal view assessment, lung ultrasound severity scoring, and soft tissue lesion evaluation, with corresponding expert-verified answers. These QA pairs provide structured supervision signals that enhance both low-level image understanding and high-level diagnostic reasoning in Dolphin models.

# H   DATASET DETAILS AND LICENSE

Table 8: Summary of Annotated Datasets Used in Public Data

| Dataset | Anatomy | Clinical scenarios | Task | Case | License |
|---|---|---|---|---|---|
| FETAL PLANES DB (Burgos-Artizzu et al., 2020) | Fetal abdomen Fetal brain Fetal femur Fetal thorax Maternal cervix other | Fetal standard plane identification | VRA | 137 | CCA 4.0I |
| DDTI (Pedraza et al., 2015) | thyroid | Thyroid nodule identification Thyroid nodule localisation | VRA LL | 110 | - |
| The Open Kidney US Dataset (Singla et al., 2023) | kidney | Kidney detection Kidney Diag view identification | VRA OD | 110 | CC BY-NC-SA |
| FPUS23 (Prabakaran et al., 2023) | Fetal abdomen Fetal arm Fetal head Fetal legs | Fetal diagnostic planes identification Fetal US report generation | VRA RP | 752 | MIT |
| Echogenic (Da Correggio et al., 2023) | Fetal abdomen | Fetal abdominal organ detection | OD | 102 | CCA 4.0 |
| FALLMUD (FALL-MUD) | Crural muscles | Muscle detection | OD | 100 | - |
| Micro-US Prostate Segmentation Dataset (Shao &Brisbane, 2024) | Prostate | Prostate localisation Prostate Diag view identification | VRA LL | 110 | CCA 4.0I |
| CAMUS (Leclerc et al., 2019) | Heart ED Heart ES Heart 2CH Heart 4CH | Heart ejection fraction estimation Heart atrium and ventricle localisation | VRA OD CVE | 316 | CC BY-NC-SA 4.0 |
| Breast Lesion Detection in US Videos (Lin et al., 2022) | Breast benign Brest malignant | Breast lesion classification | Diag | 171 | - |
| Breast US Images Dataset (Al-Dhabyani et al., 2020) | Breast | Breast cancer level classification Breast tumour localisation Brest Diag view identification | Diag VRA LL | 210 | CC0: PD |
| Dermatologic Ultrasound Images for classification (Laverde Saad et al., 2021) | Skin | Skin tumor level classification | Diag | 100 | - |

(Continued) Table 8

| Dataset | Anatomy | Clinical scenarios | Task | Case | License |
|---------|---------|-------------------|------|------|---------|
| Polycystic Ovary Ultrasound Images Dataset (Wisesty et al., 2018) | Ovary | Polycystic Ovary Syndrome localisation | VRA | 10 | CC0: PDD |
| CUBS (Meiburger et al., 2021) | Carotid | Carotid thickness estimation
Carotid detection
Catotid Diag view identification | VRA
OD
CVE | 681 | CCA 4.0I |
| Knee US dataset in a population-based cohort (Novin et al., 2023) | Knee | Knee US KL and pain grad classification
Knee Diag view identification
Knee lesion localisation | Diag
VRA
OD | 326 | CC0 1.0 |
| HC18 (van den Heuvel et al., 2018) | Fetal head | Fetal head circumference estimation
Fetal head detection | OD
CVE | 202 | CCA 4.0I |
| KFGNet (NeuronXJTU &palkia1998, 2023) | Thyroid | Thyroid nodule level classification
Thyroid nodule localisation | Diag
LL | 206 | - |
| Thyroid (Krönke et al., 2022) | Thyroid Left Thyroid right | Thyroid Diag view identification | VRA | 563 | CC BY |
| GDPHSYSUCC (Mo et al., 2022) | Breast | Breast lesion classification | Diag | 109 | - |
| LEPset (Li et al., 2023b) | Pancreas | Pancreatic cancer classification | Diag | 101 | CCA 4.0I |
| COVID-BLUES (Wiedemann et al., 2025) | Lung | COVID-19 level classification
Lung US caption generation
Lung Diag view identification | Diag
VRA
CG | 318 | ANN 4.0 I |
| Ultrasound Guided Regional Anesthesia (Tyagi et al., 2024) | Brachial plexus | Brachial plexus detection | OD | 179 | Non-commerical |
| Unity Imaging Collaborative (Shun-Shin, 2023) | Cardiac | Caridac Keypoint Detection | KD | 500 | CCANN 4.0 I |
| C-TRUS Dataset (Leenings et al., 2025) | Colon | Colon wall detection | OD | 166 | - |
| ACOUSLIC-AI (Sappia, 2024) | Fetal abdominal | Fetal abdominal circumference estimation
Fetal adominal OD | VRA
OD
CVE | 310 | CCANCSA 4.0I |

(Continued) Table 8

| Dataset | Anatomy | Clinical scenarios | Task | Case | License |
|---|---|---|---|---|---|
| PSFHS (Bai, 2024) | Fetal head
Fetal pubic symphysis | Fetal head detection
Fetal pubic symphysis detection | OD | 100 | CCA 4.0I |
| JNU-IFM (Lu et al., 2022) | Fetal head
Fetal pubic symphysis | Fetal view identification
Fetal head detection
Fetal pubic symphysis detection | VRA
OD | 202 | CC BY 4.0 |
| Dataset of B-mode fatty liver US images (Byra et al., 2018) | Liver | Liver steatosis classification
Liver fat value estimation
Liver Diag view identification | Diag
VRA
CVE | 222 | CCA 4.0I |
| African Fetal Standard Plane (Sendra-Balcells et al., 2023) | Fetal abdomen
Fetal brain
Fetal femur
Fetal thorax | Fetal standard plane identification | VRA | 10 | CCA 4.0I |
| BrEaST (Pawłowska et al., 2024) | Breast | Breast LL | LL | 100 | CC BY 4.0 |
| Ultrasound Breast Images for Breast Cancer (Sairam, 2020) | Breast | Breast cancer classification | Diag | 100 | CC0: PD |
| US simulation and segmentation (Vitale et al., 2020) | Abdominal | Abdominal OD | OD | 100 | - |
| Carotid Artery Ultrasound and Color Doppler (Pahuni Choudhary, 2023) | External carotid
left carotid
right carotid | Carotid Diag view identification | VRA | 100 | Apache 2.0 |
| AUITD (Maroua, 2020) | Thyroid | Thyroid lesion classification | Diag | 100 | - |
| Auto-PCOS classification (Maroua, 2020) | Ovary | Polycystic Ovary Syndrome classification
Ploycystic Diag view identification | Diag
VRA | 218 | CCA 4.0I |
| Auto-PCOS classification (Maroua, 2020) | Ovary | Polycystic Ovary Syndrome classification | Diag | 100 | CC BY 4.0 |

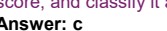 **Question:** You are analyzing an ultrasound image of a fetal abdomen; assess the visibility of key landmarks (stomach, umbilical vein, spine) to determine if the view is suitable for Abdominal Circumference (AC) measurement, and indicate the classification as bad, optimal, or suboptimal.

**Answer:** The sonographic features point to a bad condition for AC measurement.

**Question:** Clinical Lung Ultrasound Severity Score Diagnosis: - Score 0: Normal: Continuous pleural line, A-lines, sliding sign - Score 1: Mild: Scattered B-lines (≥3, non-coalescent), slightly irregular pleural line - Score 2: Moderate: Coalescent B-lines, broken pleural line, small consolidations - Score 3: Severe: Dense 'white lung' B-lines, large consolidations 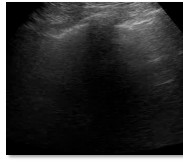
You are a clinician skilled in Lung Ultrasound (LUS) classifying a lung ultrasound image; account for probe angle variations by relying on consistent features like B-line presence to assign a severity score, and classify it as **a. 0, b. 1, c. 2, d. 3.**

**Answer: c**

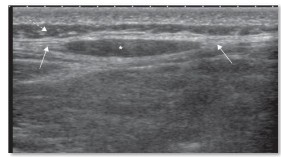 **Question:** When assessing the likelihood of malignancy in soft tissue masses, what key factors related to size and shape should be considered? How can these factors be combined with ultrasound images to analyze the nature of the current lesion?

**Answer:** In evaluating whether a soft tissue mass has the potential for malignancy, size and morphology are important reference factors. Typically, among soft tissue tumors with a diameter exceeding 5 centimeters, only 5% are benign, so larger masses require more vigilance for the possibility of malignancy. Additionally, malignant tumors (such as sarcomas) often exhibit eccentric growth, with shapes that are mostly round or oval, unclear boundaries, and stronger invasiveness. The current ultrasound image shows the lesion to have a lenticular morphology, a characteristic feature commonly seen in lipomas located within the inner layer of the fascia. Such lipomas can extend laterally to a larger size but have a shallow depth, consistent with the characteristics observed in the current image.
The ultrasound examination also shows that the mass has clear boundaries and uniform internal echoes, suggesting a higher likelihood of it being a benign lesion. Nevertheless, the final diagnosis still needs to be made in conjunction with clinical manifestations and other auxiliary examinations (such as imaging enhancement or biopsy if necessary). In summary, based on the current lesion's size, morphology, and location, it is more likely to be a benign lipoma, but further clinical verification is needed to rule out other possibilities.

# I   DOLPHIN OUTPUT EXAMPLE

## I.1   DEEP THINKING EXAMPLE

---

**Deep Reasoning \ thyroid**

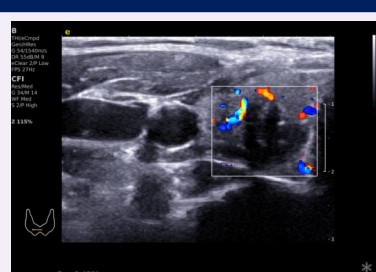

**Question:** Observe this thyroid image and perform an analysis diagnosis

**Answer:** <think>First, I need to carefully examine the various features in the ultrasound image, such as the shape of the nodule, echo characteristics, whether the margins are clear, the presence of calcifications, and the blood flow situation shown by color Doppler.
Location and Shape of the Nodule: The nodule is located in the right lobe of the thyroid gland and has an irregular shape. This may suggest a higher risk of malignancy, as benign nodules are usually more regular in shape.
Echo Characteristics: The nodule has heterogeneous internal echoes with areas of low echo, while the surrounding tissue shows higher echoes. This may indicate that the density of the nodule is different from that of the surrounding tissue and requires further analysis.
Margins: The margins of the nodule are indistinct and appear to have an infiltrative quality. This is not typically a feature of benign nodules, which usually have clear margins and a complete capsule.
Calcifications: There are tiny, hyperechoic foci within the nodule, which may represent microcalcifications. Microcalcifications are relatively common in thyroid cancer, especially papillary carcinoma, while coarse calcifications are more likely to be associated with benign conditions. However, the presence of microcalcifications does increase the likelihood of malignancy.
Color Doppler Flow Imaging (CDFI): The nodule shows abundant blood flow signals, appearing as punctate or linear blood flow, indicating a high metabolic state. Benign nodules typically have less blood flow, while malignant nodules, due to their rapid growth, require more blood supply, hence the abundant blood flow. This also supports the possibility of malignancy.
Comprehensive Analysis: The nodule's irregular shape, low echo, indistinct margins, microcalcifications, and abundant blood flow signals are all typical features of thyroid cancer. According to the TI-RADS classification system, these features may correspond to category 4C or 5, indicating a higher risk of malignancy. Therefore, fine-needle aspiration biopsy (FNA) is recommended to confirm the pathological diagnosis.
Other Considerations: Other possibilities, such as benign lesions like follicular adenomas, should also be considered, although they usually have more homogeneous echoes and less blood flow. Additionally, the quality of the image should be assessed for clarity, and other clinical symptoms, such as enlarged lymph nodes, should be taken into account. However, based on the current information, the likelihood of malignancy is higher.
Final Recommendations: When providing recommendations, it is essential to integrate clinical data with imaging features. Further examinations and pathological confirmation are advised to avoid misdiagnosis.
<think>

Based on the analysis of the provided image of the thyroid, the following diagnostic recommendations are made:

**1.Morphology and Margins**
- The nodule has an irregular shape with indistinct margins and lacks a complete capsule.
- This does not match the typical characteristics of benign nodules, which are usually round/oval and have clear margins.

**2.Echo Characteristics**
- The internal echoes are heterogeneous with visible hypoechoic areas.
- The surrounding tissue shows increased echoes, forming a distinct contrast.

**3.Calcification Features**
- Tiny hyperechoic foci (microcalcifications) are detected.
- These are consistent with the microcalcification features of thyroid cancer (diameter < 1 mm).

**4.Blood Flow Characteristics**
- CDFI shows abundant blood flow within the nodule.
- The blood flow signals are distributed in a punctate or linear pattern, which is consistent with the characteristics of a highly metabolic tumor.

**5.TI-RADS Classification**
- According to the ATA guidelines, this combination of features is consistent with category 4C (malignancy risk 15-50%).
- If scored according to the ACR TI-RADS, the total score is ≥ 7 points (malignancy risk 3%-5%).

**Suggested Diagnosis:**
Suspicious malignant thyroid nodule (TI-RADS 4C)

**Follow-up Recommendations:**
- **Fine-Needle Aspiration Biopsy (FNA):**It is recommended to perform the biopsy under ultrasound guidance.This should meet the criteria for biopsy of ACR TI-RADS category 4 or higher nodules.
- **Multimodal Imaging Assessment:**It is suggested to combine CT or MRI examinations, especially to observe for any cervical lymph node metastasis.For atypical cases, elastography may be considered as an auxiliary assessment.
- **Clinical Follow-up:**If immediate biopsy is not possible, arrange for a follow-up ultrasound in 3-6 months.Monitor changes in nodule size and evolution of its characteristics.

**Note:**
The final diagnosis should be combined with pathological results. The ultrasound findings suggest a higher probability of malignancy, and it is recommended to prioritize cytological examination to clarify the nature of the nodule.

## Deep Reasoning \ Cervix

**Question:** Observe this cervix ultrasound image and perform an analysis diagnosis

**Answer:** <think>Alright, I'm now going to analyze this transvaginal ultrasound image and provide a diagnostic opinion. First, I will carefully observe the various structures in the image, including the fetus, uterus, and placenta, and record the key measurement data.

The image shows an early pregnancy fetus with a crown-rump length (CRL) of approximately 0.93 cm, corresponding to a gestational age of 7 weeks and 0 days, and the estimated due date is December 26, 2024. The CRL measurement is within the normal range, indicating that the fetal development is essentially normal. Next, I need to check for other abnormalities, such as whether the fetal heart rate is normal, whether the amniotic fluid is adequate, and whether the placental position is appropriate. However, the fetal heart rate is not displayed in the current image, and only the contour of the fetal head can be seen.

The endometrial structure of the uterus appears relatively uniform, with no obvious thickening or irregularities, which may suggest that the endometrium is within the normal range. Additionally, the cervical part is clearly displayed, with no dilation or abnormal echoes, ruling out the possibility of cervical insufficiency. Transvaginal ultrasound usually provides a clear view of the internal structures of the uterus, so the position and shape of the uterus should be fine.

Given the gestational age of 7 weeks and 0 days, the fetus should be in the gestational sac formation period, during which the fetal bud and primitive cardiac tube pulsation may have already appeared. However, these details may not be shown in the image provided by the user, and it is necessary to confirm whether these parts were missed during the examination. In addition, it is important to pay attention to any abnormal echoes or structures, such as whether there are signs of bleeding around the gestational sac, or whether the placenta is in the correct position.

During the assessment, the following points may need to be considered:
1.Whether the fetal crown-rump length is consistent with the gestational week. The normal range is generally between 0.5 and 1.0 cm, and 0.93 cm is within this range.
2.Whether the thickness of the endometrium is normal. The thickness of the endometrium in early pregnancy is usually between 6 and 12 mm, but the specific value needs to be measured in detail in combination with the image.
3.Whether there are any abnormal effusions or cystic structures, such as uterine cavity effusion or ovarian cysts, but the structures shown in the image are relatively simple.
4.Whether the placental position is central, whether it is close to the internal os of the cervix, and whether there is a possibility of placenta previa. However, the placental position is not shown in the current image.

In addition, considering that this is a transvaginal ultrasound, the image resolution is high and can more clearly display the structures of the uterus and fetus, so it is usually more accurate in assessing early pregnancy. However, it is necessary to confirm whether there are measurement errors, such as whether the measurement of the crown-rump length is accurate, and whether there are other measurement parameters such as biparietal diameter or femur length. But these are not shown in the current image.

Finally, based on the above analysis, the image shows that the fetus is developing normally, the gestational sac structure is clear, the endometrial thickness is moderate, and no obvious abnormalities are seen. However, it is necessary to further check the fetal heart rate and fetal bud situation, and to confirm whether there are other potential problems. It is recommended that the clinical doctor combine other examination results and the symptoms of the pregnant woman for a comprehensive assessment, and if necessary, follow-up ultrasound examinations should be carried out to monitor the fetal development.<\think>

Based on the analysis of the provided ultrasound image of the cervix, the following diagnostic recommendations are made:

**1. Fetal Development**
- **Crown-Rump Length (CRL):** Measured at 0.93 cm, corresponding to a gestational age of 7 weeks and 0 days, which is within the normal range for early pregnancy (0.5-1.0 cm), indicating normal fetal development.
- **Estimated Due Date (EDD):** Calculated as December 26, 2024, which is consistent with the gestational age (typically, a CRL of about 0.9-1.0 cm at 7 weeks).

**2. Uterine Structure**
- **Endometrium:** Moderate thickness (specific measurement value needed), no signs of thickening or abnormal echoes, suggesting that the endometrium is supportive of gestational sac development.
- **Cervix:** No dilation or abnormal echoes observed, ruling out cervical insufficiency.

**3. Other Observations**
- No obvious abnormal echoes in the fetus (such as cysts, calcifications), placental position not displayed (further assessment of placental position and function needed in subsequent examinations).

**4.Recommendations**
- **Subsequent Monitoring:** It is recommended to further confirm fetal viability by assessing fetal heart rate (via Doppler) and fetal bud (cardiac tube pulsation).
- **Regular Follow-Up:** A repeat ultrasound in 7-10 days is suggested to evaluate fetal growth and amniotic fluid volume.
- **Clinical Correlation:** The patient's medical history, symptoms, and other tests (such as blood hCG and progesterone levels) should be taken into account for a comprehensive assessment.

**Conclusion**:
Normal Fetal Development: Accurate CRL measurement, consistent with gestational age, and clear gestational sac structure.
Normal Uterine Structure: No abnormalities in endometrial thickness or morphology, and good cervical condition.

**Note:**
This report is based on the available images. If there is an abnormal fetal heart rate or no fetal bud is observed, the possibility of early miscarriage or fetal arrest should be considered, and further imaging and laboratory tests are needed for confirmation.

## Deep Reasoning \ Knee

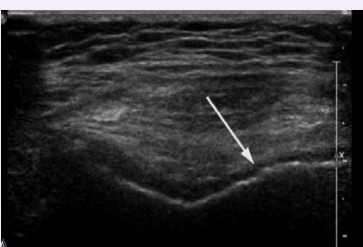

**Question:** Observe this ultrasound image of knee and perform an analysis and diagnosis

**Answer:** <think>Analysis of Knee Joint Ultrasound Images：
When analyzing knee joint ultrasound images, the first step is to focus on the morphology and echogenicity of the articular cartilage. Under normal circumstances, the surface of the knee joint cartilage should be smooth and uniform, with low echogenicity, appearing as hypoechoic or isoechoic. In this image, the area indicated by the arrow shows uneven echogenicity, which may suggest that the cartilage surface is no longer smooth and may indicate signs of damage or degeneration.
Further observation should be made to see if the low-echogenic band structure of the cartilage is intact. If the cartilage surface shows indentations, fissures, or irregular thickening, this usually indicates cartilage damage. Additionally, increased echogenicity of the subchondral bone may also be a sign of degenerative changes.
In some cases, increased echogenicity of the subchondral bone may be related to the bone response caused by cartilage lesions.
The uneven echogenicity of the cartilage surface in the image may be associated with cartilage edema, fibrosis, or other pathological changes. These changes can cause the cartilage to exhibit different echogenic properties on ultrasound images, thereby affecting the diagnostic results. Therefore, it is necessary to combine the patient's clinical symptoms and other imaging examinations (such as MRI) for a comprehensive assessment.
In terms of differential diagnosis, it is important to rule out other diseases that may cause similar ultrasound appearances, such as synovitis or meniscal injury. These conditions may also lead to abnormal echogenicity in the knee joint area, but their manifestations and locations are different. For example, synovitis usually presents as thickening of the synovial layer and increased blood flow signals, while meniscal injury may show abnormal echogenicity or morphological changes in the meniscal area.
Ultimately, based on the characteristics of the ultrasound images and clinical manifestations, it is preliminarily judged that there may be cartilage lesions in this knee joint. It is recommended to further undergo MRI examination to clarify the diagnosis and to formulate an appropriate treatment plan according to the specific situation, such as physical therapy, drug treatment, or surgical intervention.
<\think>

Based on the analysis of the provided ultrasound image of knee, the following diagnostic recommendations are made:

**Characteristic Changes:**
**1.Irregular Cartilage Surface:** The area indicated by the arrow shows uneven echogenicity of the cartilage surface, suggesting a loss of smoothness, which may be related to cartilage injury or degeneration.

**2.Abnormal Internal Cartilage Structure:** Normally, cartilage should appear hypoechoic or isoechoic. However, the image shows increased and unevenly distributed echogenicity within the cartilage, which may reflect cartilage edema, fibrosis, or tears.

**3.Variation in Cartilage Thickness:** If the cartilage is significantly thinner compared to the contralateral joint (clinical measurement required), it may indicate cartilage degeneration or wear.

 **Recommendations for Further Assessment:**
- **Clinical Symptoms:** Knee pain, swelling, or restricted movement.
- **Comparison with the Contralateral Joint:** Compare with the healthy contralateral joint to define the extent of the lesion.
- **Additional Imaging Studies:** MRI is recommended to assess full-thickness cartilage damage, bone marrow edema, or meniscal/ligamentous injuries.

**Conclusion:**
Based on the ultrasound findings, it is preliminarily considered that there is cartilage pathology in the knee joint (such as cartilage injury or degeneration). It is recommended that the patient undergoes MRI of the knee to determine the extent of cartilage damage and to develop a treatment plan based on the specific circumstances (such as conservative treatment or surgical intervention)

## Deep Reasoning \ Fetal

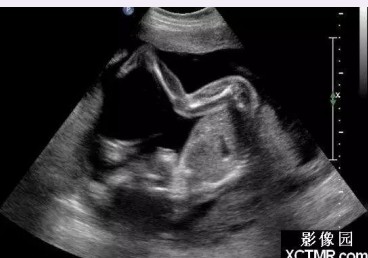

**Question:** Observe this ultrasound image of fetal and perform an analysis and diagnosis

**Answer:** <think>Alright, I'm going to analyze the fetal femur ultrasound images provided by the user and determine if there are any abnormalities. First, I need to recall the normal ultrasound appearance of the fetal femur. The fetal femur typically appears as a long, rod-like structure on ultrasound, with enlarged ends and a hyperechoic shaft surrounded by hypoechoic soft tissue. The key aspects to assess are the length, shape, position of the femur, and whether there are any other structural abnormalities.
Next, I will carefully examine the femur in the image. The femur should be located in the thigh area. Is the femur shown in the image complete, with no obvious fractures or bends? A normal femur should be straight. If it appears bent, this may indicate a fracture or other deformity. Additionally, whether the femur length is within the standard for the gestational age is a crucial indicator.
If the femur length is significantly shorter than the normal range for the same gestational age, it may suggest skeletal dysplasia, such as delayed bone development or genetic disorders.
Then, I will check the surrounding soft tissues of the femur for any abnormalities. For example, are there any masses, effusions, or other structural abnormalities? At the same time, I need to pay attention to other related structures, such as the position of the knee and hip joints, and whether there is any dislocation or deformity. For instance, if there is an abnormal bend near the proximal end of the femur, congenital hip dysplasia may need to be considered.
Also, I need to pay attention to other structures in the image. For example, if there is a mass in the abdominal area, it may indicate abnormalities such as abdominal wall defects or omphalocele. I should also check for any abnormalities in the head and other limbs, but based on the current image, the focus may primarily be on the femur.
The challenges I may encounter are related to image quality, such as insufficient resolution or artifacts that affect the judgment. In these cases, I need to combine clinical information and other test results for a comprehensive analysis. Moreover, the appearance of the femur will vary at different gestational ages, so it is essential to refer to the correct gestational age standards for comparison.
Finally, based on all the observed information, I will determine whether there are any length abnormalities, shape abnormalities, or other accompanying structural issues with the femur, and thus make a diagnosis of whether there are any abnormalities.
<\think>

Based on the analysis of the provided image of the fetal, the following diagnostic recommendations are made:

**1. Morphology and Integrity**
- The femur has a typical elongated shape, with enlarged ends (femoral head and femoral condyles), a hyperechoic shaft in the center, and symmetrical distribution of surrounding soft tissues. No obvious fractures, bends, or discontinuities are observed.

**2. Length Assessment**
- Comparison with gestational age standards (e.g., femur length of approximately 6-7 cm at 20 weeks of gestation) is necessary. If the femur length is consistent with the gestational age, there is no evident developmental abnormality.

**3. Surrounding Structures**
- No abnormal echoes in the soft tissues are observed (such as masses or effusions). The positions of the knee and hip joints are normal, with no signs of dislocation or deformity.

**4. Other Structural Findings**
- The abdominal structures are visible in the image and appear normal, with no accompanying abnormalities such as abdominal wall defects.

**Conclusion:**
The ultrasound image shows that the fetal femur's morphology, length, and surrounding structures are all within the normal range, with no evident developmental abnormalities or deformities. Regular prenatal check-ups and follow-up are recommended to dynamically monitor growth.

## I.2 EXAMPLE OF MULTI-TURN DIALOGUE

### Multiround Dialogue \ Thyroid

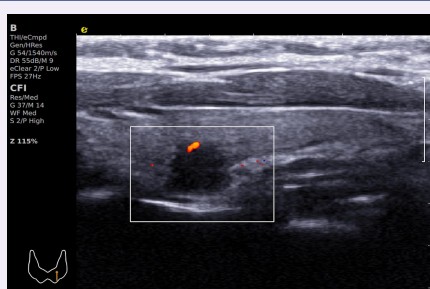

**Question 1**: Analyze the scanned area and the section of this ultrasound image.

Based on the analysis of the provided ultrasound image, the following diagnostic recommendations are made:

- The image displays the typical anatomical structure of a thyroid longitudinal section. The image shows a medium-echo area of the thyroid parenchyma, which is surrounded by the low-echo sternocleidomastoid muscle. Color Doppler ultrasound reveals abundant blood flow signals within the thyroid, which is consistent with the conventional appearance of a thyroid longitudinal scan.

**Question 2:** What structures can be observed in this thyroid ultrasound image, and are there any abnormalities in each structure?

Based on the analysis of the provided ultrasound image of thyroid, the following diagnostic recommendations are made:

**1.Thyroid Parenchyma:**
- **Abnormal Appearance:** A hypoechoic nodule is visible in the middle and upper part, with unclear boundaries and inhomogeneous internal echoes, indicating increased local tissue density or structural disorder. Normal thyroid parenchyma shows uniform medium echoes, and this area deviates from the normal characteristics.
- **Blood Flow Signals:** The nodule shows punctate blood flow signals (red and yellow areas displayed by color Doppler), indicating the presence of neovascularization within the nodule, which raises suspicion for malignancy.

**2.Nodule Characteristics:**
- **Differentiation of Benign and Malignant:** Features such as hypoechoic, unclear boundaries, microcalcifications (if present), and an aspect ratio > 1 increase the risk of malignancy. The nodule exhibits some signs of malignancy and requires further clinical assessment.

**3.Surrounding Structures:**
- **Normal Appearance:** The low-echo background of muscle tissue (such as the sternocleidomastoid muscle) around the thyroid is visible, which is consistent with anatomical positioning, and there are no signs of abnormal thickening or infiltration.

**4.Blood Flow Distribution:**
- **Overall Evaluation:** The thyroid shows a mild increase in blood flow signals, but it does not reach the degree of the "fire sea sign," which may be related to local irritation from the nodule.

**Conclusion:**
A suspicious malignant nodule is present in the middle and upper part of the thyroid. Fine-needle aspiration biopsy (FNAB) and regular follow-up are recommended. The patient's thyroid function tests (such as TSH, FT4) and clinical symptoms should be considered for a comprehensive judgment.

**Question 3:** Observe and analyze the characteristics of the nodule in this thyroid ultrasound image, and provide a TI-RADS classification and subsequent management recommendations.

Based on the analysis of the provided ultrasound image of thyroid, the following diagnostic recommendations are made:

**1. TI-RADS Classification Analysis (Based on ACR Standards):**
- Composition: Solid (2 points)；Echo: Hypoechoic (2 points)；Shape: Taller than wide (3 points)；Margin: Ill-defined/microlobulated (2 points)；Foci of echogenicity: None (0 points)
- **Total Score:** 9 points TI-RADS Category 5 (High suspicion for malignancy, probability of malignancy > 85%).

**2. Key Abnormal Features:**
- **Solid hypoechoic nodule:** Indicates increased cell density and elevated risk of malignancy.
- **Taller than wide shape:** Vertical growth pattern, associated with aggressive behavior.
- **Ill-defined/microlobulated margin:** Reflects incomplete capsule or local infiltration.
- **Abundant blood flow signals:** Neovascularization within the nodule, supporting the possibility of malignancy.

**3. Subsequent Management Recommendations:**
- **Fine-needle aspiration biopsy (FNAB):** Highly recommended; immediate biopsy when the nodule is ≥1.5 cm in maximum diameter; if <1.5 cm but with high-risk factors (such as enlarged cervical lymph nodes, family history), biopsy is still necessary.
- **Imaging supplements:** Contrast-enhanced ultrasound to assess blood perfusion patterns, or elastography to determine tissue stiffness.
- **Follow-up strategy:** If biopsy is not performed immediately, repeat ultrasound within 3-6 months to monitor the growth rate of the nodule.
- **Clinical evaluation:** Test thyroid function (TSH, FT4), calcitonin (to exclude medullary thyroid carcinoma), and perform ultrasound of cervical lymph nodes.
- **Multidisciplinary consultation:** Joint discussion of complex cases by radiologists, endocrinologists, and surgeons.

**Conclusion:**
The ultrasonic features of the nodule highly suggest malignancy, classified as TI-RADS Category 5, and pathological type needs to be confirmed by FNAB. If diagnosed with thyroid cancer, the extent of surgery (such as total thyroidectomy) and subsequent treatments (such as Iodine-131 therapy) will be determined based on TNM staging. Early intervention can significantly improve prognosis.

## Multiround Dialogue \ Heart

**Question 1**: Analyze the scanned area and the section of this ultrasound image.

The ultrasound appearance of this image is a section of the cardiac area, specifically a parasternal long-axis view of the left ventricle. The main features are as follows:

**1.Anatomical Structures:**
- Cardiac chamber structures are visible, including the left ventricle, left atrium, and interventricular septum；Clear display of the mitral or aortic valve level (color Doppler flow signals can assist in determining valve positions).

**2.Blood Flow Characteristics:**
- Color Doppler shows blood flow signals (red and blue alternating), indicating changes in blood flow direction (such as blood flow through the valves during diastole/systole)；Areas with higher blood flow velocity may correspond to stenosis or regurgitation lesions, which require further analysis with spectral Doppler.

**3.Section Localization Basis:**
- Probe position: Scanning path from the parasternal or apical region；Beam direction: Perpendicular to the long axis of the left ventricle.

This section is commonly used to assess valvular dysfunction (such as mitral regurgitation, aortic stenosis) and the coordination of left ventricular wall motion.

**Question 2:** Observe the echocardiogram to diagnose whether there is mitral regurgitation, analyze the possible causes, and provide subsequent management recommendations.

Based on the analysis of the provided ultrasound image of heart, the following diagnostic recommendations are made:

**1. Ultrasound Evidence**
- **Color Doppler Flow Imaging (CDFI):** A blue flow jet from the left ventricle (LV) into the left atrium (LA) during systole, with the width of the regurgitant jet accounting for >30% of the LA transverse diameter, indicating moderate-to-severe regurgitation.
- **Two-Dimensional Echocardiography:** The anterior leaflet of the mitral valve is bulging in the A3 region, with prolapse at the C1-C2 junction and poor coaptation (a "flail-like" change can be seen in the C2 region).
- Left Atrial/Left Ventricular Enlargement: Left atrial volume index >34 mL/m², left ventricular end-diastolic diameter >55 mm, indicating long-term volume overload.

**2. Possible Etiologies**
- **Primary Mitral Valve Prolapse:** Leaflet redundancy, myxomatous degeneration (common in young and middle-aged women).
- **Degenerative Changes:** Age-related loss of leaflet elasticity, chordal shortening.
- **Secondary Factors:** Left ventricular enlargement (e.g., hypertensive heart disease, dilated cardiomyopathy) leading to mitral annular dilation; ischemia/functional impairment of the papillary muscles due to coronary artery disease; mitral valve leaflet destruction caused by infective endocarditis.

**3. Subsequent Management Recommendations**
- **Medical Therapy:** Control hypertension (target <130/80 mmHg) to delay left ventricular remodeling. Use ARNI/β-blockers to improve left ventricular function (if ejection fraction <60%).
- **Surgical Evaluation:** Indications include worsening heart function (NYHA II-IV class), left ventricular EF ≤60%, and pulmonary hypertension (systolic pressure ≥50 mmHg). Prioritize mitral valve repair (preserves valve structure, better long-term outcomes); valve replacement if repair is not feasible.
- **Follow-Up Monitoring:** Repeat echocardiography every 6-12 months to assess regurgitation volume, left ventricular function, and pulmonary artery pressure. If atrial fibrillation or pulmonary hypertension develops, initiate targeted treatments (e.g., anticoagulation, antihypertensive therapy).

**Conclusion:**
The patient has moderate-to-severe mitral regurgitation. The underlying cause and severity should be comprehensively assessed in combination with the patient's clinical manifestations (e.g., palpitations, exertional dyspnea, history of pulmonary edema) and laboratory tests (BNP, ECG). If the patient has risk factors for coronary artery disease, coronary CTA is recommended to exclude ischemic lesions.

## I.3 COMPARISON OF GPT IN DIAGNOSING VARIOUS ULTRASOUND SITES

---

**Carotid Artery**

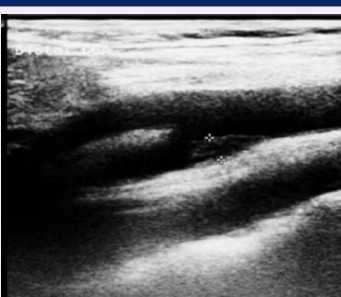

**Question:** Observe this carotid artery ultrasound image and perform an analysis diagnosis

**Doctor's observation:**
• A hypoechoic plaque is visible on the posterior wall at the bifurcation of the carotid artery, with a regular shape, smooth margins, and heterogeneous internal echoes.

**Dolphin:** Based on the analysis of the provided carotid artery ultrasound image, the following diagnostic recommendations are made:

**1. Plaque Characterization:**
- **Location:** The plaque is attached to the inner wall of the carotid artery and may affect hemodynamics.
- **Morphology and Echo:** It presents as a hypoechoic or mixed-echo area, suggesting the presence of a lipid core or hemorrhage within the plaque.
- **Surface Features:** No clear signs of surface irregularity or ulceration are observed, but stability needs to be confirmed with additional tests.
- **Size and Distribution:** The plaque has a moderate length, and specific measurements are needed to assess the risk of stenosis.

**2. Vulnerable Plaque Risk Assessment:**
- **Vulnerability Indicators:** Hypoechoic areas indicate lipid deposition and possible hemorrhage (heterogeneous internal echoes).
- **Thrombotic Risk:** No thrombus signs are seen in the lumen, but thrombus formation is possible if the plaque ruptures, which needs to be evaluated clinically.

**3. Clinical Recommendations:**
- **Further Examinations:** Intravascular ultrasound (IVUS) or photoacoustic imaging is recommended to clarify plaque stability.
- **Risk Factor Management:** Evaluate the control of hypertension, hyperlipidemia, diabetes, etc.
- **Therapeutic Interventions:** Lifestyle modifications (smoking cessation, dietary control) and pharmacological treatment (statins).

**Conclusion:**
The image shows a hypoechoic or mixed-echo plaque on the carotid artery wall. There is a need to be vigilant about the possibility of a vulnerable plaque. It is recommended to develop an individualized treatment plan based on comprehensive imaging and clinical data to reduce the risk of cardiovascular events.

**GPT:** Based on the analysis of the provided carotid arter ultrasound image, the following diagnostic recommendations are made:

**1. Imaging Basics**
- The image is a longitudinal B-mode ultrasound of the carotid artery.
- An irregular hypoechoic/mixed-echo area is visible within the lumen, located on the vessel wall.
- Localized thickening of the vessel wall and partial narrowing of the lumen are observed.
- The affected area is marked with a "cross."

**2. Ultrasound Findings Analysis**
- **Vessel Wall:** Localized thickening of the intima-media complex (IMT).
- **Plaque Appearance:** Irregular hypoechoic plaques are visible, some with areas of stronger echoes, suggesting the possible presence of fibrous components or calcification.
- **Lumen Condition:** The lumen is narrowed due to the lesion, but the image does not show complete occlusion.
- **Blood Flow (Color Doppler Not Used):** The image does not display blood flow signals, so the degree of stenosis is primarily assessed using B-mode imaging.

**3. Clinical Evaluation and Recommendations**
- **Further Examination:** It is recommended to perform color Doppler ultrasound and spectral analysis to measure the peak systolic velocity (PSV) and assess the degree of stenosis. CT angiography (CTA) or magnetic resonance angiography (MRA) may be necessary for further confirmation if needed.
- **If the stenosis is mild (<50%):** Lifestyle intervention + medication (statins for lipid-lowering, antiplatelet drugs).
- **If the stenosis is moderate to severe (≥50%):** Strict pharmacological treatment and consideration of interventional or surgical procedures (carotid endarterectomy or stenting) are necessary.

**Conclusion:**
The image shows localized thickening of the carotid artery wall with irregular hypoechoic plaques, indicating carotid atherosclerosis and lumen stenosis. Further color Doppler ultrasound is recommended to assess the degree of stenosis and to intervene based on clinical risk factors.

## Fetal

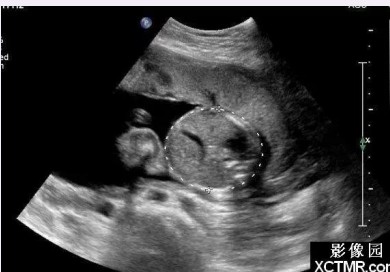

**Question:** Observe this Fetal ultrasound image and perform an analysis diagnosis

**Doctor's observation:**
- The ultrasound shows a transverse section of the fetal upper abdomen, with no obvious abnormalities.

**Dolphin:** Based on the analysis of the provided fetal ultrasound image, the following diagnostic recommendations are made:

**1. Fetal Abdominal Measurement Area:**
- The central area displays the fetal abdomen, with measurement markers indicating the measurement points for abdominal circumference (the standard plane is the transverse section of the fetal abdomen).
- It is important to check whether the measurement position conforms to the standard plane (fetal spine in the sagittal plane, abdomen in the transverse plane, and feet in the sagittal plane) to exclude measurement errors.

**2. Assessment of Surrounding Structures:**
- The Doppler display area shows no obvious abnormal blood flow signals (such as umbilical vein cysts or vascular malformations), but it is necessary to pay attention to the limitations of image quality.
- There are no obvious abnormalities in the surrounding tissues (such as placental position and amniotic fluid volume), indicating that the current image does not show other key structures (spine, limbs, etc.).

**3. Clinical Significance Indications:**
- Abdominal circumference measurement needs to be combined with gestational age. If it exceeds the normal range (such as abdominal circumference >95th percentile or <5th percentile), it is necessary to be alert to intrauterine growth restriction or post-term pregnancy.
- The image does not show the standard sections (such as the fetal chest and heart), and it is recommended to supplement other sections to assess overall development.

- **4. Limitations Statement:**
- Image resolution may affect the clear display of small structures (such as the midline of the brain and heart valves).
- It is necessary to make a comprehensive judgment in combination with clinical history (such as obstetric history and fetal echocardiography results).

**Suggested Examination Direction:**
Supplement detailed ultrasound examinations of the fetal chest, heart, and limbs. If necessary, perform fetal echocardiography to assess cardiac structure.

**GPT:** Based on the analysis of the provided fetal ultrasound image, the following diagnostic recommendations are made:

**1. Image Observation**
- **Imaging Area:** The image shows a fetal ultrasound section during a routine mid-pregnancy check-up. A transverse section of the fetal head is visible, and the measurement frame encircles the section for measuring the fetal biparietal diameter (BPD) or head circumference (HC).
- **Structural Presentation:** The fetal skull is formed, with the outer edge appearing as a clear, high-echo arc. The intracranial echoes show the midline structure, and the ventricles are symmetrical. The midline of the brain and the thalamus are visible, with no obvious hydrocephalus or skull defects. The fetal face and parts of the body are not shown in this section.
- **Measurement Significance:** The diameter value within the ultrasound measurement circle is used to assess the size of the fetal head.
- **Common Measurement Indicators:** Biparietal diameter (BPD) assesses gestational age and fetal development level; head circumference (HC) is used together with BPD to judge fetal head development and intrauterine growth.

**2. Analysis and Diagnostic Implications**
- This image is a transverse section of the fetal head for ultrasound, with BPD/HC measurement in progress. Imaging findings: The fetal head has a normal developmental shape, with a continuous and complete skull, the intracranial midline is centered, and there are no obvious deformities. It can be used to assess gestational age, estimate fetal weight, and monitor fetal growth and development.

**3. Clinical Recommendations**
- Compare the measured BPD/HC with the standard fetal development curve according to the gestational age to assess whether the fetus is growing normally. Routinely combine other parameters (femur length FL, abdominal circumference AC) for a comprehensive assessment of fetal weight and development. If abnormalities are found (such as an overly large/small head or asymmetry), further follow-up or MRI is needed for clarification.

**Conclusion:**
The image shows a transverse section of the fetal head, with ultrasound measuring BPD/HC. The fetal skull shape and intracranial structures are normal, with no obvious deformities, and can be used for gestational age assessment and intrauterine development monitoring.

## Colon

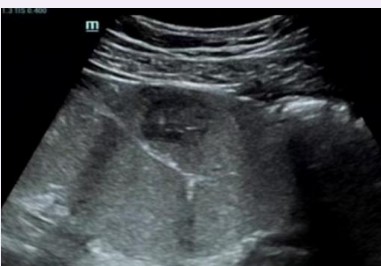

**Question:** Observe this ultrasound image of the colon after contrast agent filling and perform an analysis and diagnosis.

**Doctor's observation:**
- The multilayer structure of the bowel wall appears normal. Within the colon, an inhomogeneously hypoechoic mass is visible, presenting as a nodule with regular shape and clear margins, and with posterior acoustic shadowing. Further examination is recommended to clarify the nature of the space-occupying lesion.

**Dolphin:** Based on the analysis of the provided ultrasound image of the colon after contrast agent filling, the following diagnostic recommendations are made:

**1. Anatomical Structure Assessment**
- Whether the lumen of the stomach, duodenum, and other parts is unobstructed, and whether there is stenosis or dilation. Areas with good contrast filling (such as the antrum of the stomach and the descending part of the duodenum) indicate no obvious stenosis of the lumen.
- Whether the thickness of the gastric and intestinal walls is normal (judged in combination with the patient's body type and fasting state), and the uniformity of the wall echoes.

**2. Key Pathological Features**
- **Space-occupying lesions:** If localized thickening, protrusion, or hypo/hyperechoic masses of the intestinal wall are found, malignancy (such as gastric cancer, intestinal cancer), polyps, or foreign bodies should be suspected. Further differentiation of benign and malignant lesions should be made in combination with color Doppler (blood flow signals) and enhanced CT/PET-CT.
- **Inflammation or ulcers:** Irregular thickening of the intestinal wall with mucosal rupture and ulcer depression suggests chronic inflammation (such as gastritis, intestinal ulcers) or infection (such as Crohn's disease).
- **Obstruction or stenosis:** Obstruction of contrast agent passage in the antrum or duodenum suggests obstructive lesions (such as tumors, stenosis caused by Helicobacter pylori infection).

**3. Diagnostic Recommendations**
- **In combination with clinical symptoms:** If the patient has abdominal pain, vomiting, weight loss, etc., the possibility of malignant tumors should be considered; if accompanied by diarrhea and bloody stools, inflammatory bowel disease should be ruled out.
- **Further examinations:** Gastroscopy biopsy to clarify the pathology of the mucosa or the nature of the tumor; CT/MRI to assess the extent of the lesion and the presence of metastasis; biochemical blood tests to screen for tumor markers (such as CEA, CA19-9).

**Notes:**
**Contraindications for contrast agents:** Use with caution in patients with renal insufficiency to avoid kidney damage.
**Image quality:** If the image is not clear enough, it is recommended to repeat the examination or combine it with other imaging methods.

**GPT:** Based on the analysis of the provided ultrasound image of the colon after contrast agent filling, the following diagnostic recommendations are made:

**1. Overall Image Observation**
- **Transducer Position and Section:** The image presents a transverse section of the abdomen, with visible gastric wall and partial intestinal structures. The wall structures are clear, and the contrast agent fills the cavity, forming anechoic or hypoechoic areas.
- **Gastrointestinal Lumen Appearance:** The contrast agent within the lumen (oral ultrasound contrast agent or liquid) appears as a hypoechoic area with relatively clear boundaries. Mild mixed echo shadows within the contrast agent are visible, suggesting the possible presence of gas bubbles or mucosal folds.
- **Gastrointestinal Wall Appearance:** The wall echo stratification is still visible, with distinct echoes between the mucosa-submucosa and the muscularis propria. There is no obvious diffuse thickening of the gastrointestinal wall.

**2. Possible Imaging Indications**
- **Normal Contrast Filling Appearance:** After filling, the lumen appears hypoechoic, and the wall structures are visible. Mucosal folds may change with the flow of the contrast agent.
- **Abnormal Signals to Note:** If there is irregular thickening of the wall or disappearance of the stratification structure, inflammation (such as gastritis, enteritis) or tumors should be considered. If there is a fixed hyperechoic area with acoustic shadow within the lumen, foreign bodies or calcifications should be considered. If the contrast agent passage is obstructed or the lumen is significantly dilated, obstruction should be alerted to.

**3. Diagnostic Directions in Combination with Clinical Findings**
- If the patient has upper abdominal pain and dyspepsia, gastritis or peptic ulcer should be primarily considered.
- If there is significant weight loss and anemia, gastrointestinal tumors should be ruled out.
- If the presentation is abnormal defecation or abdominal distension, mechanical or functional intestinal obstruction should be alerted to.

**Summary:**
This gastrointestinal ultrasound image after contrast agent filling shows good filling of the gastrointestinal lumen and relatively clear wall layers. No obvious masses or significant abnormalities are currently seen. However, further judgment needs to be made in combination with patient symptoms and clinical examinations. If there are clinically suspicious symptoms, it is recommended to further examine with gastroscopy/CT.

## Brachial Plexus

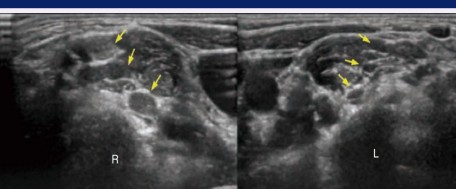

**Question:** Observe and analyze the comparison of this ultrasound image of bilateral interscalene brachial plexus.

**Doctor's observation:**
- Comparison of the ultrasound images on both sides reveals that the right brachial plexus nerve roots are thickened.

**Dolphin:** Based on the analysis of the provided brachial plexus nerve image, the following diagnostic recommendations are made:

**1. Ultrasound Feature Analysis**
- **Nerve Thickening**: The right brachial plexus nerve is significantly thicker than the left. Normally, the diameter of the brachial plexus nerve should be less than 2 mm. Abnormal thickening suggests possible pathological changes in the nerve (such as nerve edema, inflammation, or compression).
- **Unclear Internal Structure**: The internal fascicular structure of the right nerve is not clearly visible. Normally, the brachial plexus nerve shows a "honeycomb-like" hypoechoic pattern with hyperechoic septa of the nerve fascicles. The unclear structure may be associated with nerve edema, fibrosis, or inflammation.
- **Irregular Borders**：The borders of the right nerve are not clear, with a blurred demarcation from the surrounding soft tissue, which may indicate involvement of the nerve sheath or local inflammatory response.
- **Inhomogeneous Echo**: The internal echo distribution of the right nerve is inhomogeneous, showing patchy hypoechoic or mixed echo patterns, suggesting possible pathological changes such as edema, hemorrhage, or fibrosis.

**2. Differential Diagnosis**
- **Neuritis or Neurofibroma:** The edema and thickening of the interscalene brachial plexus nerve need to be differentiated from neurofibromas, perineuriomas, or inflammatory neuropathies. Neurofibromas usually appear as well-defined hypoechoic masses with rich internal blood flow signals, while inflammatory changes have blurred borders and inhomogeneous echoes.
- **Nerve Compression Syndrome**: It is necessary to assess whether the patient has a history of trauma, cervical spondylosis, or local space-occupying lesions (such as tumors, hematomas) causing nerve compression. Ultrasound can show whether the nerve continuity is interrupted and whether there is compression by surrounding tissues.
- **Nerve Trauma or Postoperative Changes**：If the patient has a history of trauma or surgery, nerve injury or postoperative scar formation leading to nerve thickening should be considered.

**3. Recommendations for Further Examination**
- **Clinical Correlation:** The patient's medical history (such as trauma, cervical spondylosis, pain location), physical examination (such as nerve traction test, Tinel's sign), and neurophysiological examinations (such as electromyography, nerve conduction velocity) should be combined.
- **Contrast Ultrasound or MRI**: If ultrasound suggests space-occupying lesions within or around the nerve, contrast ultrasound or MRI is recommended to clarify the nature of the lesion (such as neurofibroma, perineurioma).
- **Ultrasound-Guided Biopsy (if necessary)**：For cases highly suspected of malignant tumors, ultrasound-guided biopsy may be considered.

**Conclusion**
The thickening of the right interscalene brachial plexus nerve with structural abnormalities should primarily consider lesions such as neuritis or neurofibroma. It is recommended to combine clinical and further imaging examinations to clarify the diagnosis. During the diagnostic process, it is necessary to exclude nerve compression or other space-occupying lesions to avoid misdiagnosis.

**Note:** The above analysis is based on ultrasound image features. The specific diagnosis should be combined with the patient's clinical manifestations and other examination results.

**GPT:** Based on the analysis of the provided brachial plexus nerve image, the following diagnostic recommendations are made:

**1. Image Basic Information**
- **Location**：Bilateral upper clavicular areas of the neck root, showing the brachial plexus.
- **Orientation**：The left image is marked as R (right side), and the right image is marked as L (left side).
- **Annotation:** The yellow arrow indicates the hypoechoic/medium-low echo oval nerve trunk along the course of the brachial plexus.

**2. Normal Ultrasound Appearance**
- **Ultrasound Appearance**：On ultrasound cross-sectional imaging, the brachial plexus appears as a "honeycomb-like" or "bunch of grapes" pattern, characterized by uniform, well-defined, and regularly sized hypoechoic circular/oval structures separated by hyperechoic interstitial tissue.
- **Location**：Within the interscalene groove, the brachial plexus is typically located between the anterior and middle scalene muscles.

**3. Comparative Image Analysis**
- **Right side (R):** The nerve plexus is clearly visible with a relatively regular arrangement; multiple hypoechoic oval structures are present with uniform size and shape; the hyperechoic septa between nerve fascicles are distinct and structurally intact; this represents a normal or slightly thickened appearance.
- **Left side (L):** The nerve plexus appears diffusely thickened with multiple nerve trunks showing irregular hypoechoic patterns; the boundaries of nerve fascicles are relatively blurred with disordered arrangement; some nerve cross-sections are enlarged with a swollen appearance. This suggests pathological changes, commonly seen in: brachial plexus neuritis (edema and thickening due to inflammation); compression or space-occupying lesions (tumors, nodules, fibrosis); neuropathy (such as demyelinating diseases).

**Conclusion**
Right brachial plexus: The ultrasound appearance is essentially normal；Left brachial plexus: Diffuse thickening and disordered arrangement are shown, indicating brachial plexus pathology, with a higher likelihood of brachial plexus neuritis or space-occupying compressive lesions.

1998
1999
2000
2001
2002
2003
2004
2005
2006
2007
2008
2009
2010
2011
2012
2013
2014
2015
2016
2017
2018
2019
2020
2021
2022
2023
2024
2025
2026
2027
2028
2029
2030
2031
2032
2033
2034
2035
2036
2037
2038
2039
2040
2041
2042
2043
2044
2045
2046
2047
2048
2049
2050
2051

## Lung

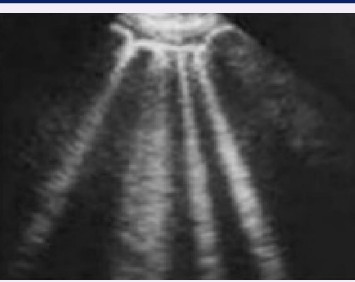

**Question:** Observe this lung ultrasound image and perform an analysis diagnosis

**Doctor's observation:**
- Abnormal thickening of the pleural line;
- diffuse B-lines;
- Consider pulmonary interstitial syndrome.

**Dolphin:** Based on the analysis of the provided lung ultrasound image, the following diagnostic recommendations are made:

**1. Imaging Findings**
- **Diffuse Increase in B-lines:** Dense B-lines (>5 per field of view) are visible throughout the lung fields, presenting a "white lung" appearance, indicative of severe interstitial syndrome. In some areas, B-lines are confluent, which may be associated with alveolar fluid exudation or fibrosis.
- **Pleural Line Abnormalities:** Bilateral thickening and local irregularity of the pleural line are observed, which may be related to chronic inflammation or previous lung injury.
- **Focal Consolidation:** A hypoechoic area with indistinct margins is seen in the left middle lobe, without clear air bronchograms, raising suspicion for atypical pathogen infection (e.g., COVID-19) or organizing pneumonia.

**2. Diagnostic Recommendations**
- **Viral Interstitial Pneumonia (e.g., COVID-19, influenza virus):** The presence of diffuse B-lines and the absence of typical air bronchograms in the consolidation areas are consistent with the pattern of viral injury.
- **Early Acute Respiratory Distress Syndrome (ARDS):** Severe interstitial syndrome with consolidation is noted; the diagnosis should be confirmed in conjunction with the $PaO_2$ /$FiO_2$ ratio and clinical course.
- **Cardiogenic Pulmonary Edema:** This should be excluded by assessing heart function, with BNP and echocardiography as ancillary tests.
- **Acute Exacerbation of Pulmonary Fibrosis:** A history of underlying idiopathic pulmonary fibrosis would support this diagnosis.

- **3. Clinical Recommendations**
- **Urgent Assessment:** Measure arterial blood gas (ABG) to evaluate oxygenation status; perform chest CT to delineate the distribution of lesions (typical viral pneumonia shows peripheral reticular opacities).
- **Laboratory Tests:** Conduct viral screening for SARS-CoV-2 antigen/PCR, influenza virus antigen; assess inflammatory markers such as CRP, IL-6, and D-dimer.
- **Treatment Direction:** Initiate antiviral therapy (e.g., nirmatrelvir/ritonavir for COVID-19); restrict fluid intake to prevent worsening of pulmonary edema; consider corticosteroids if necessary (weighing the risk of infection).
- **Precautions:** Be vigilant for multifactorial etiologies of diffuse B-lines (e.g., viral infection with secondary bacterial infection); monitor changes in ultrasound findings dynamically: a reduction in B-lines suggests effective treatment, while persistent increase may indicate progression of ARDS.

**Conclusion:**
The lung ultrasound findings suggest a high likelihood of viral interstitial pneumonia. It is recommended to promptly initiate antiviral therapy and complete imaging and laboratory investigations.

**GPT：** Based on the analysis of the provided lung ultrasound image, the following diagnostic recommendations are made:

**1. Image Characteristics**
- Multiple bright hyperechoic lines are visible in the image, extending downward from the pleural line, almost parallel to the ultrasound beam, and spanning the entire screen.
- These lines appear and disappear in synchrony with respiratory movements and are referred to as B-lines or comet-tail artifacts.
- B-lines originate from the pleural line, extend vertically downward, have strong echoes, do not attenuate with depth, and are spaced approximately evenly.

**2. Normal vs. Abnormal**
- **Normal:** In healthy lung tissue, 0–2 isolated B-lines may occasionally be present.
- **Abnormal:** If there are ≥3 B-lines in a single scanning plane, and they are diffuse and bilaterally distributed, this suggests involvement of the lung interstitium.

**3. Possible Clinical Significance**
- Increased B-lines are commonly seen in: pulmonary edema (cardiogenic/non-cardiogenic → most typical manifestation); interstitial lung diseases (e.g., fibrosis, inflammation causing interstitial thickening); ARDS (acute respiratory distress syndrome); early pneumonia or interstitial involvement. If combined with irregular or fragmented pleural lines, it also suggests chronic interstitial disease.

**Conclusion:**
This image shows multiple distinct B-lines, indicating an interstitial syndrome of the lungs. The most common causes need to be differentiated based on medical history: acute heart failure/pulmonary edema (if associated with bilateral lower limb edema, cardiac dysfunction); ARDS/pneumonia (if there is infection, respiratory distress); chronic interstitial lung disease (if there is long-term dyspnea, abnormal lung function). Further clarification of the etiology is needed in combination with clinical history (heart failure, infection, ARDS risk) and other imaging (chest X-ray/CT).

## Liver

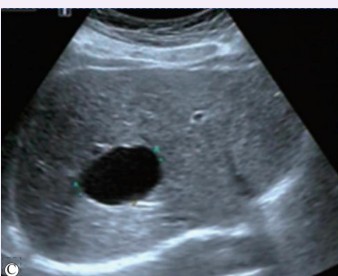

**Question:** Observe this liver ultrasound image and perform an analysis diagnosis

**Doctor's observation:**
- An oval-shaped anechoic mass is visible, with a regular shape, clear boundaries, good acoustic transmission, and marked posterior acoustic enhancement.
- Considered as a simple hepatic cyst, regular follow-up is recommended.

**Dolphin:** Based on the analysis of the provided liver ultrasound image, the following diagnostic recommendations are made:

**1.Ultrasound Image Feature Description:**
- **Lesion Morphology and Location:** An oval-shaped hypoechoic area is visible in the lower left region, with clear boundaries and a size of approximately 2.5×2.0 cm.
- **Internal Echo:** The interior presents as uniformly anechoic, with no septations or solid components observed.
- **Posterior Acoustic Effect:** There is a marked enhancement of the posterior echo.

**2.Diagnostic Analysis: Simple Hepatic Cyst**
- **Basis:** The smooth boundary, anechoic interior, and posterior enhancement are consistent with the typical appearance of a simple cyst.
- **Differential Diagnosis Excluded:** Abscess (no wall thickening, heterogeneous internal echo, or surrounding inflammatory signs); Hemangioma/Solid Tumor (not hyperechoic and lacking blood flow signals); Biliary Cyst (no biliary dilation or connecting structures observed).

**3.Recommendations:**
- **No Special Treatment Required:** Simple hepatic cysts are mostly benign. Asymptomatic individuals can undergo regular ultrasound follow-up (e.g., annually).
- **Clinical Correlation:** If the patient experiences abdominal pain, cyst enlargement, or other liver diseases, further evaluation is needed.
- **Additional Examinations:** If the diagnosis is in doubt, abdominal CT/MRI with contrast can be chosen to clarify the nature.

**Conclusion:**
The ultrasound findings in this case highly suggest a simple hepatic cyst. It is recommended to monitor with routine follow-up, with no intervention needed at present.

**GPT:** Based on the analysis of the provided liver ultrasound image, the following diagnostic recommendations are made:

**1.Basic Image Observation**
- **Background:** The liver parenchyma appears with uniform medium echo and relatively clear structure.
- **Lesion:** A round/ovoid hypoechoic area is seen within the liver parenchyma (anechoic interior with clear boundaries).
- **Acoustic Features:** The interior is purely anechoic (black), with no internal echo structures; the edge is smooth, with a thin wall; there is a marked enhancement of the posterior echo (good acoustic transmission).

**2.Ultrasound Sign Analysis**
- This appearance is consistent with the typical features of a cystic lesion: anechoic, thin-walled, with posterior echo enhancement and regular shape.

**3.Considering the Typical Imaging Features, the Most Common Consideration is**
- **Hepatic Cyst:** The most common benign liver lesion, with typical imaging features: clear boundaries, thin wall, anechoic, and posterior echo enhancement.
- **Other Differential Diagnoses (to be considered if the presentation is atypical):** Cystic metastasis (usually with irregular borders, internal septations/nodules); Liver abscess (may show flocculent or septated echoes within the cyst, with unevenly thick walls); Parasitic cyst (hydatid disease) (calcified cyst wall, internal septations, or "cyst within a cyst" sign).

**Conclusion:**
The image shows a typical simple hepatic cyst. If the patient is asymptomatic (e.g., no abdominal pain, compressive symptoms), no special treatment is usually required, and only regular ultrasound follow-up is necessary. If the cyst is large (>5 cm) and symptomatic, interventional drainage or surgical intervention may be considered.

## Knee

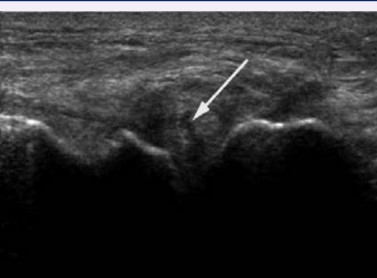

**Question:** Observe this knee ultrasound image and perform an analysis diagnosis

**Doctor's observation:**
- An irregular hypoechoic band is visible within the lateral meniscus of the knee joint, indicating a meniscal tear.
- Meniscal Protrusion

**Dolphin:** Based on the analysis of the provided knee ultrasound image, the following diagnostic recommendations are made:

**1. Anatomical Abnormalities:**
- Meniscal Echogenicity Disruption: Localized interruption of meniscal echogenicity is observed, with the tear region presenting as linear hypoechoic or anechoic areas, contrasting with the surrounding normal meniscal hyperechoic tissue.

**2. Associated Signs (Clinical Correlation Required):**
- **Effusion:** Meniscal tears often accompany joint capsule distension or synovial thickening, manifesting as hypoechoic effusion regions.
- **Joint Space Widening:** The tear may cause the meniscus to lose support, potentially leading to knee instability or widened joint spaces.

**3. Differential Diagnosis:**
- **Ligament Injury (e.g., Lateral Collateral Ligament Injury):** Ultrasound may reveal ligament thickening, heterogeneous echogenicity, or rupture.
- **Cartilage Damage:** Cartilage defects appear as hypoechoic or anechoic areas with exposed subchondral bone, though meniscal tears are more common on the lateral side of the knee.
- **Synovitis:** Synovial proliferation presents as hypoechoic tissue, with or without blood flow signals, distinct from the hyperechoic band-like structures seen in meniscal tears.

**4. Clinical Significance and Recommendations:**
- **Symptom Correlation:** Patients may experience medial knee pain, difficulty in flexion and extension, or a clicking sensation (impingement symptoms). Severe tears can lead to locking (meniscal fragments impinging the joint space).
- **MRI Examination:** To delineate the extent of the tear and assess the relationship between the meniscus and cartilage complex, providing more detailed information for surgical planning.
- **Ultrasound-Guided Treatment:** After a clear diagnosis, consider ultrasound-guided injections of corticosteroids or hyaluronic acid to alleviate symptoms.

**Conclusion**
The ultrasound image clearly shows a longitudinal tear in the lateral meniscus, which needs to be confirmed further with clinical symptoms and MRI. Timely intervention is recommended to prevent the progression of chronic injury.

**GPT:** Based on the analysis of the provided knee ultrasound image, the following diagnostic recommendations are made:

**1. Image Observation**
- **Location:** Knee joint (the image shows a longitudinal section below the patella or near the patellar tendon).
- **Arrow-Indicated Area:** Below the skin and subcutaneous tissue, near the joint surface, an irregular hypoechoic band can be seen.
- **Echo Characteristics:** A hypoechoic/irregular area is visible in the joint space; the boundary is indistinct; there is no obvious acoustic shadow below the ultrasound beam, but the continuity of the joint surface is affected.

**2. Possible Pathological Manifestations**
- **Meniscal Injury:** Characterized by hypoechoic clefts/irregular echoes within or at the edge of the meniscus, commonly seen in sports injuries or degeneration of the knee joint.
- **Articular Cartilage Damage:** If the arrow points to the articular cartilage surface of the femoral condyle, it suggests cartilage thinning or abnormal echoes.
- **Synovial Hypertrophy or Inflammatory Changes:** If accompanied by joint effusion, it may indicate synovitis or arthritis.

**3. Preliminary Diagnostic Opinion**
- **Ultrasound Findings:** Hypoechoic linear/band-like areas with abnormal continuity of the joint space structure. The image appearance is most consistent with meniscal tear/injury.
- **Further Recommendations:** Correlate with clinical symptoms (knee pain, locking, limited mobility); MRI examination (more sensitive for diagnosing meniscal injuries).

**Conclusion:**
The ultrasound image of the knee joint suggests a possible meniscal tear, which needs to be further clarified in conjunction with clinical findings and MRI.

## Thyroid

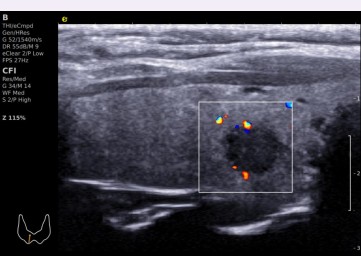

**Question:** Observe this Thyroid ultrasound image and perform an analysis diagnosis

**Doctor's observation:**
- A very hypoechoic nodule is visible in the middle of the lower pole of the right thyroid lobe, with a relatively regular shape, indistinct borders, and fine hyperechoic calcifications within. CDFI shows relatively rich blood flow signals.
- TI-RADS Category 4C nodule, ultrasound contrast or biopsy is recommended.

**Dolphin:** Based on the analysis of the provided Thyroid ultrasound image, the following diagnostic recommendations are made:

**1. Ultrasound Feature Analysis**
- **Shape and Border:** The nodule has an irregular shape with indistinct borders and microlobulated features, which suggest a possible malignancy.
- **Echo Characteristics:** The interior presents as hypoechoic or very hypoechoic, possibly with microcalcifications (confirmation should be combined with the presence of microcalcification points), which is typical of papillary thyroid carcinoma.
- **Blood Flow Signal (CDFI):** Rich blood flow signals are visible within the nodule, distributed in a multi-point or tree-branch pattern, indicating active tumor angiogenesis and supporting a malignant tendency.
- **Completeness of the Halo:** An incomplete or absent halo further suggests a possible malignancy.

**2. Diagnostic Recommendations**
- **TI-RADS Classification:** The comprehensive features may be categorized as Class 4 (suspicious for malignancy) or Class 5 (highly suspicious for malignancy).

**3. Management Suggestions:**
- **Fine-Needle Aspiration (FNA):** To clarify the pathological nature, especially sampling from areas with microcalcifications and rich blood flow.
- **Regular Follow-Up:** If surgery is not immediately indicated, it is recommended to re-examine with ultrasound every 3-6 months to monitor changes in the nodule.
- **Clinical Assessment:** Combine with patient symptoms (such as hoarseness, enlarged cervical lymph nodes), laboratory tests (TSH, thyroid antibodies), and family history for a comprehensive judgment.

**Conclusion:**
Indistinct borders, irregular shape, hypoechoic interior with possible microcalcifications, rich blood flow signal and incomplete halo all suggest a higher risk of malignancy, particularly the likelihood of papillary thyroid carcinoma. According to TI-RADS classification, it may belong to Class 4 or 5, and fine-needle aspiration (FNA) is recommended to confirm the pathological diagnosis.

Notes:
The final diagnosis should be combined with multimodal imaging (such as elastography) and clinical data. If the patient has Hashimoto's thyroiditis or other endocrine diseases, the diagnostic threshold should be adjusted. It is recommended to develop an individualized treatment plan under the guidance of a professional physician.

**GPT:** Based on the analysis of the provided Thyroid ultrasound image, the following diagnostic recommendations are made:

**1. Basic Image Information**
- The image is a transverse section of the thyroid gland (indicated by the icon in the lower left corner). The colored Doppler flow signals (CFI) are shown within the right frame. The background echo demonstrates that the nodule is markedly hypoechoic.

**2. Nodule Sonographic Features**
- **Echo:** Predominantly hypoechoic, significantly lower than the surrounding normal thyroid tissue.
- **Shape:** Roughly round/ovoid, with the length-to-width ratio close to 1.
- **Border:** Relatively clear, but with some slight roughness visible.
- **Internal Structure:** The echo is relatively uniform, with no obvious cystic components.
- **Calcification:** No typical hyperechoic points or acoustic shadows indicative of microcalcifications are seen.
- **Vascularity:** Color Doppler shows a few blood flow signals within and around the nodule (the vascularity is not abundant, but present).

**3. TI-RADS Risk Assessment Key Points**
- Hypoechoic (2 points);Shape: Width greater than height (0 points);Border: Partially irregular (2 points);Calcification: No typical calcifications (0 points);Vascularity: A few blood flows inside (indicating activity)
- The total score is approximately 4 points (ACR TI-RADS Category 4, moderate to high suspicion).

**4. Clinical Significance and Recommendations**
- This nodule is considered suspicious (TI-RADS 4) and needs to be evaluated in conjunction with its size to determine whether a biopsy is necessary. For TI-RADS 4 nodules >1 cm, fine-needle aspiration (FNA) is recommended; if <1 cm, follow-up observation is suggested (recheck in 6-12 months), with particular attention to any increase in size, increased vascularity, or new calcifications.
- If the patient has high-risk factors for thyroid cancer (family history, abnormal cervical lymph nodes, etc.), early FNA is more inclined.

**Conclusion:**
The thyroid solid hypoechoic nodule, with limited vascularity and no typical microcalcifications, is classified as TI-RADS 4. It is recommended to decide on FNA or follow-up based on the size and risk factors.

2214
2215
2216
2217
2218
2219
2220
2221
2222
2223
2224
2225
2226
2227
2228
2229
2230
2231
2232
2233
2234
2235
2236
2237
2238
2239
2240
2241
2242
2243
2244
2245
2246
2247
2248
2249
2250
2251
2252
2253
2254
2255
2256
2257
2258
2259
2260
2261
2262
2263
2264
2265
2266
2267

## Skin

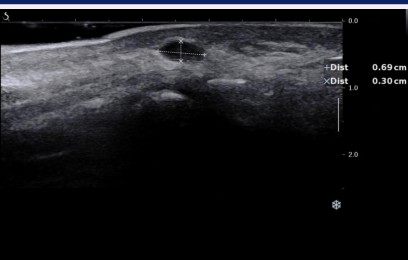

**Question:** Observe this skin ultrasound image and perform an analysis diagnosis

**Doctor's observation:**
- An anechoic cystic nodule is visible in the subcutaneous tissue of the palm, with a regular shape and clear margins.

**Dolphin:** Based on the analysis of the provided skin ultrasound image, the following diagnostic recommendations are made:

**1. Nodule Characteristics**
- **Location and Shape:** The nodule is located between the epidermis and dermis, with an uneven surface, clear margins, and a round/oval shape.
- **Size:** The maximum diameter is approximately 0.69cm × 0.30cm, indicating a small volume.
- **Echo:** Internally, it presents as hypoechoic or anechoic, with a high-echo halo visible at the edge (possibly related to the cyst wall or fibrous capsule).
- **Posterior Acoustic Shadowing:** No significant acoustic shadowing is present, suggesting that the contents are fluid or soft tissue, rather than calcified or solid hard tissue.

**2. Differential Diagnosis**
- **Epidermoid Cyst:** Typically characterized by a well-defined hypo/anechoic nodule with a high-echo edge and possible internal septations, which matches the features of this case.
- **Lipoma:** Usually a uniformly hypoechoic solid mass, but commonly located in the deeper subcutaneous tissue and not directly connected to the skin surface.
- **Sebaceous Cyst:** Similar to an epidermoid cyst but occurs near the sebaceous gland orifice and may be associated with dilated hair follicles or sebaceous ducts on the skin surface.
- **Fibroma:** Solid with high or mixed echoes and heterogeneous internal echoes, but the edges are usually not as smooth as those of a cyst.

**3. Exclusion of Malignancy**
- The nodule's small volume and clear margins make malignancy highly unlikely. If it were a rare malignancy such as dermatofibrosarcoma protuberans, it would typically present with a larger volume, infiltrative growth, and abundant blood flow.

**4. Recommendations**
- **Clinical Correlation:** A comprehensive judgment should be made in conjunction with the medical history (e.g., nodule growth rate, pain, itching, etc.).
- **Follow-up Observation:** If asymptomatic and with typical ultrasound features, regular ultrasound follow-up (e.g., every 6-12 months) is recommended to monitor stability.
- **Other Tests:** Fine-needle aspiration biopsy may be performed if necessary to aid in the diagnosis, especially for inflammatory lesions such as tuberculous granulomas.

**Conclusion:**
The ultrasound features are suggestive of a benign cystic lesion (such as an epidermoid cyst), but a final diagnosis should be made in combination with clinical and pathological results. Close observation and follow-up are recommended.

**GPT:** Based on the analysis of the provided skin ultrasound image, the following diagnostic recommendations are made:

**1. Imaging Basics**
- A focal lesion is visible in the superficial layer of the skin, measured with calipers at approximately 0.69 cm × 0.30 cm. The lesion is located relatively superficially, at the junction of the skin/dermis and subcutaneous fat.

**2. Ultrasound Characteristics of the Lesion**
- **Boundary and Shape:** The lesion is oval-shaped, with a long axis significantly larger than the short axis. The boundary is clear, with a complete capsule-like echo visible.
- **Echo:** The lesion is hypoechoic, with a homogeneous internal structure, no significant areas of high echo or anechoic regions suggestive of liquefaction. There is mild posterior acoustic enhancement, suggesting a cystic or fluid component.

**3. Ultrasound Diagnosis Considerations**
- **Epidermoid Cyst :** Superficial, oval-shaped, well-defined, hypoechoic or mixed echo, with possible mild posterior enhancement.
- **Sebaceous Cyst:** May have heterogeneous echoes if containing sebum or keratin, but this case is more homogeneous.
- **Other Small Nodules (such as Fibroma):** However, lipomas are generally larger and have internal striated echoes, which do not match this case. Overall, the characteristics of this lesion are most consistent with an epidermal cyst.

**4. Clinical Recommendations**
- If the lesion is not red, swollen, or painful, the likelihood of it being benign is high, and it can be monitored or removed at a later date. If it has recently increased in size, is tender to touch, or the overlying skin is red, secondary infection should be considered, and clinical measures such as anti-inflammatory treatment or drainage may be necessary.

**Conclusion:**
The superficial hypoechoic oval nodule in the skin, with clear boundaries and homogeneous internal structure, measuring approximately 0.69×0.30 cm, is most likely an epidermal cyst.

## Heart

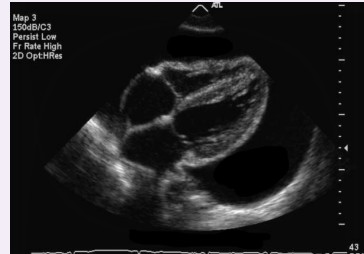

**Question:** Observe this ultrasound image of the heart and perform an analysis and diagnosis.

**Doctor's observation:**
- Echocardiography shows the separation of the visceral and parietal pericardial layers, with an anechoic area in between. This indicates that part or all of the heart is surrounded by a fluid dark area, suggesting pericardial effusion.

**Dolphin:** Based on the analysis of the provided ultrasound image of the heart, the following diagnostic recommendations are made:

**1. Identification of Anechoic Areas**
- An anechoic area is clearly visible around the inferior wall of the left ventricle, which is typical of pericardial effusion.
- The thickness of the effusion should be measured in multiple planes, but if the local thickness exceeds 20 mm or if it is widely distributed, it indicates moderate to large amounts of effusion.

**2. Cardiac Motion Abnormalities**
- There may be restricted cardiac diastole (e.g., "D-shaped" interventricular septum), but this needs to be confirmed in multiple planes.
- The heart may exhibit abnormal swinging ("rocking sign"), which suggests increased pericardial pressure.

**3. Hemodynamic Impact**
- The heart rate (BPM=43) is significantly low, which may be related to pericardial effusion compressing the conduction system or cardiac tamponade.
- Right ventricular diastolic collapse and tricuspid regurgitation, which are signs of tamponade, need to be observed.

**4. Clinical Recommendations:**
- **Emergency Assessment:** Immediate measurement of vital signs (blood pressure, pulse, respiration) to check for hypotension, pulsus paradoxus, and jugular venous distension (Beck's triad).
- **Further Examination:** Additional ultrasound planes (parasternal long-axis/short-axis, subxiphoid view) should be used to clarify the total amount and distribution of effusion. CT or MRI enhancement may be necessary if required.
- **Treatment Decision:** If there is evidence of cardiac tamponade (e.g., right heart collapse, hemodynamic instability), emergency pericardiocentesis should be performed; if there is no tamponade but a large amount of effusion, close monitoring and etiological treatment (e.g., anti-inflammatory, anti-tuberculosis) should be carried out.

**Conclusion:**
- Pericardial effusion (moderate to large amount), with a need to be vigilant for the risk of cardiac tamponade.

**Notes:**
Ultrasound is a dynamic assessment tool that needs to be used in real-time clinical judgment.The nature of the effusion (exudative/transudative) needs to be clarified through medical history, laboratory tests (e.g., BNP, infection markers), and effusion analysis.

**GPT:** Based on the analysis of the provided ultrasound image of the heart, the following diagnostic recommendations are made:

**1. Image Observation**
- **View:** This is a typical apical four-chamber view (Apical 4-Chamber View, A4C). It allows simultaneous visualization of the left atrium (LA), left ventricle (LV), right atrium (RA), and right ventricle (RV).
- **Chamber Size:** The left ventricular cavity appears to be of normal size, with no significant enlargement. The right ventricular cavity does not appear significantly larger compared to the left ventricle, and there is no apparent severe right ventricular enlargement. Neither the left atrium nor the right atrium shows significant dilation.
- **Ventricular Wall Thickness and Motion:** The left ventricular wall thickness seems to be normal, with no apparent signs of hypertrophy. Ventricular systolic function is difficult to assess in a static image, but the overall chamber morphology appears normal.
- **Heart Valves:** The mitral and tricuspid valves are visible, with no apparent deformities in their morphology. However, due to the static nature of the image, the opening and closing of the valves and any regurgitation cannot be confirmed.
- **Other Observations: There is no apparent pericardial effusion.**

**2. Preliminary Analysis**
- **Overall Image Indication:** The cardiac chamber size and structure appear to be essentially normal.There is no evidence of significant ventricular or atrial enlargement or pericardial effusion.In this frame, there are no apparent signs of lesions in the mitral and tricuspid valves.
- **Limitations of Static Imaging:** The static image has significant limitations and cannot assess cardiac function (ejection fraction, EF), valve function (regurgitation, stenosis), or hemodynamic conditions.

**3. Clinical Suggestions**
- **Doppler Echocardiography:** To assess left ventricular systolic function and wall motion.To evaluate whether there is regurgitation or stenosis of the mitral and tricuspid valves and to assess pulmonary artery pressure.
- **Cardiac Function Parameters:** Such as LVEF, E/A ratio, E/e′, etc.
- **Clinical Background:** Whether the patient has clinical manifestations of heart failure, valvular heart disease, congenital heart disease, etc.

**Conclusion:**
From this single frame of the apical four-chamber view in the echocardiogram, the structure of the cardiac chambers appears to be essentially normal, with no significant dilation or deformities, and no apparent pericardial effusion. However, since this is a static image, it cannot replace a complete echocardiographic examination and further diagnosis is needed in combination with dynamic imaging and Doppler.

## Prostate

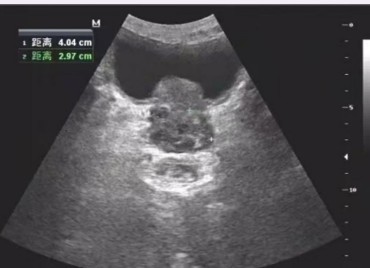

**Question:** Observe this prostate ultrasound image and perform an analysis diagnosis

**Doctor's observation:**
- The prostate is enlarged, with multiple hypoechoic areas visible inside. These areas have irregular shapes and unclear boundaries, with heterogeneous internal echoes. Multiple areas of strong echoes indicative of calcifications are also visible. The wall of the bladder neck is irregularly thickened and protruding.

**Dolphin:** Based on the analysis of the provided prostate ultrasound image, the following diagnostic recommendations are made:

**1. Abnormal Shape and Size**
- The prostate is significantly enlarged, with an anteroposterior diameter (4.04 cm) and a transverse diameter (2.97 cm) exceeding the normal range (the normal anteroposterior diameter for adults is approximately 3 cm, and the transverse diameter is approximately 4 cm). The contour is irregular and the boundary is indistinct, suggesting the possible presence of a space-occupying lesion.

**2. Echo Characteristics**
- **Hypoechoic Areas:** Multiple hypoechoic areas are visible within the prostate. These areas have unclear boundaries and irregular shapes, which is typical of prostate cancer (such as cancer in the transition zone).
- **Strong Echo Calcifications:** Some areas are accompanied by strong echo calcifications, which need to be distinguished from calcifications in prostate cancer (prostate cancer calcifications are usually scattered and punctate, with weak acoustic shadows).

**3. Diagnostic Recommendations**
- **Biopsy:** Targeted biopsy (such as MRI-TRUS fusion biopsy) is recommended to confirm the pathological diagnosis.
- **PSA Testing:** Serum PSA levels should be combined to assess the risk of prostate cancer (PSA>4ng/ml or an elevated PSA/PSA ratio suggests the possibility of malignancy).
- **Multimodal Imaging:** MRI examination is recommended (if PI-RADS score is ≥4, biopsy is suggested) to observe the depth of tumor invasion and the integrity of the capsule.
- **Diagnosis and Treatment:** If prostate cancer is confirmed, a treatment plan should be developed based on the stage (determined through MRI and staging assessment), such as radical prostatectomy, radiotherapy, or endocrine therapy.

**4. Differential Diagnosis**
- **Benign Prostatic Hyperplasia (BPH):** Typically presents with symmetrical enlargement and nodules with clear boundaries, with fewer blood flow signals.
- **Prostatitis:** Often accompanied by diffuse hypoechoic changes and thickening of the capsule. In the acute phase, abscesses may be visible (irregular hypoechoic areas with ring-like blood flow).

**Conclusion:**
This case highly suspects prostate cancer, and a definitive diagnosis and guidance for individualized treatment require biopsy and multimodal assessment.

**GPT:** Based on the analysis of the provided prostate ultrasound image, the following diagnostic recommendations are made:

**1. Imaging Basics**
- The image is a longitudinal section of the prostate via transabdominal ultrasound with a full bladder. The bladder is located at the top of the image, appearing as a full anechoic dark area. Below the bladder, prostate tissue is visible and has been measured.

**2. Prostate Sonographic Appearance**
- **Size and shape:** The anteroposterior diameter of the prostate is >4 cm, indicating enlargement (the normal anteroposterior diameter is generally <3 cm, with a total volume <20 ml).The overall shape is roughly symmetrical, but the margins are not very smooth. The capsule contour is basically clear, but some areas are irregular.
- **Internal Echo:** The internal echo is medium, uneven, with nodular or calcified high-echo areas visible.

**3. Diagnostic Considerations**
- **Benign Prostatic Hyperplasia (BPH):** Common in elderly men, characterized by increased volume and uneven internal echoes. An anteroposterior diameter >4 cm indicates significant enlargement.
- **Prostate Nodule/Calcification:** Localized areas of strong echo visible in the image may be calcifications, commonly seen in the context of chronic inflammation or hyperplasia.
- **Prostate Cancer:** If there are hypoechoic areas, asymmetrical nodules, or invasion of the capsule or bladder base, malignancy should be suspected.

**4. Clinical Significance**
- **Ultrasound suggests:** Enlarged prostate with uneven internal echoes and localized strong echo foci (calcifications).
- **Clinical correlation:** If the patient has symptoms such as difficulty urinating, frequent urination, urgency, or urinary retention, benign prostatic hyperplasia is highly suspected.
- **Further examination:** If PSA is elevated or there are abnormalities in the digital rectal examination, transrectal ultrasound (TRUS) + MRI or biopsy should be performed to rule out prostate cancer.

**Conclusion:**
This transabdominal ultrasound shows: Enlarged prostate volume (anteroposterior diameter 4.04 cm, transverse diameter 2.97 cm). Uneven internal echoes with calcifications are present. Benign prostatic hyperplasia with calcifications is considered. It is recommended to further assess with PSA and MRI, and if necessary, perform a biopsy to confirm the diagnosis.

## Breast

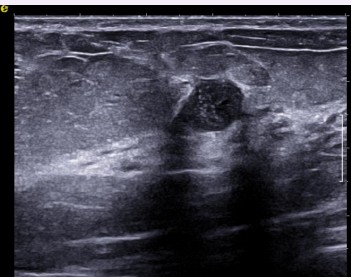

**Question:** Observe this breast ultrasound image and perform an analysis diagnosis

**Doctor's observation:**
- An irregular-shaped solid hypoechoic nodule with slightly lobulated margins and heterogeneous internal echoes is visible in the breast. Multiple areas of strong echoes indicative of calcifications are also visible. BI-RADS 4C. Ultrasound contrast imaging or biopsy is recommended.

**Dolphin:** Based on the analysis of the provided breast ultrasound image, the following diagnostic recommendations are made:

**1. Location and Shape**
- The lesion is located in the upper outer quadrant of the right breast (3 o'clock position), with a size of approximately 2.0cm × 1.5cm.
- The ==shape is irregular, with indistinct or spiculated margins== (a malignant sign).

**2. Echo**
- The internal echo is ==heterogeneous, predominantly hypoechoic,== which may suggest tumor heterogeneity (a possible malignancy).
- Mild posterior acoustic shadowing is present, which is commonly seen in lesions with high density or abundant fibrous components (other features should be considered together).

**3. Diagnostic Implications**
- **Higher possibility of malignancy:** ==Irregular shape, spiculated margins, and heterogeneous internal echo ar==e typical malignant signs.
- **Lower possibility of benignity:** Fibroadenoma and cysts need to be ruled out, but the morphological features are more indicative of malignancy.

**4. Recommendations**
- **Pathological biopsy:** Ultrasound-guided fine-needle aspiration (FNA) is recommended to confirm the pathological type.
- **Multimodal imaging:** Mammography and MRI should be combined to assess the presence of calcifications or invasive features.
- **Clinical evaluation:** A comprehensive judgment should be made in combination with the patient's age, family history, and symptoms (such as the growth rate of the lump).
- **Follow-up:** If the biopsy is negative but the imaging highly suspicious, short-term follow-up is recommended every 3-6 months.

**Conclusion:**
==Solid hypoechoic nodule with irregular shape, slightly lobulated margins, heterogeneous internal echoes, and multiple microcalcifications visible within, all of which suggest a higher risk of malignancy.== According to the BI-RADS classification, it may belong to category ==4 or 5==, and fine-needle aspiration (FNA) is recommended to confirm the pathological diagnosis

**Note:**
The final diagnosis needs to be confirmed by pathological results. The above analysis is for clinical reference only.

**GPT:** Based on the analysis of the provided breast ultrasound image, the following diagnostic recommendations are made:

**1. Imaging Basics**
- This is a longitudinal B-mode ultrasound image of the breast. A focal space-occupying lesion is visible within the breast parenchyma. The lesion is located beneath the superficial glandular layer and has an irregular shape.

**2. Ultrasound Characteristics of the Lesion**
- **Shape:** ==Irregular, with the long axis nearly equal to the short axis. In some areas, the vertical diameter is greater than the horizontal diameter ("vertical orientation").==
- **Boundary:** Indistinct, with spiculated or lobulated margins, appearing irregular.
- **Echo:** Hypoechoic, with heterogeneous internal echoes and some areas showing even lower echoes.
- **Posterior Features:** Acoustic shadowing is visible, suggesting high tissue density.
- **Surrounding Tissue:** Local breast structures are compressed and disordered.
- **Calcifications:** Punctate areas of high echo are visible within, possibly indicating microcalcifications.

**3. Analysis of Ultrasound Findings**
- These characteristics are consistent with the common ultrasound appearance of malignant breast nodules: irregular shape, spiculated margins, hypoechoic/ heterogeneous internal echoes, acoustic shadowing, and possible microcalcifications.
- **According to the BI-RADS classification:** The lesion is consistent with BI-RADS category 4C-5 (high suspicion of malignancy).

**4. Recommendations**
- **Breast Biopsy (CNB/FNA):** To confirm the pathological nature.
- **Breast MRI:** To assess the extent of the lesion and the contralateral breast if necessary.
- **Clinical Correlation:** If associated with nipple discharge, skin retraction, or axillary lymphadenopathy, high vigilance is warranted.

**Conclusion:**
A hypoechoic, irregular, spiculated, and acoustically shadowed nodule is visible within the breast, which is highly suggestive of a malignant breast tumor. A biopsy is recommended to confirm the diagnosis.

## Kidney

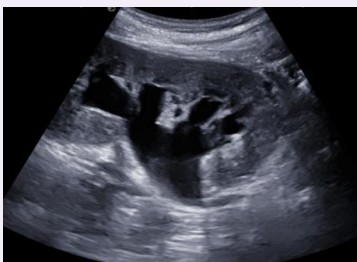

**Question:** Observe this kidney ultrasound image and perform an analysis diagnosis

**Doctor's observation:**
- The renal pelvis and calyces are markedly dilated, with a distinct anechoic fluid dark area visible inside, indicating hydronephrosis.

**Dolphin:** Based on the analysis of the provided kidney ultrasound image, the following diagnostic recommendations are made:

**1. Ultrasound Image Analysis**
- **Kidney Morphology and Structure:** The kidney contour is irregular, with an abnormal shape in the lower region, suggesting the possibility of space-occupying lesions (such as cysts, tumors, or calyceal dilation). The renal parenchyma shows inhomogeneous echoes, especially in the lower region, which may indicate chronic lesions or local tissue abnormalities (such as fibrosis or scarring).
- **Pelvic and Calyceal Hydronephrosis:** ==The renal pelvis and calyces are dilated, indicating hydronephrosis.== Hydronephrosis is usually caused by urinary tract obstruction (such as stones, tumors, or strictures), and further assessment is needed in combination with patient symptoms. ==The left kidney shows more significant hydronephrosis, suggesting that the obstruction may be located in the left kidney or the left ureter.==
- **Blood Flow and Kidney Size:** The kidney size is close to normal, but further assessment of blood flow (such as resistance index, RI) is needed to determine whether there is renal function impairment. ==Hydronephrosis of the left kidney may lead to compensatory enlargement of the contralateral kidney, and comparison of the sizes of both kidneys is required.==

**2. Comprehensive Diagnosis**
- ==**Hydronephrosis (mainly in the left kidney):**== Indicates urinary tract obstruction, which needs to be confirmed in combination with clinical symptoms (such as flank pain, hematuria, and dysuria) and urinalysis, as well as urography.
- **Inhomogeneous Renal Parenchymal Echoes:** May be due to chronic kidney disease, renal cysts, or space-occupying lesions (such as tumors). Further differentiation is needed through contrast-enhanced ultrasound or CT/MRI.

**3. Recommendations**
- **Clinical Assessment:** Inquire whether the patient has symptoms such as flank pain, hematuria, fever, and dysuria. ==Check the size of both kidneys and the presence of bilateral hydronephrosis, and assess renal function (serum creatinine, blood urea nitrogen).==
- **Imaging Examinations:** CT urography to clarify the site of obstruction (stones, tumors, or anatomical abnormalities); contrast-enhanced ultrasound to differentiate the nature of space-occupying lesions in the renal parenchyma (cysts vs. tumors).
- **Treatment Direction:** If the obstruction is caused by stones or blood clots, urological intervention is required (extracorporeal shock wave lithotripsy, ureteroscopic stone removal). If it is due to tumors or structural abnormalities, surgical or interventional treatment is needed.

**Summary**
==Ultrasound suggests hydronephrosis of the left kidney with renal parenchymal abnormalities.== Further clarification of the etiology is needed in combination with clinical symptoms and additional imaging examinations to formulate an individualized treatment plan.

**GPT:** Based on the analysis of the provided kidney ultrasound image, the following diagnostic recommendations are made:

**1. Image Overview**
- The image displays a longitudinal section of the kidney. The renal parenchymal echoes are visible, and the central renal sinus and collecting system structures are clearly defined. Multiple irregular, anechoic tubular/cystic dilatations are observed centrally.

**2. Ultrasound Feature Analysis**
- Collecting System Dilatation: The renal pelvis and calyces show marked anechoic expansion, presenting a "staghorn" appearance. The renal parenchyma is thinned, with relatively enhanced echoes, indicating compression and atrophy. ==The internal fluid dark areas are uniformly hypoechoic, with no distinct solid echo masses.== The acoustic features are typical of renal pelvis and calyceal dilatation.

**3. Diagnostic Considerations**
- The findings are consistent with hydronephrosis: multiple dilated calyces and renal pelvis communicating with each other, with thinning of the renal parenchyma. Possible causes include upper urinary tract obstruction (e.g., ureteral stones, stricture, extrinsic compression); bladder outlet obstruction (e.g., prostatic hyperplasia); or congenital ureteropelvic junction stenosis.

**4. Clinical Recommendations**
- **Grading (Reference for hydronephrosis grading):** Mild: only renal pelvis dilatation; Moderate: renal pelvis + calyceal dilatation; Severe: marked dilatation of renal pelvis and calyces with thinning of renal parenchyma. Based on the image, this case presents with moderate-to-severe hydronephrosis.
- **Further investigations:** Perform a urinary system color Doppler ultrasound and CTU (CT urography) to clarify the site and cause of obstruction. If there are symptoms such as flank pain, hematuria, or fever, be alert for stones or infection.
- **Management:** Depends on the cause. Stones can be treated with lithotripsy or stone removal, strictures require surgical correction, and acute obstruction should be relieved as soon as possible.

**Conclusion**
The image shows moderate-to-severe hydronephrosis with thinning of the renal parenchyma, likely caused by upper urinary tract obstruction. Further imaging and clinical investigations are needed to confirm the etiology.

## Fetal head

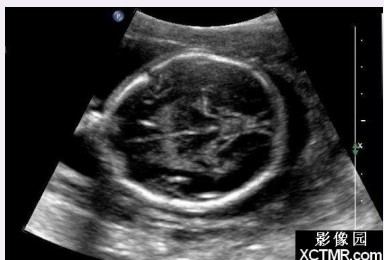

**Question:** Observe this fetal head ultrasound image and perform an analysis diagnosis

**Doctor's observation:**
- Ultrasound shows that the fetal cranial brain structure is basically normal, with the brain midline centered, no dilation of the ventricular system, uniform echoes of the white matter, and no obvious developmental abnormalities or lesions.

**Dolphin:** Based on the analysis of the provided fetal head ultrasound image, the following diagnostic recommendations are made:

**1. Integrity of Brain Structures**
- The midline of the brain is clearly visible and centrally located, with symmetry on both sides and no displacement or abnormal partitioning.
- The ventricular system (ventricles) shows no dilation, and the choroid plexus echoes are uniform, with no dilation or effusion.
- The cerebral cortex echoes are uniform, with no signs of underdevelopment, bleeding, or calcification foci.

**2. Other Observations**
- The cerebellum and brainstem are not fully displayed and need to be comprehensively assessed in combination with other sections (such as the coronal plane).
- The white matter echoes are uniform, with no cystic changes or areas of high echo (such as calcification).

**3. Recommendations**
- A comprehensive evaluation should be made in combination with the clinical history and other examination results of the pregnant woman (such as NT, serum screening, etc.).
- Fetal MRI or follow-up ultrasound should be performed if necessary to rule out occult abnormalities (such as underdevelopment of the cerebellum, small cysts, etc.).
- Imaging suggests the need for further follow-up to ensure normal fetal brain development.

**Conclusion**
The fetal cranial ultrasound shows normal brain structure, with no obvious malformations or developmental abnormalities. Note that this analysis is based on a single image section, and the actual diagnosis requires a combination of multiple sections and clinical information.

**GPT:** Based on the analysis of the provided fetal head ultrasound image, the following diagnostic recommendations are made:

**1. Overall Image Condition**
- The image is a typical transverse section of the fetal head (routine obstetric examination). The ring-shaped high echo of the skull is clearly visible, with a regular contour. The intracranial structures are clearly displayed, belonging to the transverse section of the brain (the plane for measuring biparietal diameter/head circumference).

**2. Observation of Intracranial Structures**
- **Falx cerebri (midline structure):** The midline echo band is visible, centrally located, with no obvious displacement.
- **Thalamus:** Located on both sides of the midline, with symmetrical isoechogenic structures.
- **Lateral ventricles:** Visible on both sides of the thalamus, with no obvious dilation in the size of the cavities.
- **Cavum septi pellucidi (CSP):** A small cystic dark area is visible at the front, indicating its presence, which is a normal manifestation.
- **Cerebellum/Posterior fossa:** No obvious abnormalities are seen in this section (the cerebellum usually needs to be observed separately in the transverse section of the cerebellum).

**3. Diagnostic Analysis**
- From the image presentation: The fetal skull has a regular shape, with no defects or deformities; the intracranial midline is centered, with no displacement; the thalamus is symmetrical, and the size of the lateral ventricles is normal, with no obvious signs of hydrocephalus; the presence of the cavum septi pellucidi indicates normal neurological development. These manifestations all suggest that the fetal cranial brain structure is normal, with no obvious abnormalities.

**4. Clinical Significance**
- This image is the biparietal diameter/head circumference measurement plane, mainly used for the assessment of fetal growth and development.
- If the biometric values (BPD, HC) are within the gestational age range, it indicates normal fetal cranial development.
- It can also rule out some cranial malformations (such as anencephaly, ventricular dilation, absence of the cavum septi pellucidi, etc.).

**Conclusion:**
This ultrasound image of the fetal head transverse section shows the skull, thalamus, lateral ventricles, and cavum septi pellucidi, with an overall normal cranial brain structure and no obvious signs of abnormality.

## Pancreas

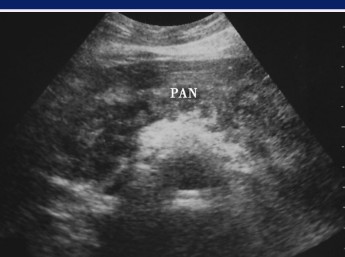

**Question:** Observe this ultrasound image of the pancreas and perform an analysis and diagnosis.

**Doctor's observation:**
- Ultrasound reveals diffuse enlargement of the pancreatic gland, with a full appearance and markedly convex anterior border, and an irregular surface. Multiple irregular hypoechoic areas are visible within the glandular tissue.

**Dolphin:** Based on the analysis of the provided ultrasound image of the pancreas, the following diagnostic recommendations are made:

**1. Abnormalities in Pancreatic Morphology and Structure**
- **Pancreatic Enlargement:** The pancreas appears spindle-shaped or locally bulging, indicating an inflammatory response.
- **Heterogeneous Echogenicity:** The pancreatic parenchyma shows patchy hypoechoic areas, which may be caused by edema or necrotic tissue.
- **Indistinct Margins:** The pancreas has unclear boundaries with surrounding tissues, suggesting inflammatory infiltration or exudation.

**2. Advanced Imaging Features**
- **Pancreatic Duct Dilation:** The main pancreatic duct may be dilated throughout or in certain sections (specific measurements should be provided). The presence of the "double duct sign" (dilation of both the pancreatic and bile ducts) may indicate a pancreatic head mass or obstruction at the ampulla of Vater.
- **Peripancreatic Fluid Collection:** An anechoic area around the pancreas (representing exudate) supports the diagnosis of acute pancreatitis.

**3. Differential Diagnosis**
- **Acute Pancreatitis (AP):** Typical ultrasound findings include pancreatic enlargement, heterogeneous echogenicity, and peripancreatic fluid collection. These should be correlated with elevated serum amylase/lipase levels.
- **Pancreatic Tumor (e.g., Pancreatic Cancer):** Be alert for a hypoechoic mass in the pancreatic head, uniform dilation of the pancreatic duct ("pancreatic duct type"), lymph node metastasis, or liver metastasis. However, tumors typically show richer internal blood flow signals on Color Doppler Flow Imaging (CDFI).
- **Chronic Pancreatitis:** Atrophy of the pancreas, calcifications, and a beaded appearance of the pancreatic duct are more indicative of chronic lesions.

**4. Suggested Further Examinations**
- **Laboratory Tests:** Serum amylase, lipase, C-reactive protein (CRP), and calcium levels.
- **Contrast-Enhanced Imaging:** Contrast-enhanced CT/MRI to clarify the nature of space-occupying lesions **Endoscopic Ultrasound (EUS):** To assess peripancreatic lymph nodes, vascular invasion, and the extent of tumor infiltration.

**Conclusion:**
The findings of pancreatic enlargement with heterogeneous echogenicity, peripancreatic fluid collection, and pancreatic duct dilation are consistent with the typical presentation of Acute Pancreatitis (AP). However, it is important to rule out neoplastic lesions, and further laboratory tests and contrast-enhanced imaging are recommended.

**GPT:** Based on the analysis of the provided ultrasound image of the pancreas, the following diagnostic recommendations are made:

**1. Overall Observation**
- **Organ Localization:** The image is labeled with "PAN" in the center, indicating that the scanning target is the pancreas. The overall display range of the pancreas appears somewhat indistinct, with medium to slightly hypoechoic tissue visible.
- **Echogenicity Characteristics:** Normally, the pancreas should present with a uniform, fine-granular medium echogenicity, slightly higher than the liver and lower than the spleen. In this image, the pancreatic echogenicity seems not entirely uniform, with some areas appearing coarser.
- **Margins and Surrounding Tissues:** The margins of the pancreas are not sharply defined, and there is no clear separation of the surrounding fat planes. No focal mass or obvious cystic structure is observed.

**2. Imaging Findings**
- **Diffuse Lesion Possibility:** Heterogeneous echogenicity and ill-defined margins can be seen in the early stages of acute pancreatitis (edema, heterogeneous parenchyma) or chronic pancreatitis (fibrosis, coarse parenchyma). If accompanied by pancreatic enlargement and peripancreatic fluid collection, acute inflammation is more likely.
- **No Focal Lesion Indications:** No focal hypoechoic or mixed echogenic mass is observed, and there are no typical imaging features of pancreatic tumors.
- **Limited Display:** Pancreatic ultrasound is often affected by bowel gas interference, and some details may be obscured. If clinical suspicion of a lesion is high, further CT/MRCP examination is recommended.

**3. Possible Diagnostic Directions in Combination with Clinical Findings**
- If the patient presents with acute abdominal pain and elevated blood amylase/lipase levels, acute pancreatitis should be considered.
- If the patient has chronic abdominal pain, weight loss, and a history of diabetes, chronic pancreatitis should be considered.
- If a tumor or cystic lesion is clinically suspected, enhanced CT or MRI is recommended for further evaluation.

**Summary:**
The pancreatic ultrasound image shows heterogeneous echogenicity and ill-defined margins of the pancreatic parenchyma, which are more suggestive of inflammatory changes (acute or chronic pancreatitis). No definite mass or cystic lesion is observed. It is recommended to combine laboratory indicators (amylase, lipase), clinical manifestations, and further CT/MR examination for a definitive diagnosis.

## Cervix

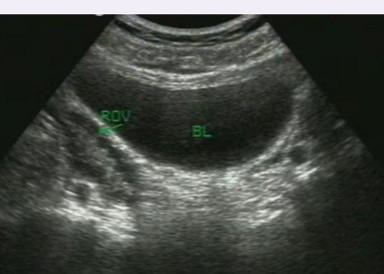

**Question:** Observe this ultrasound image of the cervix and perform an analysis and diagnosis.

**Doctor's observation:**
- Ultrasound reveals multiple anechoic space-occupying lesions within the left ovary, with regular shapes, clear boundaries, and good acoustic transmission, suggesting ovarian cysts.

**Dolphin:** Based on the analysis of the provided ultrasound image of the cervix, the following diagnostic recommendations are made:

**1. Key Imaging Features**
- **Morphology and Size :** Enlarged ovaries with unilocular or multilocular cystic structures visible. If the cyst wall thickness is <3 mm, smooth, and without septations, it is mostly benign (e.g., follicular cyst). Thick walls or papillary projections should raise suspicion for tumors.
- **Internal Echoes :** Anechoic or homogeneous low echoes suggest simple cysts (physiological or serous cystadenoma). A reticular or punctate high echo may indicate a mucinous cystadenoma or endometriotic cyst (chocolate cyst).
- **Blood Flow Signals (CDFI) :** A small amount of blood flow in the cyst wall or septum indicates a high likelihood of benignity. Rich blood flow in solid areas should raise suspicion for malignancy (e.g., ovarian cancer).

**2. Differential Diagnosis**
- **Physiological Cysts :** Follicular cysts (unilocular, thin-walled, no blood flow, often spontaneously regress), corpus luteum cysts (may have annular blood flow, related to the menstrual cycle).
- **Pathological Tumors :** Serous/mucinous cystadenomas (multilocular, poor acoustic transmission, rarely malignant), borderline tumors (cystic-solid, slow-growing, pathological confirmation required), malignant tumors (predominantly solid, rich blood flow, with ascites or lymph node enlargement).
- **Other Lesions :** Endometriotic cysts (thick-walled, dense punctate echoes inside), hydrosalpinx (sausage-shaped, separate from the ovary).

**3. Clinical Recommendations**
- **Short-term Follow-up :** Asymptomatic, simple cysts (<5 cm): It is recommended to recheck ultrasound in 3 months to observe changes.
- **Further Examination :** Contrast-enhanced MRI or CT to clarify the nature and extent of the cyst; tumor marker testing: such as CA125, HE4 (ovarian cancer screening).
- **Surgical Indications :** Cysts >5 cm, continuous enlargement, suspicious for malignancy, or with significant symptoms (e.g., torsion, rupture), laparoscopic exploration is recommended.

**Conclusion :**
Ovarian cystic space-occupying lesions are more likely to be benign, but malignancy should be ruled out in combination with clinical findings. It is recommended to complete the collection of medical history and laboratory tests, and if necessary, a multidisciplinary team (MDT) consultation should be held to develop an individualized plan.

**GPT:** Based on the analysis of the provided ultrasound image of the cervix, the following diagnostic recommendations are made:

**1. Overall Observation**
- **Anatomical Localization:** The image is labeled with "BL" for the bladder, which appears as a distended anechoic (black) area serving as an acoustic window. The image is also labeled with "ROV" for the right ovary. Behind the bladder, the echo structure appears relatively uniform.
- **Ovarian Appearance:** The contour is clear, and the size range is visible. The internal echoes present a relatively uniform background, with some small anechoic round areas likely representing follicles.

**2. Imaging Findings**
- **Normal Appearance:** On transabdominal ultrasound, normal ovaries typically show follicles, usually as multiple small round anechoic areas.
- **No Clear Abnormalities:** The low-echo, small cystic structures visible at the ROV location are consistent with follicular characteristics. There are no obvious solid masses or cystic lesions with irregular wall thickness. No large cysts or complex cystic echoes (such as those with internal septations or solid nodules) are seen. No imaging features of solid masses are present.

**3. Clinical Correlation**
- **Fertile Women:** The image is consistent with the normal appearance during the follicular development phase. The size and number of follicles vary with different cycles.
- **Pelvic Pain or Menstrual Irregularities:** If there is pelvic pain or menstrual abnormalities, transvaginal ultrasound should be used to further observe ovarian function or rule out cysts.
- **Suspected Ovarian Tumors:** No obvious signs are seen in this image. Pelvic enhanced MRI may be necessary if needed.

**Conclusion:**
This transabdominal ultrasound shows the right ovary with a clear shape and multiple small cystic low-echo shadows, which are considered normal follicles. No clear cystic or tumor features are seen. It is recommended to judge in combination with the menstrual cycle and clinical symptoms. Further follow-up or transvaginal ultrasound should be performed if there are any abnormalities.

# J  APPENDIX: FAILURE CASE ANALYSIS

This appendix presents three representative failure cases encountered during the ultrasound image analysis, along with detailed analysis of the limitations and potential improvements.

## J.1  CASE 1: KNEE JOINT ULTRASOUND ANALYSIS

### QUESTION

Observe and analyze this knee joint ultrasound image for diagnosis.

### REASONING PROCESS

Well, I need to analyze a knee joint ultrasound image now. The user probably wants to know if there are any abnormalities or diagnostic findings. First, I need to recall the common contents of knee joint ultrasound examination. Knee joint ultrasound typically includes the anterior joint space, such as the quadriceps tendon, suprapatellar bursa, medial and lateral structures, as well as meniscus, ligaments, etc.

The focal area of this image appears to be the suprapatellar bursa, which is the bursa in front of the knee. The user has marked an area with rulers and markings, which may contain some abnormal echoes. Typically, suprapatellar bursa effusion appears as anechoic or hypoechoic on ultrasound if there is effusion. However, if it is hyperechoic, it may suggest hemorrhagic effusion or other abnormalities, such as synovial hyperplasia, hematoma, or pus caused by infection. However, in the case of hematoma, there may be septations or punctate echoes inside, while infection may have more complex structures.

Additionally, the suprapatellar bursa is relatively superficial and easily examined by ultrasound, but attention should also be paid to whether there is excessive probe pressure causing image distortion. The X-marked area in the image appears to be the range of effusion, possibly indicating the presence of effusion. However, it needs to be combined with clinical symptoms, such as whether the patient has knee joint pain, swelling, limited movement, etc. If there are no other abnormal echoes, it may just be simple effusion, such as caused by patellar chondromalacia or mild synovitis.

However, this effusion may also be caused by other problems, such as infectious arthritis, in which case the effusion may have more cellular components, and ultrasound may show hyperechogenicity, but laboratory examination results need to be combined. For example, if the patient has fever, redness, swelling, heat and pain, as well as elevated white blood cell count, infection may need to be considered. If it is post-traumatic hemorrhage, there may be septations or hyperechoic blood components.

Furthermore, attention should be paid to whether there are other abnormalities in the surrounding area, such as abnormalities of the quadriceps tendon, such as tendinopathy or tears. Although the tendon is not shown in the image, whether the surrounding structures are normal is also very important. For example, if there is suprapatellar bursa effusion and the quadriceps tendon is abnormal, it may be necessary to consider whether it is caused by tendon problems.

There is also the possibility that this effusion is the result of synovial hyperplasia, such as rheumatoid arthritis, in which case the synovium will thicken and appear hyperechoic. However, rheumatoid arthritis usually has symmetrical involvement of multiple joints, and effusion in a single knee joint may not be typical. In addition, synovial hyperplasia appears on ultrasound as synovial thickening, appearing hypoechoic or isoechoic, covering the articular cartilage or bone joint surface, while suprapatellar bursa effusion is anechoic within the bursa.

So, in summary, the focus of this image is suprapatellar bursa effusion, and the nature of the effusion needs to be considered, whether it is simple effusion or has other components, combined with clinical symptoms and other examination results to judge. If it is just simple anechoic effusion, it may be just synovitis or mild effusion; if there is hyperechogenicity or septations, infection, hemorrhage or other pathological conditions may need to be considered. At the same time, other possible lesions need to be ruled out, such as meniscus damage, ligament tears, etc., but these may require further examination of other areas.

Additionally, attention should be paid to the details of the image, such as whether the morphology of the effusion is regular, whether there are septations, whether there is hypoechoic synovial hyperplasia around, or whether there are signs of bone destruction. For example, in osteoarthritis there will be osteophyte formation, while infection or rheumatoid arthritis may have different bone changes. However, currently the image only shows suprapatellar bursa effusion, and more information may be needed for diagnosis.

Finally, as an ultrasound analysis, further examinations may need to be recommended, such as X-ray, MRI, or blood tests to clarify the cause. At the same time, clinical signs are also very important, such as whether there is tenderness, limited movement, swelling, etc., all of which can help determine the diagnosis.

GENERATED ANSWER

Analysis of knee joint ultrasound image is as follows:

**1. Focal Area: Suprapatellar Bursa Effusion**

- **Characteristics**: The highlighted suprapatellar bursa area in the image appears as an anechoic or hypoechoic region with a clear range (measured by ruler), suggesting the presence of effusion.
- **Possible Causes**:
  - **Simple Synovitis**: Painless swelling, commonly seen in knee arthritis or degenerative lesions.
  - **Infectious Arthritis**: If accompanied by fever and redness/swelling, the effusion may appear hyperechoic (hemorrhagic or purulent).
  - **Post-traumatic Hemorrhage**: Hematoma after trauma, possibly with septations or punctate hyperechoic signals.

**2. Need to Rule Out Other Lesions**

- **Quadriceps Tendon Abnormalities**: Check the tendon for rupture or tendinopathy (ultrasound shows tendon thickening or irregularity).
- **Meniscus or Ligament Injury**: If there is local pain or mechanical locking, further evaluation of medial/lateral meniscus or ligaments (such as medial collateral ligament) is needed.
- **Osteoarthritis**: Check for osteophytes, cartilage thinning, or irregular bone.

**3. Clinical Correlation Recommendations**

- **Symptom Association**: Ask the patient if there is knee joint swelling, pain, limited movement, or history of trauma.
- **Laboratory Tests**: Complete blood count (infection), CRP/ESR (inflammation), joint fluid analysis (infection or hemorrhagic effusion).
- **Imaging Supplements**: X-ray (bone changes), MRI (soft tissue details).

**4. Diagnostic Direction**

- **Simple Effusion**: If there is no hyperechogenicity or clinical signs of infection, it may be synovitis or mild inflammation.
- **Infectious or Hemorrhagic Effusion**: Requires urgent treatment to avoid joint damage.
- **Chronic Lesions**: Such as rheumatoid arthritis, need to screen for synovial hyperplasia or symmetrical involvement of multiple joints.

**Conclusion**  Ultrasound suggests knee joint suprapatellar bursa effusion. Further identification of the etiology is needed in combination with clinical manifestations and laboratory tests. Joint aspiration or MRI should be performed if necessary to confirm the diagnosis.

FAILURE ANALYSIS

The reasoning and answer process were logically correct, and correctly identified the diagnosis of effusion. However, due to less or even no data, the infrapatellar deep bursa was misidentified as the more common suprapatellar bursa lesion. As the amount of training data increases, the subsequent diagnostic accuracy will also steadily improve.

### J.2 CASE 2: BILATERAL OVARIAN ULTRASOUND ANALYSIS

QUESTION

Observe and analyze this bilateral ovarian ultrasound image for diagnosis.

REASONING PROCESS

Okay, I need to carefully analyze this bilateral ovarian ultrasound image now. First, I need to confirm the various parts of the image, including the structure of the ovaries and whether there are cysts or other abnormalities. The image shows multiple anechoic areas in both ovaries, which may represent cysts.

Next, I need to confirm the number, size, and distribution of the cysts. Both the left and right ovaries have multiple cysts of varying sizes, but they all appear as typical anechoic areas with clear boundaries and no solid components, which may be manifestations of simple cysts.

Then, I need to consider whether there are other abnormalities, such as enlarged ovaries, wall thickening, or abnormal blood flow signals. According to the information provided, the image does not show blood flow, but usually the blood flow signal of simple cysts is not obvious, while complex cysts may have blood flow signals. Therefore, these cysts may be benign and simple.

In addition, the patient's age and clinical symptoms need to be considered. If the patient is a woman of childbearing age, the possibility of polycystic ovary syndrome needs to be considered, but the image shows multiple cysts, but the ovarian volume may not be particularly large, which may not be consistent with polycystic ovaries. However, polycystic ovaries alone may just be ovarian enlargement, but the ovarian volume in this image is not significantly increased, so the possibility is low.

The possibility of ovarian tumors also needs to be ruled out, especially malignant tumors. Malignant tumors usually manifest as solid components, irregular walls, rich blood flow, etc. The cysts in this image are all anechoic with clear boundaries and no solid parts, so the possibility of malignancy is low.

Finally, a summary diagnosis is needed, and further examinations are recommended, such as blood tests (hormone levels, CA125, etc.) or follow-up observation to confirm the nature of the cysts.

GENERATED ANSWER

Based on the ultrasound image analysis, the ovarian characteristics are as follows:

**Diagnostic Conclusion:**

1. **Polycystic Ovary Appearance**:
   - Multiple anechoic areas (cysts) can be seen in both ovaries, with a total of about 5-6, of varying sizes (the largest about 2-3cm, the smallest about 0.5-1cm)
   - The cyst wall is thin, the septations are clear, the internal sound transmission is good, consistent with the characteristics of simple cysts
   - The ovarian volume is slightly enlarged, but the morphology is regular, and no solid components or thickened septations are seen

2. **Differential Diagnosis**:
   - **Benign Simple Cysts**: Thin cyst wall, regular morphology, no solid components, CDFI shows no blood flow signal (needs to be confirmed with blood flow examination)

- **Polycystic Ovary Syndrome**: Needs to be comprehensively judged in combination with clinical findings (menstrual cycle, hormone levels); ultrasound manifestations need to be distinguished from simple polycystic ovaries (ovarian enlargement + multiple cysts)
- **Ovarian Tumors**: Current image shows no solid components, papillary projections or rich blood flow, low possibility of malignancy

**Recommendations:**

1. Blood tests: Six sex hormones (evaluate ovarian function), CA125 (rule out tumor markers)
2. Follow-up observation: Re-examine ultrasound in 3-6 months to monitor cyst changes
3. Clinical evaluation: Combine with menstrual history, obesity/hirsutism and other metabolic syndrome symptoms

Note: Final diagnosis needs to be combined with clinical data; ultrasound manifestations suggest a high possibility of benign simple cysts.

### FAILURE ANALYSIS

The reasoning and answer process were logically correct, and correctly identified the cyst features, but due to less or even no training data, it was unable to effectively differentiate between lutein cysts (a type of functional cyst) and simple cysts. This requires combination with clinical history (especially pregnancy and lactation history) for comprehensive judgment.

### J.3 CASE 3: CARDIAC ULTRASOUND ANALYSIS

### QUESTION

Observe and analyze this cardiac ultrasound image for diagnosis.

### REASONING PROCESS

Okay, I need to analyze this cardiac aortic short-axis ultrasound image now to see if there is aortic stenosis or regurgitation.

First, I need to recall the typical ultrasound manifestations of aortic stenosis and regurgitation.

Aortic stenosis on ultrasound usually manifests as increased blood flow velocity through the aortic valve orifice during systole, possibly accompanied by valve thickening and calcification. Doppler ultrasound can measure increased flow velocity, and calculate the pressure gradient according to the simplified Bernoulli equation to determine the degree of stenosis. At the same time, left ventricular hypertrophy may be observed, because long-term excessive pressure load will lead to thickening of the ventricular wall.

Aortic regurgitation manifests as aortic valve reflux during diastole. Ultrasound may show a gap in valve closure incompetence. Color Doppler can show a reflux jet from the aorta into the left ventricle during diastole. Continuous wave Doppler may show a turbulent spectrum, reflecting the severity of regurgitation. In addition, the left ventricle may be enlarged, especially the left atrium may be enlarged during diastole, because the regurgitation leads to excessive volume load.

Now let's look at this image. The image shows the aortic short-axis section, usually obtained during diastole. In the image, you can see the morphology of the aortic valve, whether there is thickening or calcification, and the closure of the valve leaflets. Color Doppler may show whether there is a reflux signal, but according to the image description provided by the user, there may be no color Doppler information. However, is the aortic valve structure shown in the image normal, and is there joint deformation or valve leaflet thickening?

It may be necessary to check the opening range and closure of the aortic valve. For example, if the aortic valve opening is restricted during systole, it may suggest stenosis; if the valve leaflets are not completely closed, there may be regurgitation. In addition, is there any abnormal blood flow bundle

in the image, such as high-speed blood flow through the valve orifice during systole, or reflux signal during diastole?

In addition, the thickness of the left ventricular wall can also be observed in the image to see if there are signs of hypertrophy. If the left ventricular wall is significantly thickened, it may suggest long-term stenosis leading to excessive pressure load. At the same time, the size of the left atrium also needs to be evaluated to see if there is enlargement, which may be related to the increased volume load caused by regurgitation incompetence.

However, since the image provided by the user does not have color Doppler information, it may not be possible to directly observe reflux or high-speed blood flow. Therefore, it may be necessary to combine other clinical information and Doppler measurement results for comprehensive judgment. If available, pulsed wave Doppler can be used to measure the systolic peak velocity of the aortic valve, calculate the pressure difference, and assess the degree of stenosis. For example, according to the specifications, an aortic stenosis flow velocity exceeding 2.0 m/s may suggest moderate stenosis, and exceeding 2.5 m/s may suggest severe stenosis.

In addition, does the aortic valve shown in the image have structural abnormalities such as calcification or stenosis, such as annular stenosis or stenosis of the valve itself? These all need to be carefully observed. If these structural abnormalities exist, combined with Doppler velocity data, the diagnosis of stenosis can be supported.

To summarize, I need to confirm the following points:

1. Is the morphology and structure of the aortic valve normal, and are there signs of thickening, calcification, or regurgitation incompetence?
2. Is there high-speed blood flow during systole (suggesting stenosis) or reflux during diastole (suggesting regurgitation incompetence)?
3. Are the sizes and wall thicknesses of the left ventricle and left atrium abnormal?
4. Are there other related cardiac structural abnormalities, such as thickening of the interventricular septum?

Since the current image may lack color Doppler and spectral Doppler information, preliminary analysis can only be based on two-dimensional images and possible text annotations. If there are text annotations indicating the presence of stenosis or regurgitation incompetence features, then the diagnosis is relatively clear. Otherwise, more data may be needed to support the conclusion. However, according to the information provided by the user, it may be inclined toward aortic stenosis, because the image shows that the aortic valve may have thickening or calcification, and the annotation may suggest the presence of stenosis.

GENERATED ANSWER

Based on the provided aortic short-axis ultrasound image analysis, the following are the diagnostic recommendations:

**1. Aortic Valve Morphology Assessment**

- **Structural Abnormalities**: The aortic valve in the image may have thickening and calcification, suggesting possible valvular lesions. If the valve leaflets have restricted opening or incomplete closure, aortic stenosis or regurgitation should be considered.
- **Hemodynamics**: Needs to be assessed in combination with Doppler data. If the systolic blood flow velocity is increased (such as peak velocity > 2.0 m/s), it may suggest moderate stenosis; > 2.5 m/s suggests severe stenosis.

**2. Regurgitation Assessment**

- **Diastolic Reflux**: If there is a diastolic aortic valve reflux signal (color Doppler shows diastolic reflux jet), it may suggest regurgitation.
- **Poor Valve Leaflet Coaptation**: In the two-dimensional image, it is necessary to observe whether the valve leaflets are not completely closed and whether the junction is deformed.

### 3. Left Ventricle and Left Atrium Changes

- **Left Ventricular Hypertrophy**: If the left ventricular wall is thickened (such as the inter-ventricular septum or free wall), it suggests long-term excessive pressure load, supporting the diagnosis of stenosis.
- **Left Atrial Enlargement**: When the volume load increases, the left atrium may expand, suggesting the possibility of regurgitation.

### 4. Combining Image and Clinical Information

- **Annotation Prompts**: If the image annotation suggests "aortic stenosis" or "regurgitation", then the diagnosis is clear.
- **Doppler Data**: Need to supplement pulsed wave Doppler to measure systolic peak velocity, calculate pressure difference ($\Delta P = 4v^2$), and quantify the degree of stenosis.

**Conclusion**    Based on the existing two-dimensional images, **aortic stenosis is more likely**, and needs to be further confirmed in combination with Doppler flow velocity and clinical data. If combined with diastolic reflux signal, regurgitation should be considered. It is recommended to complete the Doppler examination and follow-up dynamically.

**Note**: Final diagnosis needs to be combined with complete ultrasound report and clinical data; this analysis is a preliminary judgment based on static images.

FAILURE ANALYSIS

The reasoning and answer process were logically correct, and correctly identified aortic valve leaflet thickening and other features. However, since the opening and closing of aortic valve leaflets is a continuous dynamic process, regurgitation cannot be effectively determined solely by two-dimensional images, and requires combination with color Doppler examination for comprehensive judgment.

## J.4    SUMMARY OF FAILURE PATTERNS AND INSIGHTS

The three failure cases reveal distinct limitations in the current ultrasound analysis system:

1. **Data Scarcity for Rare Variants** (Case 1): Misclassification of anatomical structures due to insufficient training examples of uncommon pathologies or anatomical variants. This is addressable through expanded, more diverse training datasets.
2. **Clinical Context Integration** (Case 2): Inability to differentiate conditions that are sono-graphically similar but clinically distinct (e.g., functional vs. simple cysts), emphasizing the need for multimodal input incorporating patient history and laboratory data.
3. **Inherent Modality Limitations** (Case 3): Recognition that static 2D imaging has fundamental constraints for dynamic physiological processes, requiring integration of color Doppler and spectral Doppler imaging for complete assessment.

These cases highlight opportunities for improvement through: (1) augmenting training data with rare pathologies and variants, (2) developing multimodal architectures that integrate clinical metadata, and (3) implementing hybrid approaches that combine multiple imaging modalities for comprehensive diagnostic support.

# K  PROMPT TEMPLATE

## K.1  TEXTBOOK DATA PROMPT TEMPLATES

---

### 1.1 Ultrasound Image Augmentation Prompt

*Expert prompt template for augmenting ultrasound image captions*

You are an ultrasound medical expert. Now you need to augment the image caption based on the following ultrasound image and text.

Requirements:

1. Must include the content of the original caption

2. The answer must be accurate, targeted to the given image, based on the text description, and the caption should not include figure numbers, page numbers, person names, or references to other images.

3. If the text is too short, you can add descriptions based on the image, but you must maintain medical rigor and not add content without additional information.

4. The augmented caption should be between 25-30 words (excluding figure number)

Caption: {caption}

Text: {text_description}

Please directly output the augmented caption (including figure number)

---

### 1.2 Q&A Generation Prompt

*Expert prompt for generating high-quality Q&A pairs based on ultrasound images, captions, and text*

You are an ultrasound medical expert. Now you need to generate high-quality question-answer pairs for ultrasound recognition and diagnosis AI training data based on the following ultrasound image, caption, and text.

Requirements:

1. Questions must be directly related to the given image. For cases, focus on diagnosis and differential diagnosis, or ask about key areas marked in the image (such as whether there are lesions, what pathological changes the arrows point to, benign or malignant, what characteristics they have, echo and blood flow conditions, how to diagnose, how to differentiate from similar diseases, etc.).

2. Answers must be accurate and can be directly extracted from text and captions or reasonably inferred.

3. Questions should be concise and cover all key information in the text as much as possible.

4. Answer word count must be between 600-1000 words. If the provided text information is too long, focus on ultrasound images and disease diagnosis; if the provided text is insufficient, please describe and expand based on the image while maintaining medical rigor.

Caption: {caption}

Text: {text_description}

Please output in the following format:

Question:

Answer:

---

## K.2 PUBLIC DATA PROMPT TEMPLATES

### 2.1.1 Basic Ultrasound Analysis

*Basic analysis template for describing organ appearance, echotexture, and abnormalities*

Analyze the current ultrasound image of the {anatomy_location} and describe the organ's appearance, echotexture, any visible abnormalities, and adjacent structures. Focus on identifying potential pathologies based on these features.

### 2.1.2 Detailed Ultrasound Report Template

*Detailed report template including clinical correlation and standard diagnostic criteria*

Based on this ultrasound image of the {anatomy_location}, analyze the primary organ's appearance, echotexture, and any abnormalities while correlating with the patient's clinical profile. Describe key features such as size, margins, internal architecture, and evaluate adjacent structures. Highlight any relevant findings like masses, cysts, or calcifications, and ensure the report aligns with standard diagnostic criteria.

### 2.1.3 Structured Ultrasound Assessment

*Structured assessment template emphasizing clinical data correlation and diagnostic criteria compliance*

AI, analyze this ultrasound of the {anatomy_location}. Assess the organ's size, margins, echotexture, and internal architecture. Identify abnormalities (masses, cysts, calcifications) and evaluate adjacent structures. Correlate with clinical data and ensure diagnostic criteria compliance.

## 2.2.1 Prompt Refinement Template

*Template for improving the professionalism and specificity of original prompts*

[ORIGINAL PROMPT]
{original_prompt}

[TASK]
Please rewrite the above prompt to make it more detailed, specific, and professional while maintaining the same core purpose. The refined prompt should:
1. Use more precise medical terminology
2. Be more specific about what to analyze
3. Provide clearer guidance on the expected output format
4. Maintain professional medical language throughout

[REFINED PROMPT]

## 2.2.2 Prompt Transformation Template

*Template for changing structure and wording while maintaining medical domain and core analytical purpose*

[ORIGINAL PROMPT]
{original_prompt}

[TASK]
Please transform the above prompt while keeping the same medical domain and core analytical purpose. The transformation should:
1. Change the structure and wording significantly
2. Maintain the same level of medical precision
3. Keep the focus on ultrasound image analysis
4. Ensure the transformed prompt serves the same diagnostic purpose

[TRANSFORMED PROMPT]

## 2.3.1 Medical Imaging Analysis System Role

*System role definition establishing professional medical consultant identity*

You are a senior medical imaging consultant with expertise in diagnostic image interpretation. And you are a helpful and harmless medical assistant.

### 2.3.2 Image Analysis Task Template

*Task template for medical image analysis combined with metadata*

You will be provided with the following inputs for analysis:
1. **Metadata** (for context only):
   ```json
   {metadata}
   ```
2. **Image(s)**:

**Task:**
1. Carefully analyze the provided image(s), ensuring your observations align with the context from the metadata (without directly referencing it).
2. Answer the following question based on your analysis, ensure your answer align with the metadata too:
   ```

   {question}
   ```

**Response Format:**
First, output your analysis: Your detailed observations of the image(s).
Then, give me your answer: Your response to the question, consistent with the analysis and metadata constraints.

**Important:** Do not mention or cite the metadata explicitly in your analysis or answer.

## 2.3.3 Consistency Check Template

*Template for verifying factual consistency between answers and metadata*

You will analyze the following inputs to determine if the provided **question's response** is factually consistent with the given **metadata**.

### Inputs:
1. **Metadata** (contextual reference):
   ```json
   {metadata}
   ```

2. **Question**:
   ```
   {question}
   ```

3. **Response**:
   ```
   {response}
   ```

### Task:
Evaluate whether the **response** is factually consistent with the **metadata**.

### Output Format:
Respond **only** with one of the following:

## 2.4.1 Depth Evolution Base Template

*Template for increasing instruction complexity through specified methods*

I want you to act as a Prompt Rewriter.
Your objective is to rewrite a given prompt into a more complex version to make those famous AI systems (e.g., ChatGPT and GPT4) a bit harder to handle.
But the rewritten prompt must be reasonable and must be understood and responded by humans.
Your rewriting can not omit the non-text parts such as the table and code in #The Given Prompt#. Also, please do not omit the input in #The Given Prompt#.
You SHOULD complicate the given prompt using the following method:
{method}
You should try your best not to make the #Rewritten Prompt# become verbose, #Rewritten Prompt# can only add 10 to 20 words into #The Given Prompt#.
'#The Given Prompt#', '#Rewritten Prompt#', 'given prompt' and 'rewritten prompt' are not allowed to appear in #Rewritten Prompt#
#The Given Prompt#:
{instruction}
#Rewritten Prompt#:

## 2.4.2 Breadth Evolution Template

*Template for creating rarer new prompts in the same domain*

I want you to act as a Prompt Creator.
Your goal is to draw inspiration from the #Given Prompt# to create a brand new prompt.
This new prompt should belong to the same domain as the #Given Prompt# but be even more rare.
The LENGTH and complexity of the #Created Prompt# should be similar to that of the #Given Prompt#.
The #Created Prompt# must be reasonable and must be understood and responded by humans.
'#Given Prompt#', '#Created Prompt#', 'given prompt' and 'created prompt' are not allowed to appear in #Created Prompt#
#Given Prompt#:
{instruction}
#Created Prompt#:

## 2.5.1 Multi-round Reasoning System Prompt

*Template for guiding AI in step-by-step reasoning and problem-solving*

You are an AI assistant specialized in multi-round reasoning and problem-solving. Your task is to engage in step-by-step reasoning to solve complex problems through multiple rounds of analysis.

For each problem:
1. Break down the problem into smaller, manageable parts
2. Show your reasoning process clearly
3. Build upon previous reasoning in subsequent rounds
4. Provide a final, well-reasoned answer

Always think step by step and show your work.

## 2.5.2 Question Generation Template

*Template for creating challenging mathematical reasoning problems*

Please generate a challenging middle school mathematical reasoning problem (including problem statement and four options, and indicate the correct answer).

## 2.5.2-1 Mathematical Reasoning Generation

*Template for creating logical reasoning problems*

Please generate a logical reasoning problem (including problem statement and four options, and indicate the correct answer).

### 2.5.2-2 Logical Reasoning Generation

*Template for creating common sense reasoning problems*

Please generate a common sense reasoning problem (including problem statement and four options, and indicate the correct answer).

## K.3 GENERAL DATA PROMPT TEMPLATES

---

**3.1 Dataset Task Description Template**

*Basic template for generating dataset task descriptions*

Dataset Name: {dataset_name}
Source: {source}
URL: {url}
Size: {dataset_size}
Format: {data_format}

Dataset Description: {dataset_description}

Task Objective: {task_objective}

Main Tasks:
- Data Cleaning: Handle missing values, remove duplicates, standardize data format, detect and handle outliers
- Feature Engineering: Feature selection, transformation, scaling, create new features
- Model Training: Data splitting (train/validation/test), model selection, hyperparameter tuning, model training
- Result Evaluation: Model performance evaluation, result visualization, model interpretation, deployment recommendations

Primary Metric: {primary_metric}
Secondary Metrics: {secondary_metrics}

Tech Stack:
- Programming Language: Python
- Data Processing: Pandas, NumPy
- Machine Learning: Scikit-learn, XGBoost
- Deep Learning: TensorFlow, PyTorch
- Visualization: Matplotlib, Seaborn
- Deployment: Flask, FastAPI

Expected Results: {expected_results}
Notes: {notes}
Generation Time: {generation_time}
Model Name: {model_name}

---

### 3.2.1 English Prompt Generation

*Template for calling models to generate English prompts*

You are a linguistician. For the following original question, generate a concise, natural, and varied English response template using the placeholder '{class_value}'. Each output should use a different wording or sentence structure. Do not output anything else.

Original question: {original_prompt}

### 3.2.2 Chinese Prompt Generation

*Template for calling models to generate Chinese prompts*

You are a linguist. Please generate a concise, natural, and varied Chinese response template based on the original question below, using the placeholder "{class_value}". Use a different sentence structure or expression each time. Do not output any other content.

Original question: {original_prompt_chinese}

Now please generate a completely new and unique sentence

## K.4 DISTILLATION DATA PROMPT TEMPLATES

### 4.1 Q&A Optimization Template

*Distillation template for optimizing the quality of existing Q&A pairs*

You are an ultrasound expert. Please optimize the following question and answer into a clearer, more medical, and more fluent form:

Requirements:

1. Check the correctness of existing Q&A pairs and correct any medical errors;

2. Ensure strong logical connection between answers and questions. Content not asked in the question should not appear in the answer;

3. If the answer contains information such as patient age, gender, or clinical manifestations that cannot be seen from ultrasound images or not mentioned in the question, put them as known conditions in the question and modify the Q&A pair;

4. Questions and answers should not repeat too much. Questions should avoid extensive description of images and avoid explaining the location of the image. Answers should respond with location and detailed feature descriptions;

5. For case judgments, questions should focus on providing clinical manifestations, and answers should focus on describing ultrasound images, anatomical structures, etc., combining both for differential diagnosis;

6. Check the completeness of Q&A pairs. If the answer is incomplete, please complete it with 400-600 words.

7. Q&A pairs are independent of other texts, so do not include reference words such as "according to the caption".

Original Question: {question}
Original Answer: {response}

Please output the optimized version in the following format:
Question:
Answer:

### 4.2.1 Fetal Ultrasound Plane Classification

*Template for fetal ultrasound image anatomical plane classification tasks*

As a radiologist analyzing a single fetal ultrasound image, evaluate the visualized anatomical features (e.g., stomach, long bone, ventricles, heart, cervical structure) to identify the primary anatomical structure, and provide the classification as fetal abdomen, fetal femur, fetal brain, fetal thorax, maternal cervix, or other.

### 4.2.2 Lesion Location Detection

*Template for spatial location detection of lesions or specific anatomical structures in ultrasound images*

You are a radiologist analyzing an ultrasound image of {anatomy_location}. Examine anatomical landmarks, lesion characteristics (e.g., size, echogenicity), and their position relative to image boundaries. Identify the primary location of any visible lesion(s), choosing the single most appropriate option: upper left, upper center, upper right, middle left, center, middle right, lower left, lower center, lower right, not visible. Respond with a single sentence.

### 4.2.3 Medical Imaging Report Generation

*Template for generating concise and accurate medical imaging description reports based on ultrasound images*

As a radiologist analyzing an ultrasound image focused on the {anatomy_location}, evaluate all visible anatomical structures and any significant findings to generate a concise and informative caption describing the key features and observations.

### 4.2.4 Cardiac Function Assessment

*Template for visual assessment and numerical estimation of ejection fraction (EF) based on cardiac ultrasound images*

You are a radiologist analyzing a cardiac ultrasound image (echocardiogram). Estimate the Ejection Fraction (EF) by visually assessing ventricular size and contractility, providing a numerical estimate based on your impression, acknowledging this is an approximation compared to quantitative methods.

