# OpenReview forum: "Dolphin: A multimodal large language model for Ultrasound Understanding"
_ICLR.cc/2026/Conference — ICLR 2026 Conference Desk Rejected Submission_

### Official Review · Reviewer_HA3u · 2025-10-29

**Soundness:** 2
**Presentation:** 2
**Contribution:** 2
**Rating:** 4
**Confidence:** 5

**Summary:**

This paper presents Dolphin, a VLM for ultrasound images understanding. The authors compile and standardize a vast ultrasound dataset for reinforcement learning purposes. The models, Dolphin-7B and Dolphin-72B, achieve state-of-the-art performance on the ultrasound benchmark, U2-Bench.

**Strengths:**

- I recognize the engineering effort involved in compiling and processing vast ultrasound datasets. It would be very helpful for the community if the authors could release standardized data publicly.

- The Dolphin achieves very promising performance on the ultrasound benchmark, while preserving good performance on general medical benchmarks.

**Weaknesses:**

- One of the main findings in the paper: "cross domain reasoning universality" has been already systematically explored by previous work [1].

- Lacking methodology novelty: the training paradigm of SFT with distilled Chain-of-Thought data, followed by GRPO-based RL, is widely used by a lot of works. The proposed UARPO makes no difference to me from general GRPO.

- The core contribution of this paper would be the processing and standardizing of vast public ultrasound data. But the key data processing details, such as source data and task distribution, are missed in the current paper.

[1] Liu, Qianchu, et al. "X-reasoner: Towards generalizable reasoning across modalities and domains." arXiv preprint arXiv:2505.03981 (2025).

**Questions:**

- Can you provide more data processing details? Where does your source data come from, and what is the distribution across each ultrasound task? Have you checked the quality of this public data? Will this processed data be made publicly accessible?

- As you mentioned, "Dolphin has been successfully deployed on real-world ultrasound devices," can you provide more details about how the Dolphin is involved in the clinical process and the feedback from ultrasound experts when using the Dolphin?

---

> ### Author Response · Authors · 2025-11-20
>
> Thank you for your valuable feedback on our paper!
>
> W1: Regarding your comment that existing work [1] has conducted a systematic exploration of "cross-domain reasoning generalizability" but has not been validated on real vertical-domain tasks—additionally, that work uses purely reasoning data during training without incorporating general data—our approach differs in that we enable models to emerge reasoning capabilities based on injected medical knowledge, whereas work [1] primarily leverages reasoning data to stimulate the model’s inherent medical reasoning. Moreover, we have conducted relevant experiments showing that under the u2-bench evaluation framework, solely adding reasoning data as in work [1] leads to a drop in u2-score. Our work not only builds a general large model for the ultrasound domain but also explores and provides a feasible solution for how visual reasoning large models should be developed in the medical field.
>
> W2: Our adopted UARPO method is identical to GRPO in terms of training methodology, as already stated in the paper. The difference lies in our modeling all ultrasound tasks into two URA reward functions and validating their effectiveness, achieving improvements across diagnosis, measurement, and detection tasks. This can serve as a reference for other medical large models, avoiding the need to design distinct reward functions for each task type.
>
> W3&Q1: Your suggestions regarding data are highly valuable. We have provided detailed补充 descriptions in Section 3.1 of the paper and supplemented the full list of public datasets along with their licenses in the appendix. Each dataset has undergone manual verification, with each team member responsible for 3–4 datasets, and final data quality confirmed by human medical experts. We have released all used public datasets, disclosed the specific format of DUDP in the appendix, and provided prompt templates for synthetic data to facilitate reproducibility.
>
> Q2: Currently, during the diagnostic process, Dolphin assists physicians in analyzing ultrasound images after they complete the image acquisition. Through multi-image input and open-ended questioning, it helps doctors determine whether the images meet quality control standards, assess the risk of hidden lesions, and provide further treatment recommendations. Simultaneously, it assists physicians in directly generating ultrasound reports, thereby aiding primary care doctors in diagnosing conditions. Experts currently using Dolphin believe that it can provide them with appropriate diagnostic suggestions and plan to develop a series of large models specialized in various ultrasound anatomical regions based on Dolphin. They hope that Dolphin's capabilities in measurement accuracy and dynamic response will be further enhanced in the future, and we will continue to conduct corresponding research.
> Additionally, we have performed a series of related experiments. In Appendix J, we present a comparison between our model's output and that of GPT-5, with medical experts annotating the reasonable and unreasonable parts of the model's responses. The results show that our model's analytical accuracy surpasses that of GPT in most anatomical regions. Furthermore, in Appendix B, we have supplemented with a clinical expert evaluation study, which is divided into two main aspects. The first is clinical rationality evaluation: we randomly selected 30 ultrasound images and obtained diagnostic results from both Dolphin and GPT-5. Two ultrasound specialists with over five years of experience then conducted a clinical rationality assessment, with scores ranging from 1 to 5 (5 being the highest). The results indicate that Dolphin (4 ± 0.276) significantly outperforms GPT-5 (3.73 ± 0.890) in both overall clinical relevance and variance. Moreover, Dolphin's lowest score was 3, while GPT-5's lowest score was 1, demonstrating that Dolphin's diagnostic performance on ultrasound images is more reliable.
> Additionally, we conducted a comparative experiment between Dolphin and human physicians. Specifically, we selected a subset of u2-bench to create a web-based ultrasound question bank, which was then answered by eight human experts. The results show that Dolphin's accuracy of 73.5% is higher than the human experts' average of 60.63%. In the VAR task, Dolphin achieved a result of 60.2%, surpassing the human experts' average of 52.08%, highlighting Dolphin's exceptional capability in understanding ultrasound images.

---

> > ### Comment · Reviewer_HA3u · 2025-11-25
> > **Response for the Rebuttal**
> >
> > Thanks the author for providing the rebuttal. Thanks you for providing me the details about the data recipe. However, I still have the following concerns:
> >
> > 1. You mentioned in rebuttal that "we enable models to emerge reasoning capabilities based on injected medical knowledge". To me this is also a common practice in training of medical specific model, which is firstly introduced by HuatuoGPT-o1. These findings still lack valuable insights.
> >
> > 2. "The difference lies in our modeling all ultrasound tasks into two URA reward functions", combination of format reward and outcome reward is still a very common setting for most of RL post-training.
> >
> > Overall, I suggest the authors to revise the claim about the UARPO algorithm and reasoning transfer. Instead, I suggest you to provide more results with ultrasound domain insight. For example, what is the major difference between ultrasound tasks and general medical domain? Are there any unique challenge for this domain comparing to other medical tasks? How do you incorporate these domain specific insights into your data recipe? These analysis would benefit the soundness and novelty of this paper.

---

> > > ### Author Response · Authors · 2025-11-26
> > >
> > > Thank you for your valuable suggestions!
> > >
> > > 1. Regarding the HuatuoGPT-o1 you mentioned, it is indeed an excellent work that constructs verifiable Chain-of-Thought (CoT) and successfully enhances medical reasoning capabilities. However, there is a fundamental difference between HuatuoGPT-o1 and our work: Dolphin is a medical Vision-Language Model (VLM), whereas HuatuoGPT-o1 is a purely text-based model. For text-based training data, it is possible to construct reasonable CoT and validate it through another LLM or other methods. However, this approach is not feasible for medical VLMs because images are inherently part of the problem, and arbitrarily constructing CoT would inevitably introduce hallucinations. Therefore, our training approach essentially provides a new direction for medical VLM training—using deterministic image QA pairs to improve the model's foundational capabilities, leveraging reasoning data to stimulate the model's thinking ability, and achieving emergent cognition of ultrasound images through an appropriate mixing ratio. We have conducted relevant experiments, and if only deterministic image QA pairs are used for training, the model cannot provide diagnostic evidence or articulate its reasoning process.
> > >
> > > 2. Ultrasound imaging exhibits unique complexity and diversity in the field of medical imaging, with significant differences compared to other modalities such as MRI and CT. MRI and CT scans are typically based on standardized anatomical positioning protocols, with relatively fixed and procedural acquisition positions. In contrast, ultrasound examinations highly depend on the operator's technique and real-time judgment, covering multiple anatomical regions (e.g., heart, abdomen, thyroid, blood vessels) with distinct scanning protocols, diagnostic criteria, and clinical requirements for each region. Based on the clinical workflow characteristics of ultrasound examinations, we systematically abstract all tasks into four core categories: **View Identification**, **Disease Diagnosis**, **Lesion Localization**, and **Biometric Measurement**. To establish a unified evaluation framework, we designed each task in two forms, constructing the **Ultrasound-Aware Reward (UAR)**, which includes **structured multiple-choice questions** to assess the model's diagnostic capabilities and **open-ended questions** to evaluate its understanding of clinical semantics.
> > >
> > > We have added explanations about ultrasound imaging and the construction of UAR in Section 3.2.1 of the paper.
> > >
> > > Additionally, we have updated the evaluation results for MedLLaVA in the paper, including its performance on the U2-Bench and general datasets. We observed that MedLLaVA excels in lesion localization tasks on U2-Bench but underperforms in disease diagnosis and view identification tasks. On general benchmarks, MedLLaVA's performance is subpar, indicating room for improvement in instruction-following capabilities. We believe this is related to its data mixing ratio, training strategy, and lack of reasoning capabilities during training. The evaluation results have been added to **Table 1** and **Appendix D** for comparison.
> > >
> > > Thank you again for your insightful suggestions. We wish you a wonderful day!

---

> ### Author Response · Authors · 2025-11-27
>
> Dear Reviewer,
>
> Hello, we have supplemented the U2-bench results for Qilin-Med. During the evaluation, we noticed that Qilin-Med can only output responses in Chinese. Therefore, we additionally translated its outputs into English before conducting the assessment. Below are the test results:
> Qilin-Med-VL & 0.1952 & 0.0224 & 0.3004 & 0.0400 & 0.3227 & 0.2665 & 0.2800 & 0.3296 & 0.2997 & 29.7608 & 0.0001 & 3.2969 & 61.6669 & 1.5770 & 5.7315 & 66.5793 & 0.2608
>
> The U2-score is 0.2608. For comparison, Dolphin's U2-score is 0.58. In practice, we found that the response quality of Qilin-Med is not ideal, and it can only reply in Chinese, which is why we did not include it in the main text table.
>
> We look forward to your feedback. Wishing you a wonderful day!

---

> ### Author Response · Authors · 2025-12-01
> **Summary**
>
> **Summary of Reviewer Feedback and Our Revisions**
>
> **Initial Reviewer Concerns:**
> 1. Clarify the differences between **Dolphin** and **X-Reasoner**.
> 2. Explain the distinctions between **UARPO** and **GRPO**.
> 3. Supplement details on **data sources and processing**.
> 4. Describe **Dolphin’s role in clinical workflows and expert decision-making**.
>
> **After Initial Revisions**, the reviewers requested further clarification on:
> 1. The differences between **Dolphin** and **HuatuoGPT-o1**.
> 2. Adding **UARPO’s unique design for ultrasound tasks** in the main text.
>
> **Our Adjustments in Response:**
> 1. **Dolphin vs. HuatuoGPT-o1**:
>    - **Dolphin** is a **vision-language model (VLM)** with novel training methodology for **medical visual reasoning**, while **HuatuoGPT-o1** is a **pure text-based model**.
> 2. **UARPO for Ultrasound Tasks**:
>    - Added explanations in **Section 3.2.1** detailing how **UARPO** uniquely formulates ultrasound tasks using **two UAR reward functions**.
>
> **Responses to Original Reviewer Questions:**
> 1. **Dolphin vs. X-Reasoner**:
>    - Dolphin employs **hybrid reasoning training**, whereas **X-Reasoner** only trains on **chain-of-thought (CoT)**, representing a fundamental difference.
> 2. **UARPO vs. GRPO**:
>    - **UARPO** models **all ultrasound tasks** under **two unified UAR reward functions**, unlike GRPO.
> 3. **Data Processing & Sources**:
>    - Added **data processing details** (e.g., ratios per training phase) in the main text.
>    - Included a **supplementary table** listing **data sources and licenses**.
> 4. **Clinical Application of Dolphin**:
>    - Demonstrated Dolphin’s real-world deployment for:
>      - **Clinical decision support** (e.g., **ultrasound report generation**).
>      - **Direct integration with ultrasound devices**.
>    - Outlined future plans to develop **specialized models for different ultrasound subspecialties**.

---

### Official Review · Reviewer_R3QE · 2025-10-30

**Soundness:** 3
**Presentation:** 3
**Contribution:** 3
**Rating:** 6
**Confidence:** 4

**Summary:**

This paper presents Dolphin, a multimodal large language model (MLLM) for ultrasound understanding with cross-domain reasoning transfer. It introduces the DUDP protocol to standardize multimodal ultrasound data and adopts a three-stage training process (post-training, instruction tuning, reinforcement learning). Experiments show that Dolphin significantly outperforms existing state-of-the-art models in both the specialized ultrasound domain tasks.

**Strengths:**

- The paper is well-motivated and explores an important clinical application area, with a clear and well-structured presentation.
- It proposes a practical and comprehensive domain adaptation framework that can be readily applied to heterogeneous ultrasound data in real-world settings.
- The experimental evaluation is comprehensive, covering multiple datasets and metrics, and including medical expert validation to verifiy its reliability

**Weaknesses:**

- The reported 0.8:0.2 ratio in mixed reasoning is empirical and lacks systematic ablation.
- Despite achieving strong results on accuracy-driven, annotated benchmarks, the paper lacks a systematic evaluation of Dolphin's LLM text generation quality, specifically regarding its clinical relevance and expert human assessment for report and chat outputs.
- While the paper states Dolphin is capable of understanding videos. The paper lacks a detailed evaluation of Dolphin's performance on on time-series ultrasound video data.

**Questions:**

- Is general reasoning data mixed during post-training or added as a separate fine-tuning phase?
- Does the 0.8:0.2 ratio remain optimal across different models or modalities such as CT or MRI?
- How does Dolphin perform on some other tasks on general biomedical domain? Such as text-only benchmarks like MedQA. Will there be a degradtion of improvement in performance compared to the base model?

---

> ### Author Response · Authors · 2025-11-20
>
> Thank you for your valuable feedback on our paper!
>
> **Response to Question 1:**
> Our reasoning data is mixed during the post-training phase, as illustrated in the top-left corner of Figure 2. Additionally, we have provided a detailed description of the data used in the post-training stage and their respective proportions in Section 3.1. Thank you for your suggestion.
>
> ---
>
> **W1 & Q2:**
> The 0.8:0.2 ratio for mixing data is an empirical setting commonly adopted in large-scale supervised fine-tuning (SFT), such as in the technical report of LLaMA 3.1 Instruct, which explicitly states a Reasoning and Tools Data ratio of 21.19%. We have also conducted ablation studies on different mixing ratios, with detailed results presented in Appendix A.5. Our findings indicate that model performance does not scale linearly with the proportion of reasoning data.
>
> Currently, there is no prior work discussing optimal mixing ratios specifically for MRI and CT domains. However, we believe that a 20% ratio of reasoning data is reasonable in these medical imaging contexts. This is because constructing long-chain reasoning data without inducing hallucinations is extremely challenging in medicine, where datasets often consist of numerous precise short QA pairs. Excessive long-chain reasoning data may hinder the model’s ability to learn new visual medical knowledge, while too little may fail to activate its reasoning capabilities.
>
> ---
>
> **W2:**
> In Appendix J, we present a comparison between the outputs of our model and GPT-5, with annotations from medical experts indicating which responses are clinically reasonable or flawed. The results show that our model outperforms GPT-5 in diagnostic accuracy across most anatomical regions.
>
> Furthermore, as detailed in Appendix B, we conducted a clinical expert evaluation study. It consists of two main components. First, for clinical plausibility assessment, we randomly selected 30 ultrasound images and generated diagnostic outputs using both Dolphin and GPT-5. Two board-certified radiologists with over five years of experience in ultrasound independently rated the clinical relevance of each response on a 5-point scale (5 being the highest). The results show that Dolphin (4.0 ± 0.276) significantly outperformed GPT-5 (3.73 ± 0.890) in both average score and consistency. Notably, Dolphin’s lowest score was 3, whereas GPT-5 received several scores as low as 1, indicating that Dolphin delivers more reliable diagnostic performance.
>
> Additionally, we conducted a comparative study between Dolphin and human experts. Specifically, we constructed a web-based ultrasound question bank using a subset of u2-bench and invited eight human experts to complete the test. Results show that Dolphin achieved an accuracy of 73.5%, surpassing the human experts’ average of 60.63%. In the VAR task, Dolphin scored 60.2%, again exceeding the human average of 52.08%, demonstrating Dolphin's superior capability in understanding ultrasound images.
>
> ---
>
> **W3:**
> Currently, there is a lack of standardized benchmarks for evaluating models on ultrasound video data, making it difficult to quantitatively assess video understanding capabilities. However, our model has been evaluated on existing benchmarks that test multi-image comprehension. On u2-bench, our model demonstrates outstanding performance, and illustrative cases of multi-image understanding are provided in Appendix I, which indirectly reflect its ability to process sequential visual inputs. We plan to continue this line of research by developing dedicated benchmarks for ultrasound video evaluation in the future. Thank you for your suggestion.
>
> ---
>
> **Q3:**
> We evaluated two general medical assessment datasets, MedFramQA and VQA-RAD. On these two datasets, Dolphin demonstrated significant advantages over models like Med-LLaVA, GPT-5, and Gemini 1.5, with particularly outstanding performance on the ultrasound subset of MedFramQA. For MedQA, we have supplemented the test results and included them in the subsequent comments.

---

> > ### Author Response · Authors · 2025-11-26
> >
> > Dear reviewer,
> >
> > We have updated the evaluation results of Medllava in the paper, including the evaluation results on U2-Bench and general datasets. We found that Medllava performs well in the lesion localization task of U2-Bench but performs poorly in the diagnosis and section recognition tasks. Medllava's U2-score is 0.32, leading over models like lingshu and MedGamma, but still has a significant gap with Dolphin's 0.5835. On the general test benchmark, Medllava performs poorly, with room for improvement in instruction-following ability, significantly lagging behind general models such as GPT, Gemini, Qwen, and Dolphin across multiple datasets. We believe this is largely related to the data ratio, training strategy, and lack of reasoning ability during Medllava's training. We also added the test results of MedQA, where Dolphin achieved an accuracy of over 70% on both subsets, while Medllava only managed 41.01% and 25.66%. Compared to the base model Qwen 72B, Dolphin performs better on general medical datasets such as MedQA_MCMLE, VQA-RAD, and MedFramQA, but not on MedQA_USMLE. It's worth noting that we have better robustness, and we did not specifically optimize for general medical question-answering tasks. We have added the evaluation results to Table 1 and Appendix D for easy comparison.
> >
> > Thank you for your valuable suggestions, and we look forward to your further feedback.

---

> ### Author Response · Authors · 2025-11-27
>
> Dear Reviewer,
>
> Hello, we have supplemented the U2-bench results for Qilin-Med. During the evaluation, we noticed that Qilin-Med can only output responses in Chinese. Therefore, we additionally translated its outputs into English before conducting the assessment. Below are the test results:
> Qilin-Med-VL & 0.1952 & 0.0224 & 0.3004 & 0.0400 & 0.3227 & 0.2665 & 0.2800 & 0.3296 & 0.2997 & 29.7608 & 0.0001 & 3.2969 & 61.6669 & 1.5770 & 5.7315 & 66.5793 & 0.2608
>
> The U2-score is 0.2608. For comparison, Dolphin's U2-score is 0.58. In practice, we found that the response quality of Qilin-Med is not ideal, and it can only reply in Chinese, which is why we did not include it in the main text table.
>
> We look forward to your feedback. Wishing you a wonderful day!

---

> ### Author Response · Authors · 2025-12-01
> **Summary**
>
> **Summary:**
>
> The reviewers identified the following main shortcomings in our work:
> 1. **Lack of ablation study and explanation for the inference data ratio**—how the inference data was incorporated.
> 2. **Need for additional clinical relevance and expert evaluation.**
> 3. **Requirement to supplement MedQA evaluation results and performance on temporal reasoning tasks.**
>
> **Our adjustments in response:**
> 1. We included an **ablation study on data ratios** in the supplementary materials, referencing **Gemini’s technical report** to justify the **0.8:0.2 split** as an empirically supported baseline.
> 2. We added **two clinical experiments**:
>    - **Diagnostic comparison** between Dolphin and human doctors.
>    - **Expert evaluation** of clinical relevance, comparing Dolphin and GPT-5.
>    - **Results**: Dolphin **outperformed human doctors** in diagnostic accuracy. In clinical relevance assessment (scored out of 5), Dolphin achieved **higher average scores and greater stability** than GPT-5 (Dolphin’s **minimum score: 3** vs. GPT-5’s **minimum score: 1**).
> 3. We supplemented **MedQA results in the appendix**:
>    - Dolphin **outperformed baseline models on one test subset** but was slightly weaker on another, though with **superior overall stability**.
>    - We included **multi-image subjective results** (appendix figures) and evaluated **temporal reasoning ability indirectly via multi-image comprehension tasks**.

---

### Official Review · Reviewer_B8y3 · 2025-11-01

**Soundness:** 3
**Presentation:** 3
**Contribution:** 2
**Rating:** 4
**Confidence:** 4

**Summary:**

The paper presents Dolphin, an MLLM for ultrasound understanding with chat and reasoning modes. The model uses a three-stage pipeline that includes post-training on ultrasound and general data, instruction tuning for dialogue and step-by-step analysis, and reinforcement learning with Ultrasound Answer Reward Preference Optimization, named UARPO, to shape reasoning. The training corpus is built from textbooks, public ultrasound sets, distilled knowledge, and general sources, and is standardized with the Dolphin Ultrasound Data Protocol, or DUDP. The model is evaluated on the U2-Bench and various general and medical imaging tasks, outperforming existing models, as shown by state-of-the-art U2-Bench results. The study also analyzes the impact of mixed-domain reasoning data and the theoretical grounding of cross-domain reasoning transfer, highlighting the emergence of transferable reasoning capabilities in ultrasound tasks.

**Strengths:**

1. The analysis of cross-domain reasoning transfer is clear and thought-through. The paper gives a Bayesian account of how a prompt that asks for deep reasoning shifts the prior over chains of thought and can help even without in-domain reasoning data. The text explains the interference-to-transfer transition and ties it to a sufficient share of reasoning data, which matches the reported trend.

2. The paper reports top U2-Bench performance with a U2-score of 0.5835 and shows consistent benefits from the deep reasoning mode on diagnosis accuracy, detection accuracy, and measurement error. It also states that the system has been integrated on real ultrasound devices, which speaks to practical relevance.

**Weaknesses:**

1. The pipeline uses GPT-4 semantic screening, another LLM for response checks, and a brief mention of medical expert validation, yet the paper does not report who the experts are, how many experts participated, what specialties they represent, what rubric or thresholds were used, or any inter-rater agreement. This makes it hard for peers to reproduce, audit bias, or estimate error rates. Given ultrasound’s dependence on subtle imaging cues like artifacts and view selection, reliance on general LLM judges without detailed safeguards raises the risk that flawed pairs enter the data at scale.


2. The set of medical baselines in Table 1 is narrow. The medical-specific block lists MiniGPT-Med and MedDr, while several commonly cited medical models are absent. As a result, the state-of-the-art claim on U2-Bench rests on a comparison that omits strong medical references, which weakens the strength of the claim for the medical VLM landscape.

3. Table 1 lists only MiniGPT-Med and MedDr in the Medical-Specific Open-Source Models block, while widely used medical baselines such as LLaVA-Med and Huatuo-O1 are missing (e.g., I think LLaVA-Med can achieve more than 60% acc on VQA-Rad). Instead, it places LLaVA-1.5-13B under general open-source models, not the medical variant. As a result, the comparison set is narrow and likely weaker, which makes it difficult to judge whether the reported gains truly represent state-of-the-art performance in the medical VLM landscape.

4. Some of the images in Figure 4 have been compressed, affecting readability.

**Questions:**

Please see the weaknesses above.

---

> ### Author Response · Authors · 2025-11-20
>
> Thank you for your valuable feedback on our article!
>
> Question 1:
>
> Our medical experts consist of three experienced radiologists from the ultrasound department, each with at least five years of clinical expertise. These experts were responsible for validating the quality of the constructed question-answer (QA) pairs, focusing on linguistic style, overall clarity, and whether the diagnostic information was accurately reflected. A QA pair was only accepted if all three experts reached a consensus; otherwise, we iteratively refined our data construction process to resolve discrepancies.
>
> The expert review primarily focused on QA pairs derived from textbook materials and publicly available datasets. For public datasets, we carefully selected those with established ground-truth annotations to ensure accuracy. For textbook-based QA pairs, the physicians verified that all questions and answers were correctly extracted and logically grounded in the source material. We have incorporated this information into the Data Filtering section of the revised manuscript. Thank you for this valuable suggestion.
>
> Question 2:
>
> We evaluated several state-of-the-art medical-specific models, including MiniGPT-Med, MedDr, Lingshu-7B, and MedGemma-4B. Our model consistently outperformed these specialized models, demonstrating Dolphin’s superior capability in the domain of ultrasound.
>
> Question 3:
>
> In Table 1, we report comparative results against the aforementioned medical-specific models (MiniGPT-Med, MedDr, Lingshu-7B, and MedGemma-4B). It is important to note that our model is specifically optimized for ultrasound applications and has not been fine-tuned for other imaging modalities such as MRI or CT. The evaluation on VQA-RAD was included to demonstrate that our model retains strong general medical reasoning capabilities, even outside its primary domain—achieving performance competitive with or superior to general-purpose models like GPT and Gemini.
>
> In response to your suggestion, we are currently conducting additional experiments using Med-LlaVA on U2-Bench, and the results will be included in the final version of the paper.
>
> Question 4:
>
> Thank you for your correction. Figure 4 has now been enlarged to its original resolution for improved readability.

---

> > ### Author Response · Authors · 2025-11-26
> >
> > Dear reviewer,
> >
> > We have updated the evaluation results of Medllava in the paper, including the evaluation results on U2-Bench and general datasets. We found that Medllava performs well in the lesion localization task of U2-Bench but performs poorly in the diagnosis and section recognition tasks. Medllava's U2-score is 0.32, leading over models like lingshu and MedGamma, but still has a significant gap with Dolphin's 0.5835. On the general test benchmark, Medllava performs poorly, with room for improvement in instruction-following ability, significantly lagging behind general models such as GPT, Gemini, Qwen, and Dolphin across multiple datasets. We believe this is largely related to the data ratio, training strategy, and lack of reasoning ability during Medllava's training. We also added the test results of MedQA, where Dolphin achieved an accuracy of over 70% on both subsets, while Medllava only managed 41.01% and 25.66%. Compared to the base model Qwen 72B, Dolphin performs better on general medical datasets such as MedQA_MCMLE, VQA-RAD, and MedFramQA, but not on MedQA_USMLE. It's worth noting that we have better robustness, and we did not specifically optimize for general medical question-answering tasks. We have added the evaluation results to Table 1 and Appendix D for easy comparison.
> >
> > Thank you for your valuable suggestions, and we look forward to your further feedback.

---

> ### Author Response · Authors · 2025-11-27
>
> Dear Reviewer,
>
> Hello, we have supplemented the U2-bench results for Qilin-Med. During the evaluation, we noticed that Qilin-Med can only output responses in Chinese. Therefore, we additionally translated its outputs into English before conducting the assessment. Below are the test results:
> Qilin-Med-VL & 0.1952 & 0.0224 & 0.3004 & 0.0400 & 0.3227 & 0.2665 & 0.2800 & 0.3296 & 0.2997 & 29.7608 & 0.0001 & 3.2969 & 61.6669 & 1.5770 & 5.7315 & 66.5793 & 0.2608
>
> The U2-score is 0.2608. For comparison, Dolphin's U2-score is 0.58. In practice, we found that the response quality of Qilin-Med is not ideal, and it can only reply in Chinese, which is why we did not include it in the main text table.
>
> We look forward to your feedback. Wishing you a wonderful day!

---

> ### Author Response · Authors · 2025-12-01
> **Summary**
>
> **Summary:**
>
> The reviewers identified the following main shortcomings in our work:
> 1. The data filtering and expert validation process was not described in detail.
> 2. Comparative testing of Dolphin against other medical VLMs needs to be supplemented, including MedQA results.
>
> **Our adjustments in response:**
> 1. We added a detailed explanation of the **data filtering and expert validation process** in the **"Data Filtering"** section of the paper.
> 2. We supplemented the comparative testing by evaluating **LLaVA-Med** and **Qilin-Med** on **U2-Bench**, as well as testing **LLaVA-Med** on **three general medical datasets (VQA-RAD, MedFramQA, and MedQA)**. The results show that LLaVA-Med underperforms Dolphin across all these benchmarks (including U2-Bench).
>
> **Current scope of comparisons:**
> - **5 medical-specialized VLMs** tested.
> - **3 general medical datasets** evaluated (covering both **VLM-based and text-only assessments**).

---

### Official Review · Reviewer_GLPu · 2025-11-03

**Soundness:** 3
**Presentation:** 2
**Contribution:** 3
**Rating:** 4
**Confidence:** 4

**Summary:**

This paper proposes DOLPHIN, a multimodal large model for ultrasound understanding. The authors construct a large-scale, multi-source training corpus and define the Dolphin Ultrasound Data Protocol (DUDP) to unify data formats across various ultrasound tasks. The model is trained in three stages: post-training, instruction tuning, and reinforcement learning via UARPO. The paper reports a SOTA U2-score of 0.5835 on the U2-Bench, and observes that incorporating cross-domain deep reasoning data improves ultrasound reasoning capabilities. A Bayesian formulation with latent variable \(c\) is proposed to explain how deep reasoning prompts act as a prior guiding transfer. Additionally, the reasoning mode significantly outperforms the standard mode in diagnosis, detection, and measurement tasks.

**Strengths:**

1. DUDP provides a unified structured JSON format for multi-task/multimodal ultrasound data, facilitating training, evaluation, and reproducibility.
2. Cross-domain reasoning transfer is explored via two system prompts (vanilla vs. deep reasoning), revealing a U-shaped transition curve and backed by a Bayesian framework.
3. Comprehensive evaluation on 8 U2-Bench tasks and general/medical VQA benchmarks. Reasoning mode shows gains in diagnosis, detection, and measurement.
4. Deployment on real ultrasound devices is claimed, suggesting promising application potential.

**Weaknesses:**

1. Data size and source inconsistency: Abstract claims ">2,000,000" instruction-response pairs; the body and figures refer to both 2B (billion) and 2M, which conflicts. Clear explanation of counting method and usable data ratio is needed.
2. Key data curation details are missing: What are the exact sources of teaching materials? Are licenses obtained? What are the annotators' qualifications? Was annotation agreement assessed? Source ratio, deduplication strategy, and bias in synthetic/distilled data are not discussed.
3. Lack of statistical robustness: No significance testing, no variance across multiple seeds, and unclear if reported gains are beyond randomness.
4. Limited comparability and reproducibility: Comparisons with closed-source models (GPT-4/5, Gemini) lack disclosure of settings (prompt format, context length, decoding parameters, image resolution, CoT/multi-turn settings). This affects the fairness of evaluation.
5. Too few comparisons with other medical multimodal models: Only two models are compared. Classical medical VLMs such as LLaVA-Med, Qilin-Med-VL, LLaVA-Ultra, and EchoCLIP should be included.
6. Ethics and compliance concerns are insufficiently addressed: Textbook licensing, public data usage rights, and the legal status of synthetic/distilled content are not clarified. Real device deployment lacks information about clinical trials, safety measures, and human-in-the-loop design.

**Questions:**

1. Please clarify the number of effective samples used in each stage (post-training, IFT, RLHF-like), and the ratio of public vs. private vs. synthetic/distilled data. How were duplicates handled?
2.  Did you consider continuous rewards (e.g., IoU for detection, L2 for keypoints, MAE for measurements)? Does binary reward cause policy collapse or overfitting to format?
3. Please report all main results with mean ± std over 3–5 random seeds. Are U2-Bench improvements statistically significant?
4. Were decoding parameters (temperature, top-p, max tokens), image preprocessing (resolution, cropping), system prompts, CoT/multi-turn settings consistent across models? If not, please provide detailed comparisons.
5. Are textbooks and guidelines licensed for use? How were clinical/public datasets de-identified?
6. Please provide qualitative failure cases (image/video + reasoning steps) and analyze failure causes (non-standard views, artifacts, occlusion, parameter mismatch).
7. What are the latency, memory, and token usage differences between reasoning and standard modes? Are they acceptable for deployment?
8. Section 2 Data Curation: Which versions of GPT and DeepSeek were used for distillation? How were "overly short responses" or "excessive images" defined? What is the data volume before and after filtering?

**Details Of Ethics Concerns:**

- Data compliance: No license or permission details are given for textbooks/guidelines. The legality of public ultrasound dataset use is unclear.
- Distilled model rights: Knowledge and reasoning distilled from GPT/DeepSeek raise unclear IP and usage issues.
- Clinical deployment: No evidence of IRB approval, clinical trials, or human oversight for real-world use.

---

> ### Author Response · Authors · 2025-11-20
>
> Thank you for your valuable feedback on our paper!
>
> W1: The Abstract incorrectly stated ">2,000,000" pairs; the correct number is 2 million (2M), as reflected in the main text and figures. The typo ("2B") in the Introduction has been corrected (line 75).
>
> Q1: Figure 2 illustrates the detailed data distribution, including publicly available datasets used for training, the proportion of distilled data, and the number of samples used in each training stage. We acknowledge that the lack of textual explanation may have caused confusion for readers. Therefore, we have added comprehensive descriptions in Sections 3.1, 3.2, and 3.3. Since our data construction method is task-based, it does not contain duplicated QA pairs. However, the same image may be used across different tasks—such as anatomical localization and disease detection—to train the model’s ability to recognize diverse aspects of ultrasound images.
>
> Q2:  Currently, we use UARPO primarily to enhance the model's overall ultrasound understanding capability. For measurement-related tasks, we mainly formulate them as multiple-choice questions, asking the model to select the closest option. The main challenge with continuous rewards lies in designing task-specific reward functions, which requires extensive exploration—we plan to investigate this in future work. That said, we have observed that modeling these complex ultrasound tasks using UAR leads to a steadily decreasing loss, steadily improving accuracy, and indeed reduces RMSE on measurement tasks.
>
> Q3: Following your suggestion, we have added the mean and variance of results across four random seeds in Appendix B.3. The results show that Dolphin achieves an average U2-score of 0.58 with a variance of 0.01 across four random seeds, indicating consistent and generalizable performance.
>
> W4 & Q4: For all evaluations, we default to temperature = 0, top_p = None, max_tokens = 2048, consistent system prompts, and input resolution of 512×512. We have now included these evaluation settings in Section 5 of the paper. Thank you for pointing this out.
>
> W5: In fact, we have compared our model against several specialized medical models, including MiniGPT-Med, MedDr, Lingshu-7B, and MedGemma-4B, and our method outperforms all of them. We apologize that the previous table formatting was unclear and could have misled readers. Regarding Enco-CLIP, it is based on the CLIP architecture and thus difficult to evaluate in our generation-focused setup. LLava-ultra has not yet released its weights, making evaluation impossible at this time. Qilin-Med-VL is designed specifically for Chinese, while our benchmark datasets are in English. We are currently evaluating LLava-Med and will include the results in the final version of the paper.
>
> W2 & Q5: All datasets we used are publicly available educational resources. Importantly, the QA pairs we constructed were generated by large language models based on knowledge learned from textbooks—not direct copies of original textbook content. We have carefully reviewed the licensing terms of all datasets and confirmed that each is used within permitted bounds. These details have been added to Appendix I.
>
> W6 & Q9: We have reviewed the licenses of all datasets used and confirmed that every dataset is covered under a granted license; details are provided in Appendix I. For components involving human subjects, we obtained ethical approval from the collaborating hospital. The corresponding ethics approval number will be included in the formal version of the paper.
>
> Q6: We have added case studies and analysis of failure cases in Appendix C. During testing, we found that minor artifacts do not significantly affect model performance, demonstrating that Dolphin possesses genuine image understanding capabilities. The main limitations stem from: (1) scarcity of rare disease samples in public datasets, (2) dependence between ultrasound images and patient reports, and (3) limitations of static image modalities. For example, in Case 3, the model explicitly noted the need to incorporate color Doppler imaging for accurate inference.
>
> Q7: The memory footprint of the reasoning mode is identical to that of standard mode. However, responses generated in reasoning mode are significantly longer on average. Based on our deployment tests, a 7B model can be successfully deployed on a single GPU with 48GB VRAM (e.g., RTX 4090) and meet inference requirements under reasoning mode. A 72B model, however, would require server-level deployment due to higher resource demands.
>
> Q8: We used GPT-4 and DeepSeek-r1 for data generation. "Overly short responses" refer to replies with fewer than 5 characters (excluding multiple-choice answers), and "excessive images" refer to QA pairs involving more than 7 images. These filters were applied primarily to reduce GPU memory consumption during training. Our entire filtering process occurs during the data generation phase—if low-quality data is detected, it is discarded immediately.

---

> > ### Author Response · Authors · 2025-11-26
> >
> > Dear reviewer,
> >
> > We have updated the evaluation results of Medllava in the paper, including the evaluation results on U2-Bench and general datasets. We found that Medllava performs well in the lesion localization task of U2-Bench but performs poorly in the diagnosis and section recognition tasks. Medllava's U2-score is 0.32, leading over models like lingshu and MedGamma, but still has a significant gap with Dolphin's 0.5835. On the general test benchmark, Medllava performs poorly, with room for improvement in instruction-following ability, significantly lagging behind general models such as GPT, Gemini, Qwen, and Dolphin across multiple datasets. We believe this is largely related to the data ratio, training strategy, and lack of reasoning ability during Medllava's training. We also added the test results of MedQA, where Dolphin achieved an accuracy of over 70% on both subsets, while Medllava only managed 41.01% and 25.66%. Compared to the base model Qwen 72B, Dolphin performs better on general medical datasets such as MedQA_MCMLE, VQA-RAD, and MedFramQA, but not on MedQA_USMLE. It's worth noting that we have better robustness, and we did not specifically optimize for general medical question-answering tasks. We have added the evaluation results to Table 1 and Appendix D for easy comparison.
> >
> > Thank you for your valuable suggestions, and we look forward to your further feedback.

---

> ### Author Response · Authors · 2025-11-27
>
> Dear Reviewer,
>
> Thank you for your feedback. As per your suggestion, we have conducted additional testing on Qilin-Med's U2-bench results. During the evaluation, we noticed that Qilin-Med can only output responses in Chinese. Therefore, we translated its outputs into English before performing the assessment. Below are the test results:
>
> Qilin-Med-VL & 0.1952 & 0.0224 & 0.3004 & 0.0400 & 0.3227 & 0.2665 & 0.2800 & 0.3296 & 0.2997 & 29.7608 & 0.0001 & 3.2969 & 61.6669 & 1.5770 & 5.7315 & 66.5793 & 0.2608
>
> The U2-score for Qilin-Med is 0.2608. For comparison, Dolphin's U2-score is 0.58.
>
> In practice, we observed that the response quality of Qilin-Med is not ideal, and it is limited to Chinese replies. For these reasons, we did not include these results in the main table of the paper.
>
> We look forward to your further comments. Wishing you a wonderful day!

---

> ### Author Response · Authors · 2025-12-01
> **Summary**
>
> **Summary:**
>
> The reviewers identified the following main shortcomings in our work:
> 1. The data proportions were not clearly explained, and the sources of public data were not disclosed.
> 2. Comparative testing with Dolphin against other medical VLMs needs to be supplemented, including MedQA results and results under multiple random seeds.
> 3. Ethical considerations need to be addressed.
>
> **Our adjustments in response:**
> 1. We added detailed descriptions of the training data usage in the paper and included a table in the appendix listing the sources of public data.
> 2. We supplemented the testing by comparing with **LLaVA-Med** and **Qilin-Med**, as well as **MedQA**. (*LLaVA-Ultra* is not open-source, and *EchoCLIP* is a CLIP-based model, making evaluation infeasible.) We also added results from **four random seeds** and included them in the appendix.
> 3. We supplemented the **licenses of public datasets** in the appendix.
>
> **Additionally, we made the following improvements:**
> - Added **test-related configurations** (e.g., hyperparameters, evaluation protocols).
> - Provided **two clinical validation experiments** to demonstrate real-world applicability.
> - Included **three failure cases** with analysis to enhance transparency.
> - Added **deployment details** (e.g., inference settings, computational requirements).

---

### Note · Program_Chairs · 2026-01-17
**Submission Desk Rejected by Program Chairs**

The following references in this submission do not refer to real documents and/or have major errors in bibliographic information:

 Chaoyi Chao, Xiaoman Lin, Haodong Liu, Dinggang Shen, et al. Radfm: A foundation model for radiology understanding. arXiv preprint, 2024.